# Echolocation-related reversal of information flow in a cortical vocalization network

Francisco García-Rosales [1,2 ✉], Luciana López-Jury [1], Eugenia González-Palomares[1], Johannes Wetekam [1], Yuranny Cabral-Calderín [3], Ava Kiai[1], Manfred Kössl[1] & Julio C. Hechavarría [1 ✉]

The mammalian frontal and auditory cortices are important for vocal behavior. Here, using local-field potential recordings, we demonstrate that the timing and spatial patterns of oscillations in the fronto-auditory network of vocalizing bats (*Carollia perspicillata*) predict the purpose of vocalization: echolocation or communication. Transfer entropy analyses revealed predominant top-down (frontal-to-auditory cortex) information flow during spontaneous activity and pre-vocal periods. The dynamics of information flow depend on the behavioral role of the vocalization and on the timing relative to vocal onset. We observed the emergence of predominant bottom-up (auditory-to-frontal) information transfer during the post-vocal period specific to echolocation pulse emission, leading to self-directed acoustic feedback. Electrical stimulation of frontal areas selectively enhanced responses to sounds in auditory cortex. These results reveal unique changes in information flow across sensory and frontal cortices, potentially driven by the purpose of the vocalization in a highly vocal mammalian model.

[1] Institut für Zellbiologie und Neurowissenschaft, Goethe-Universität, 60438 Frankfurt/M, Germany. [2] Ernst Strüngmann Institute (ESI) for Neuroscience in Cooperation with Max Planck Society, 60528 Frankfurt am Main, Germany. [3] Research Group Neural and Environmental Rhythms, Max Planck Institute for Empirical Aesthetics, 60322 Frankfurt/M, Germany. ✉email: francisco.garcia-rosales@esi-frankfurt.de; hechavarria@bio.uni-frankfurt.de

Vocal production is a crucial behavior that underlies the evolutionary success of various animal species. Several cortical and subcortical structures in the mammalian brain support vocalization[1], their activities related to vocal control[2–4], motor preparation[2,5,6], and feedback correction[7,8]. However, the precise neural dynamics that underpin vocalization, and the nature of long-distance interactions in large-scale neural networks related to vocal utterance, remain poorly understood.

The connectivity patterns of the frontal cortex make it a major hub for cognitive control and behavioral coordination[9–11]. Frontal cortical areas are anatomically connected with structures directly involved in vocal production, such as the periaqueductal gray[12] and the dorsal striatum[13]. Experimental evidence demonstrates that the neural activity in frontal regions relates to vocalization[4,14–16], correlating with the acoustic and behavioral properties of produced calls[14,16]. Frontal regions are also anatomically and functionally connected with the auditory cortex (AC; refs. [17–22]), which exhibits suppression to self-produced sounds generated by body movements[23–26] or vocalizations[27–30]. This suppression is thought to be mediated by preparatory signals originating in the motor system (i.e., "corollary discharges" or "efference copies"; refs. [31–33]). The attenuation of neural responses in AC during vocal production supports precise vocal control by means of feedback mechanisms[7,8] in coordination with frontal cortical areas[34–37]. Although current evidence shows that fronto-auditory circuits are essential for accurate vocalization control, neural interactions in these networks remain obscure.

In this study, we addressed the neural mechanisms of vocal production in the fronto-auditory system of a highly vocal mammal: the bat *Carollia perspicillata*[38–41]. Bats constitute an excellent model to study vocalization because they rely heavily on vocal behavior for communication and navigation (i.e., echolocation). Communication and echolocation utterances are emitted for different behavioral purposes, and typically differ in their spectrotemporal design[40]. The production of these calls is distinctly controlled at the level of the brainstem[42], possibly mediated by frontal cortical circuits involving regions such as the anterior cingulate cortex[43] and the frontal-auditory field (FAF; ref. [16]).

Vocalization circuits were studied by measuring local-field potential (LFP; ref. [44]) oscillations simultaneously in frontal and auditory cortices of vocalizing bats. In frontal and sensory cortices, LFPs are involved in cognitive processes, sensory computations, and interareal communication via phase coherence[17,45–49]. In the FAF, a richly connected auditory region of the bat frontal cortex[20,50], LFP activity predicts vocal output and synchronizes differentially with dorso-striatal oscillations according to vocalization type[16]. Oscillations in the frontal cortex synchronize across socially interacting bats[51], and are involved in the cognitive aspects of social exchange[52]. The roles of auditory cortical oscillatory activity in vocal production are less clear, although human studies suggest that oscillations mediate synchronization with frontal and motor areas for feedback control[35,53,54]. However, the precise dynamics of information exchange in the fronto-auditory circuit during vocalization are unknown.

The goal of this study was to unravel the nature of information exchange in the bat's FAF-AC network, and to understand whether information between these structures flows in accordance with the canonical roles of the frontal cortex for vocal coordination, and of the AC for feedback control. We further aimed to address whether the distinct behavioral contexts of echolocation and communication affect the dynamics of information transfer in the fronto-auditory circuit. We found complex causal interactions (within a transfer entropy framework) between frontal and auditory cortices during spontaneous activity and periods of vocal production. Directed connectivity in the FAF-AC network varied dynamically according to whether animals produced communication or echolocation calls, and to the timing relative to vocal onset. For echolocation the changes were drastic, resulting in a reversal of information flow from pre-vocal to post-vocal periods. Our data suggest that dynamic information transfer patterns in large-scale networks involved in vocal production, such as the FAF-AC circuit, are shaped by the behavioral consequences of produced calls.

## Results

Neural activity was studied in the FAF and AC of *C. perspicillata* bats (3 males) while animals produced self-initiated vocalizations. From a total of 12494 detected vocalizations, 138 echolocation and 734 non-specific communication calls were preceded by a period of silence lasting at least 500 ms and were therefore considered for subsequent analyses. Representative echolocation and communication vocalizations are shown in Fig. 1a. Overall, the two types of vocalizations did not differ significantly in terms of call length (Wilcoxon rank sum test, $p = 0.78$; Fig. 1b), although call length distributions differed significantly (2-sample Kolmogorov-Smirnov test, $p = 7.93 \times 10^{-7}$). There were clear differences in the power spectra of echolocation and communication calls (Fig. 1c, left), such that peak frequencies of echolocation utterances were significantly higher than their communication counterparts (Wilcoxon rank sum test, $p = 2.24 \times 10^{-66}$; Fig. 1c, right). These differences are consistent with the structure of echolocation and communication sounds in bats (*C. perspicillata*) described in previous studies[40,55].

**Oscillations in frontal and auditory cortices predict vocalization type**. Figure 1d illustrates electrophysiological activity recorded simultaneously from FAF and AC at various cortical depths, as the echolocation and communication vocalizations shown in Fig. 1a were produced (see location of recording sites in Fig. S1). Single-trial LFP traces revealed conspicuous pre-vocal oscillatory activity in low and high-frequencies, more pronounced in frontal regions, and strongest when animals produced echolocation pulses. Power spectral densities (PSD) obtained from pre-vocal LFP segments (i.e., −500 to 0 ms relative to vocal onset; Fig. 1f) indicated low- and high-frequency power increases (relative to a no-vocalization baseline, or "no-voc") associated with vocal production, particularly in FAF and for electrodes located at depths > 100 μm (Fig. 1e illustrates this at 300 μm; black arrows). Differences in AC across types of vocal outputs were less pronounced and appeared limited to low LFP frequencies (gray arrows in Fig. 1e). These pre-vocal spectral patterns were analyzed using canonical LFP frequency bands, namely: delta (δ), 1–4 Hz; theta (θ), 4–8 Hz; alpha (α), 8–12 Hz; low beta ($\beta_1$), 12–20 Hz; high beta ($\beta_2$), 20–30 Hz; and three sub-bands of gamma (γ): $\gamma_1$ (30–60 Hz), $\gamma_2$ (60–120 Hz), and $\gamma_3$ (120–200 Hz). Pre-vocal LFP power in each band was normalized to no-voc periods on a trial-by-trial basis.

There were significant power changes between no-voc and pre-vocal periods across frequency bands (Figs. 1f, S2). Notably, the power increase in low- (δ-α) and high-frequency ($\gamma_2$) LFP bands of the FAF was different when animals produced echolocation and communication vocalizations, with the highest increase in the pre-vocal echolocation case. The opposite pattern was observed in the AC, where differences between ensuing vocalization types were most prominent in $\beta_1$ (but not δ-α or γ) frequencies, and were explained by higher pre-vocal power increase for communication than for echolocation vocalizations (Fig. 1f).

We addressed whether pre-vocal LFP power in frontal and auditory cortices was a significant predictor of ensuing call type.

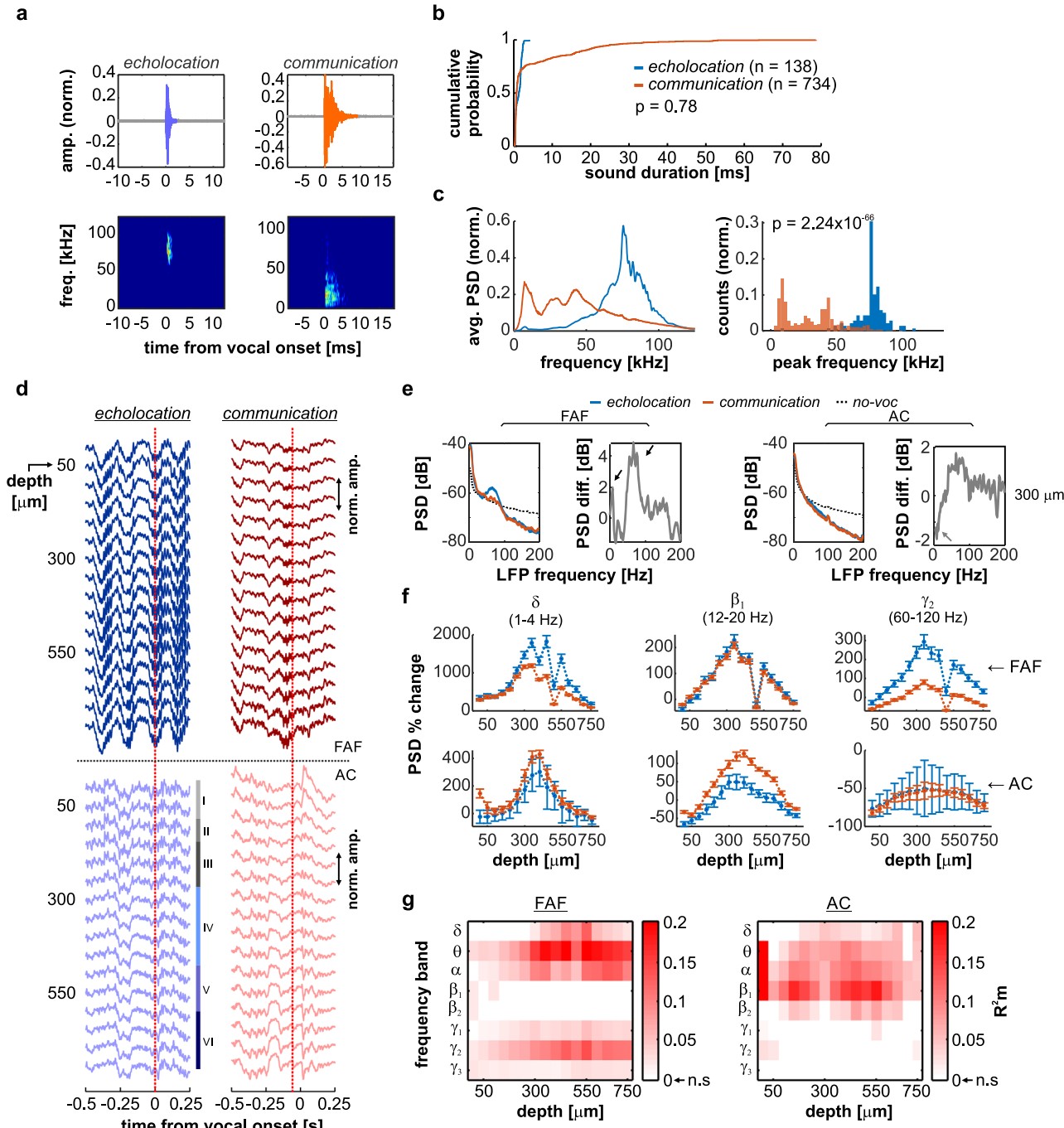

**Fig. 1 Pre-vocal oscillations in frontal and auditory cortices predict ensuing vocal output. a** Oscillograms (top) and spectrograms (bottom) of exemplary echolocation (left) and communication calls produced by *C. perspicillata*. **b** Cumulative probability distribution of echolocation (blue, *n* = 138) and communication (orange, *n* = 734) call lengths. No significance differences were observed (two-sided Wilcoxon rank sum test, *p* = 0.12). **c** (*Left*) Normalized average power spectral density (PSD) of echolocation (blue) and communication (orange) calls. (*Right*) Distribution of peak frequencies of echolocation and communication utterances; communication calls were significantly higher in frequency than their counterparts (two-sided Wilcoxon rank sum rest, $p = 2.24 \times 10^{-66}$). **d** Single-trial LFPs recorded simultaneous to echolocation and communication utterances. The vertical red dashed line, at time 0, indicates the moment of vocalization onset. The top 16 traces correspond to LFPs recorded in the FAF; the bottom 16 LFP traces were recorded from the AC. Auditory cortical layers are marked. LFP amplitude was normalized within structure independently. **e** Average pre-vocal (−500 to 0 ms, relative to call onset) power spectral densities (PSD) at a representative depth (300 μm) in FAF and AC. Blue: echolocation; orange: communication; black, dashed: no-voc periods. The difference between echolocation and communication PSDs is depicted in gray (right). **f** Percentage pre-vocal power change across representative LFP bands (δ, 1–4 Hz; $\beta_1$, 12–20 Hz; $\gamma_2$, 60–120 Hz), relative to a no-voc baseline, across cortical depths in FAF (top) and AC (bottom). Values related to echolocation utterances (*n* = 138) are depicted in blue; those related to communication utterances (*n* = 734) are depicted in orange. Data shown as mean ± sem. **g** Pre-vocal power change in frontal and auditory regions predict vocalization type. Effect size ($R^2m$) of GLMs considering all frequency bands and channels, both in frontal and auditory cortices. Effect sizes were considered small when $R^2m < 0.1$, and medium for $R^2m >= 0.1$. For illustrative purposes, effect size values from non-significant models were set to 0. Source data are provided as a Source Data file.

Generalized linear models (GLMs) were fit using echolocation and communication pre-vocal power changes as predictors, for all channels and frequency bands. A summary of these models is given in Fig. 1g (outcomes of two representative GLMs illustrated in Fig. S2). Low- and high-frequency power increase (mostly in the δ-α and $\gamma_2$ bands) in FAF predicted whether animals produced echolocation or communication calls, typically with moderate effect sizes ($p < 0.05$; $R^2m \geq 0.1$), highest in middle-to-deep electrodes (i.e., depths > 300 μm; Fig. 1g, left). The fact that low frequencies predict ensuing call type complements a previous study in C. perspicillata's FAF[16], wherein low-frequency effects went unnoticed likely due to a local referencing scheme that affected low-frequency signals correlated across neighboring electrodes (see Fig. S13). In the AC, pre-vocal power predicted ensuing call type mostly in the α-β bands of the spectrum, although more strongly in $\beta_1$ frequencies. Moderate effect sizes were also observed ($p < 0.05$; $R^2m \geq 0.1$). Overall, pre-vocal oscillatory power significantly predicted ensuing call type in frontal and auditory cortices with complementary frequency specificity and functionally opposite effects.

We evaluated whether differences in the spectral dynamics of pre-vocal LFPs could be explained by differences in the frequency content of echolocation and communication utterances. To that end, communication calls were separated into two groups: high-frequency and low-frequency communication (HF- and LF-communication, respectively). The spectral content of pre-vocal LFPs predicted ensuing call type, even when HF-communication calls were pitched against echolocation utterances ($p < 0.05$; $R^2m \geq 0.1$; Fig. S3). Additionally, pre-vocal spectral differences were considerably less noticeable when comparing HF- vs. LF-communication vocalizations, with even significant models ($p < 0.05$) performing poorly in FAF and AC ($R^2m < 0.1$; Fig. S3b, d). Thus, pre-vocal spectral differences are not fully accounted for by differences in the spectral content of echolocation and communication utterances. However, an influence of differences in the spectral content of echolocation and communication calls on LFPs cannot definitely be excluded.

**Directed connectivity in the FAF-AC circuit related to vocal production.** Oscillations in FAF and AC predict ensuing vocal output with functionally opposite patterns, but how rhythms in this network interact during vocal production remains unknown. In previous work, we reported low-frequency (1–12 Hz) phase coherence in the FAF-AC circuit during spontaneous activity, with emergence of γ-band (>25 Hz) coherence at the onset of external acoustic stimulation[17]. To study FAF-AC oscillatory dynamics during vocalization, we looked beyond phase correlations and examined causal interactions (within a transfer entropy framework) in the fronto-auditory circuit. Causal interactions were quantified using directed phase transfer entropy (dPTE), a metric that measures the degree of preferential information transfer between signals based on phase time series[56,57]. dPTE calculations were performed across vocal conditions for all channel pairs, and for frequency bands which most strongly predicted vocalization type: δ, θ, α, $\beta_1$, and $\gamma_2$.

Average dPTE connectivity matrices across conditions (echolocation and communication pre- and post-vocal periods, and no-voc segments) are illustrated in Fig. S4. dPTE matrices were used as adjacency matrices for directed graphs, which characterized patterns of directional information flow in the FAF-AC network (Fig. 2). In each graph, nodes represent adjacent channels pooled according to cortical depth and layer distribution in AC (where layer borders are well-defined anatomically; Fig. S1): superficial (*sup*), channels 1–4 (0–150 μm); top-middle (*mid1*), channels 5–8 (200–350 μm); bottom-middle (*mid2*), channels 9–12

(400–550 μm); and *deep*, channels 13–16 (600–750 μm). Directed edges were weighted according to a directionality index (DI), obtained from normalizing dPTE values to 0.5 (dPTE = 0.5 indicates no preferred direction of information transfer). Only edges with significant DI values, based on bootstrapping, are shown.

Upon inspection of the connectivity graphs, we noticed general patterns that entailed strong top-down preferential information transfer (i.e., in the FAF → AC direction) during spontaneous activity, pre-vocal periods irrespective of vocalization type, and post-vocal communication periods (Fig. 2). Top-down information flow (blue arrows in the figure) was strongest for δ, θ, and $\gamma_2$ frequencies, although also occurred sparsely in α and $\beta_1$ bands with patterns that depended on ensuing call type. Within FAF and during pre-vocal echolocation periods, information flowed predominantly from deep to superficial layers in δ and $\beta_1$ frequencies (Fig. 2b), and in the opposite direction for α-LFPs, also during no-voc periods (Fig. 2a, b). Within AC, information flowed in the superficial to deep direction during no-voc and pre-vocal communication periods in $\gamma_2$ frequencies.

To our surprise, a predominance of bottom-up information flow (i.e., AC → FAF direction) appeared to be specific to post-vocal echolocation periods in the δ and $\beta_1$ bands, although bottom-up information transfer did occur in α frequencies during post-vocal communication epochs (Fig. 2c). Note that, for echolocation, there was strong top-down information transfer before vocalization onset, particularly in the δ-band (compare Fig. 2c with Fig. 2b, top). These results hint toward a pre- to post-vocal reversal of information flow in the FAF-AC network during echolocation, evident in low frequencies of the LFP. Considering within-structure information transfer, patterns were diverse in FAF, consisting of information exchange in the deep-to-superficial (bands: δ, echolocation and communication; α, communication; and $\beta_1$, both) and the superficial-to-deep (α-band, for echolocation) directions. Within AC, predominant information flow occurred both in the superficial-to-deep (δ-band, echolocation) and in the deep-to-superficial directions (bands: θ, echolocation; α, echolocation and communication; $\beta_1$, echolocation, $\gamma_2$, communication) as well.

Taken together, the data in Fig. 2 illustrate rich patterns of information exchange within and between frontal and auditory cortices. Information transfer patterns depended on whether a vocalization was produced and on its type, either considering within-structure connectivity, or information transfer across regions. Differences in the information flow dynamics of the fronto-auditory circuit, across vocal conditions and call-types, are depicted in Figs. S5, S6, and quantified in detail in the Supplementary Materials.

**Information flow in the fronto-auditory circuit reverses for echolocation production.** The contrasts between pre- and post-vocal periods for echolocation (Fig. 2b, c) suggest a reversal in preferred directionality of information flow. Differences in the direction of information transfer between pre-vocal and post-vocal activities were addressed by statistically comparing connectivity graphs associated to each case (Fig. 3). Paired statistics were performed for these comparisons (Wilcoxon singed-rank tests, significance when $p < 10^{-4}$; see "Methods"); edges, representing significant differences in dPTE, were only shown for large effect sizes ($|d| > 0.8$). As expected from the data depicted in Fig. 2, echolocation-related FAF → AC preferred information flow was significantly higher during pre-vocal than post-vocal periods in the δ and θ bands (Fig. 3a, top). In $\gamma_2$ frequencies, the effect was the opposite: FAF → AC directionality was highest during post-vocal periods than during pre-vocal ones.

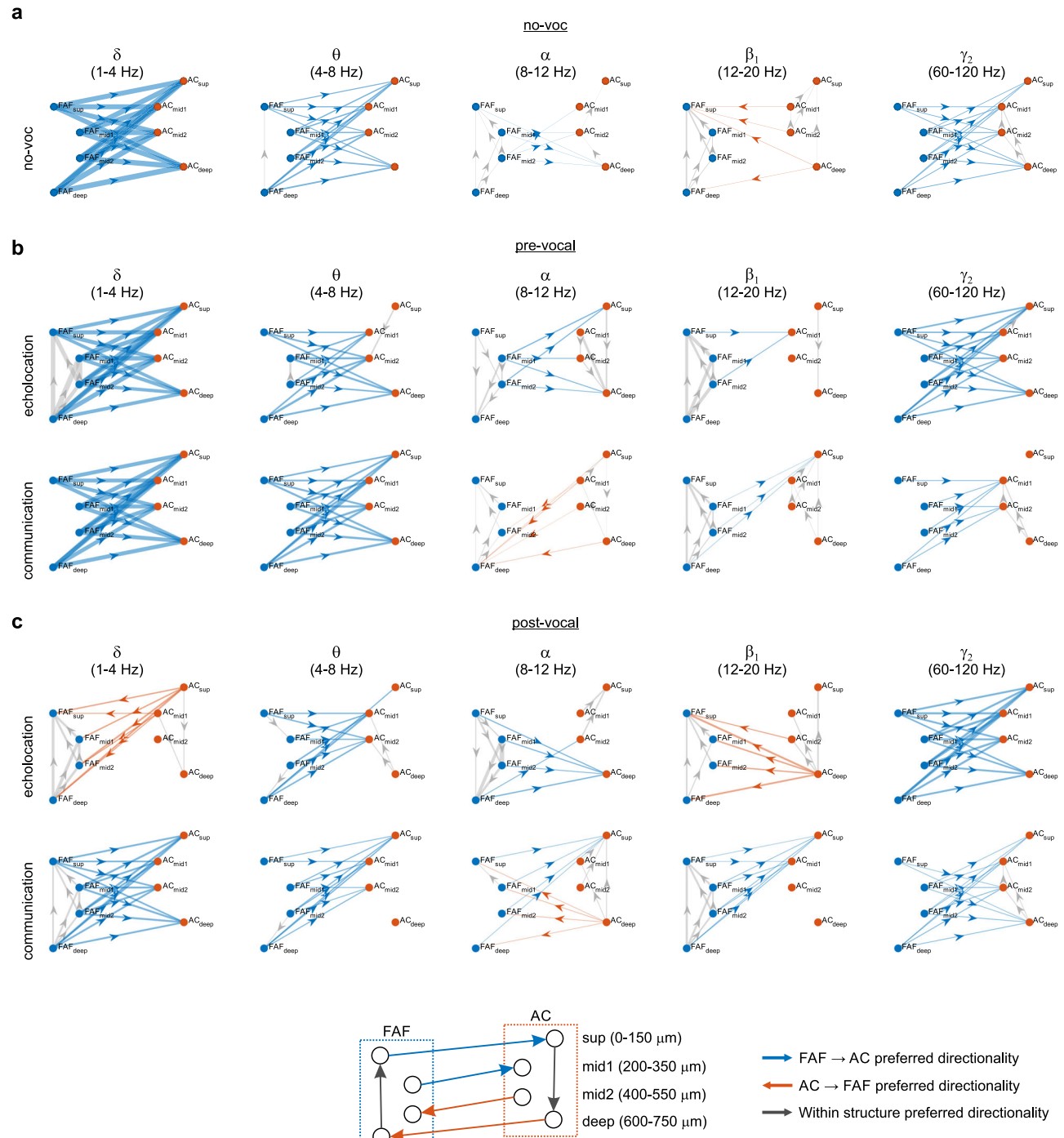

**Fig. 2 Directed connectivity patterns in the FAF-AC network. a** Graph visualization of directed connectivity between FAF and AC during no-voc periods. Channels in frontal and auditory cortices were combined into four categories: top (0–150 µm), mid1 (200–350 µm), mid2 (400–550 µm), and bottom (600–750 µm). Graph edges are weighted according to the strength of the preferred directionality (FAF→AC in blue; AC→FAF in orange; within-area directionality in gray). Edges are only shown if there was significant preferred directionality according to a threshold defined by bootstrapping. **b** Similar to **a**, but directed connectivity was calculated in the pre-vocal echolocation and communication conditions. **c** Same as **b**, with connectivity patterns obtained for post-vocal echolocation and communication conditions.

Remarkably, AC → FAF preferred directionality of information flow was significantly stronger during post-vocal periods in δ and β₁ frequency bands (Fig. 3a, top). Within FAF, differences in preferred information flow occurred in frequency bands δ, α, and β₁. Within AC, differences in dPTE occurred mostly in α and β₁ bands (Fig. 3a, top). Information flow was strongest in the deep-to-superficial direction during post-vocal periods, and in

superficial-to-deep direction during pre-vocal periods. In the case of communication call production (Fig. 3b, top), differences in dPTE occurred only in the δ and θ bands. Values were significantly higher (with large effect sizes) in the FAF → AC direction during pre-vocal periods.

Changes in directional information transfer in the FAF-AC network were quantified by calculating the net information

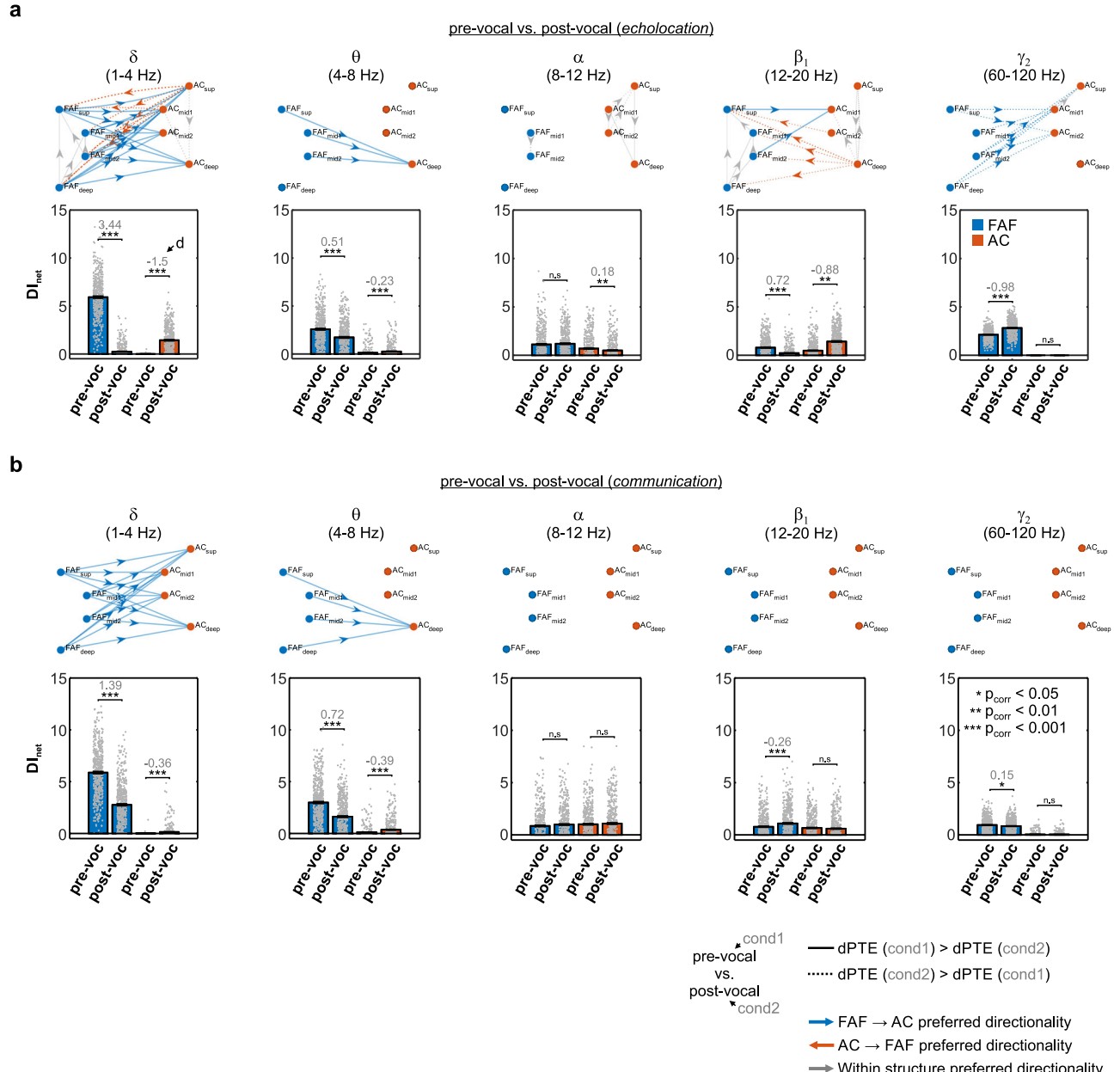

**Fig. 3 Pre-vocal and post-vocal directionality differences in the FAF-AC network. a** (*Top*) Graphs illustrating the differences between pre-vocal and post-vocal directionality, across frequency bands and during the production of echolocation calls. Edges were shown if three conditions were met: (i) the differences were significant (FDR-corrected Wilcoxon signed-rank tests, $p_{corr} < 0.05$), (ii) the effect size was large ($|d| > 0.8$), and (iii) edges were already significantly directional (see edges in Fig. 2). Edge thickness is weighted according to the effect size of the comparison. Continuous lines indicate pre-vocal dPTEs (first condition) higher than post-vocal dPTEs (second condition). Dashed lines indicate the opposite. (*Bottom*) Net information outflow ($DI_{net}$) from FAF (blue bars) and AC (orange bars), in the two conditions considered (pre-vocal vs. post-vocal). Significant differences across conditions are marked with stars (two-sided, FDR-corrected Wilcoxon signed-rank tests; *$p_{corr} < 0.05$, **$p_{corr} < 0.01$, ***$p_{corr} < 0.001$, n.s.: not significant; $n = 500$ repetitions). Gray numbers in the panels indicate effect sizes (*d*; not shown for non-significant differences). Values were considered independently of whether there was previous significant directionality in any of the two conditions. Data shown as mean ± sem. **b** Same as in **a**, but illustrating comparisons of directionality between pre-vocal vs. post-vocal conditions related to the vocalization of communication calls. Source data are provided as a Source Data file.

outflow ($DI_{net}$) from each area. The $DI_{net}$ represents the sum of DI values obtained from outgoing connections per region (e.g., all edges in FAF related to FAF → AC connections, representing the net strength of preferential information flow in the fronto-auditory direction). $DI_{net}$ values were used to statistically compare pre- and post-vocal periods in terms of information transfer from one cortical area to another. Considering this metric, significant differences (FDR-corrected Wilcoxon singed-rank tests, $p_{corr} < 0.05$) with large effect sizes ($|d| > 0.8$) occurred mostly for low

and intermediate frequency bands (i.e., δ and $β_1$) of the LFP. Specifically, for the pre-vocal vs. post-vocal echolocation condition (Fig. 3a, bottom), the information outflow from FAF was significantly higher in the δ band during pre-vocal periods related to echolocation call production ($p_{corr} = 1.63 \times 10^{-82}$, $d = 3.44$). Notably, the net information outflow from AC was significantly higher when considering post-vocal periods than pre-vocal ones ($p_{corr} = 3.27 \times 10^{-64}$, $d = -1.5$). In the $β_1$ frequency range, there were no significant differences with large effect sizes between pre-

vocal and post-vocal net information outflow from the FAF. However, $DI_{net}$ values from AC were significantly higher during post-vocal periods ($p_{corr} = 3.71 \times 10^{-36}$, $d = -0.88$). Pre-vocal vs. post-vocal comparisons of $DI_{net}$ values from FAF and AC related to communication calls revealed significant differences with large effect sizes only for δ frequencies in FAF (Fig. 3b, bottom; $DI_{net}$ higher for pre-vocal periods: $p_{corr} = 1.63 \times 10^{-82}$, $d = 1.39$).

The passive listening of echolocation-like or communication sounds did not account for the data above (Figs. S7–9), suggesting that mere feedback from the calls was not sufficient to explain echolocation-related, δ-band information flow reversal in the network. Likewise, the production of HF-communication sounds did not account for the patterns observed during echolocation (Fig. S10), indicating that the information transfer dynamics for echolocation are not fully explained by the frequency content of the vocalizations themselves. These results unveil dynamic changes of predominant connectivity patterns in the FAF-AC network from pre- to post-vocal periods, exhibiting frequency specificity and particularly associated with echolocation production.

**Electrical stimulation of the FAF enhances auditory cortical responses.** The data thus far indicate strong top-down modulation in the FAF-AC network, which can nevertheless be significantly altered when animals produce echolocation sounds. However, the dPTE analyses cannot rigorously establish whether FAF activity indeed modulates AC responses. To examine this question, we conducted perturbation experiments of the FAF to evaluate whether manipulations in this region affect auditory cortical responses to external stimuli.

The FAF was stimulated electrically with biphasic pulse trains (6 pulses/train; pulse interval: 500 ms) while simultaneously recording from the AC ($n = 20$ penetrations; Fig. 4a). Electrical stimulation of FAF did not produce detectable artefacts or LFP power changes in AC (Figs. 4c and S11), nor did it elicit vocalization production (Fig. S12), potentially due to weaker stimulation as compared to previous work reporting behavioral outputs[43,58]. Acoustic stimuli were presented after the train (either a distress -a type of communication sound- or an echolocation call; see Fig. S7b and Fig. 4b) at different latencies. Response strengths in AC to sounds after FAF electrical stimulation ("Estim" condition) were compared to response strengths related to the same sounds, but presented without prior electrical stimulation ("no-Estim" condition). Representative responses to distress and echolocation calls for both conditions and for a latency of 135 ms (in the Estim case) are depicted in Fig. 4d.

We observed differences between the ERP energy measured in Estim and no-Estim conditions (red and blue, respectively), more evidently when considering AC responses to echolocation stimuli (Fig. 4d). These differences occurred consistently at a population level. Figure 4e depicts response strengths for all AC depths across recordings, related to a distress syllable presented with a latency of 135 ms (in the Estim case; red traces). Response strengths from the no-Estim case are shown for one example iteration out of 500 conducted for comparisons (see "Methods"). A trend was present, wherein responses in the Estim condition were stronger than those in the no-Estim condition, although without statistically significant differences (Fig. 4f; FDR-corrected Wilcoxon signed-rank tests, significance if $p_{corr} < 0.05$; graphs split for clarity). Overall, when acoustic stimulation was done with a distress syllable, significant differences between Estim and no-Estim occurred in ~40 % of out 500 iterations in total. Such differences were concentrated mostly in middle-to-deep layers (depths > 300 μm) with small to medium values of Cliff's delta (ref.[59]; Fig. 4i, j, *left*).

In terms of AC responses to echolocation sounds (Fig. 4g), differences between Estim and no-Estim conditions appeared most prominent in superficial-to-middle layers (depths 50–350 μm). Responses were significantly stronger in the Estim condition than in the no-Estim condition at depths of 50–300 μm (Fig. 4h; $p_{corr} < 0.05$; graphs split for clarity), in particular for latencies of 135 ms. Significant differences between Estim and no-Estim conditions were very reliable, observed in up to 90.6% of the iterations for a latency of 135 ms and a cortical depth of 150 μm (Fig. 4i, *right*). At depths ranging 50–350 μm, for the same latency, reliability was larger than 70%, with medium effect sizes (Fig. 4j, *right*). These data indicate that electrical stimulation in FAF enhances AC response strength, with particularly high reliability when the animals listen to echolocation sounds.

As suggested by the data depicted in Fig. 4, changes in response strength between Estim and no-Estim conditions depended both on sound onset latency and AC depth (N-way ANOVA tests; for latency, channel and channel*latency: $p < 10^{-6}$; detailed statistics in Tables S1, S2). Figure 5 illustrates that effects of latency on response increase (i.e., Estim related vs. no-Estim related response strengths, expressed as a percentage) were most evident when considering responses to echolocation sounds (Fig. 5a). Notably, response increase was largest when the echolocation pulse was presented at a latency of 135 ms (see also Fig. 4i). Response strength increase depended on sound latency with a certain degree of periodicity (e.g., compare sound latencies of 135, 385, 635, and 885 ms with others), with approximately double the frequency of electrical stimulation (4 Hz). Thus, electrical stimulation of the FAF increased responses in AC with a long-lasting effect (hundreds of milliseconds) exhibiting a particular temporal pattern, suggesting that response enhancement is potentially mediated by local circuits in auditory cortex with intrinsic properties.

## Discussion

In this study, we addressed the dynamics of information exchange between the frontal and auditory cortices of vocalizing bats (Fig. 6). Consistent with previous reports[4,14,16], we show that neural activity in the frontal cortex predicts vocal outputs. Taken together, the data from this and previous work suggest that oscillations in frontal regions may be instrumental for vocal production. From our perspective, the above is further supported by call-type specific, pre-vocal LFP spectral dynamics and information transfer patterns in the FAF-AC network. The relationship between oscillations and vocal production remains, nevertheless, correlational: our results do not allow to rigorously assert a causal role of LFPs for the initiation or planning of vocalizations.

Neural activity in the AC also relates to vocalization[27], but the involvement of auditory cortical oscillations in vocal production is still to be fully understood. Our results indicate that pre-vocal LFPs in AC, as previously reported with single-unit spiking[27,29,30], relate to vocal initiation. We show, for the first time to our knowledge, that pre-vocal oscillatory patterns in AC are call-type specific and, remarkably, complementary to those observed in frontal cortex in frequency and effect (Fig. 1). These patterns may be explained by our current understanding of the roles of AC for vocal production. Neuronal activity in the AC is predominantly suppressed during vocalization, with inhibition occurring hundreds of milliseconds prior to call onset[8,27,28,60]. Vocalization-related inhibition is mediated by motor control regions, which send a copy of the motor command to the AC as "corollary discharge" or "efferent copy" signals[33,61]. These signals, respectively, have either a general suppressive effect, or carry specific information about the produced sound which potentially

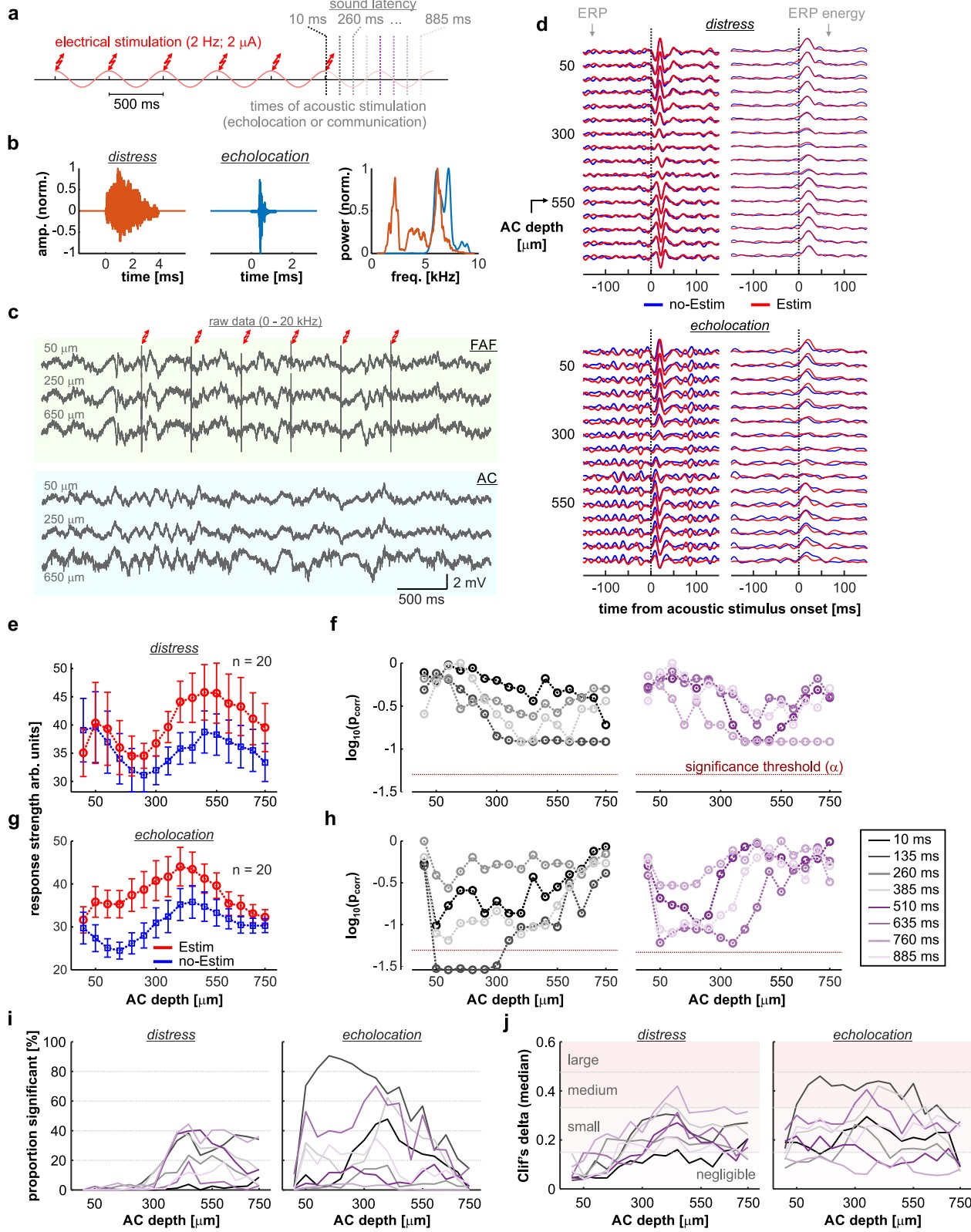

facilitates feedback processing[32]. Thus, pre-vocal, call-type unspecific power changes in low frequencies could reflect general inhibitory mechanisms in AC consistent with corollary discharges from higher order structures. Directed connectivity analyses support the notion of top-down (FAF → AC) control of pre-vocal low-frequency activity (Fig. 2). In contrast, pre-vocal β-

band LFPs might constitute oscillatory correlates of efference copies, given the observed call-type specificity. Because FAF → AC causal influences did not equally extend to the β frequencies, pre-vocal β activity in AC might be influenced instead by specialized regions such as the premotor cortex, providing a more specific copy of the motor commands required for vocalization.

**Fig. 4 Electrical stimulation of the FAF increases response strength in AC. a** Schematic representation of the paradigm for electrical and acoustic stimulation. The timestamps for acoustic stimulation (colored according to latency; see also panels **e**–**g**) represent the latency of sound onset relative to the end of electrical stimulation train. **b** Oscillograms of the natural distress syllable and echolocation pulse used for acoustic stimulation. On the right, the normalized power spectra of both calls are shown (orange, distress; blue, echolocation). **c** Broadband (0–20 kHz), raw data recorded simultaneously from FAF and AC (at representative depths of 50, 250, and 600 μm) illustrating a single trial of electrical stimulation. Note that no electrical artefacts are visible in AC. **d** Auditory cortical LFPs (left column), and time-course of their energy (right), in response to either the distress syllable (top) or the echolocation pulse (bottom). Responses corresponding to the no-Estim condition shown in blue; responses related to the Estim condition, in red. **e** Strength of auditory cortical ERPs in response to the distress syllable, across all recorded columns ($n = 20$ independent penetrations) and depths. In blue, responses associated to the no-Estim condition; in red, those associated to the Estim condition (data as mean ± s.e.m). **f** Corrected $p$-values obtained after statistical comparisons between response strengths related to Estim and no-Estim conditions, across all channels and latencies (paired, two-sided FDR-corrected Wilcoxon signed rank tests, alpha = 0.05). **g**, **h** Same as in **e**, **f**, but dealing with responses to the echolocation pulse. **i** Proportion of iterations (out of $n = 500$ total iterations) in which responses associated to the Estim condition were significantly larger than those associated to the no-Estim condition (same test as above). Data are presented across all channels and latencies analyzed, for responses to the distress and echolocation sounds. **j** Median effect size (Cliff's delta) for the same comparisons summarized in (**i**). Source data are provided as a Source Data file.

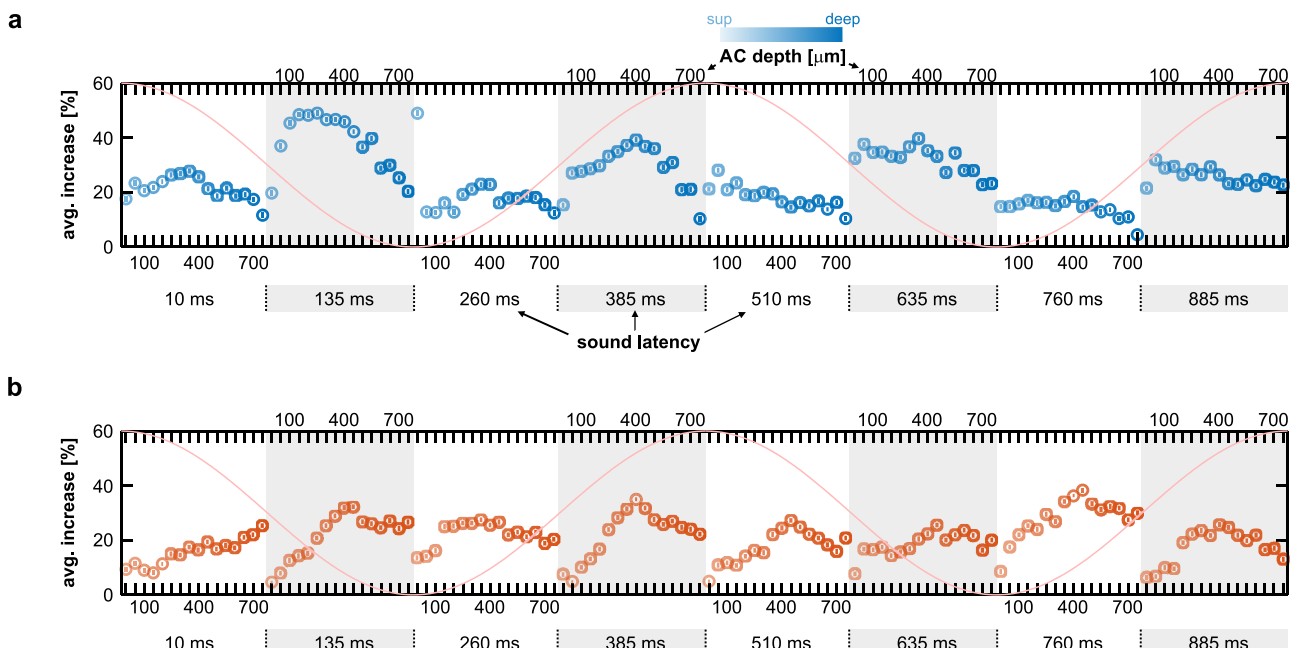

**Fig. 5 Relative AC response increase after FAF electrical stimulation depends on sound-onset latency. a** Average relative increase in response strength to the echolocation pulses during the Estim conditions relative to the no-Estim condition (expressed as percentage increase). Each circle corresponds to a channel in AC; note that all channels and sound latencies are shown (lighter colors correspond to electrodes located at lower depths). The center of the circle indicates the mean of the group ($n = 500$ iterations), while vertical bars indicate the 95% confidence intervals. In the figure, values are significantly different if their confidence intervals do not overlap (*multcompare* function in Matlab; alpha = 0.05, multiple comparisons corrected using Tukey–Kramer critical values). **b** Data related to responses to the distress syllable, shown as in (**a**). The purple cosine curve tracks the corresponding phase of the 2 Hz electrical stimulation (although note that electrical stimulation had stopped before sound onset; see also Fig. 4a). Data shown in this figure are the same used for Fig. 4.

Channels for motor-auditory communication (see refs. [62,63]) could in fact operate over β frequencies[53,64,65].

Differences in spectral patters cannot be solely explained by the distinct frequency content of echolocation and communication calls (Fig. S3). However, considering that orofacial movement in primates[15,66] and vocalization-specific movements in bats[50] are associated to neural activity in frontal areas, distinct pre-vocal motor related activity for echolocation or communication calling is a plausible explanation for our results. Microstimulation of *C. perpsicillata*'s FAF can result in motor effects such as pinna and nose-leaf movements, as well as vocalizations (including echolocation-like calls; ref. [50]). These movements also occur naturally before spontaneous vocalization[50], suggesting that the FAF may be involved in the motor aspect of vocal production. Nevertheless, vocalization-specific neural populations in primates coexist with those related to orofacial movements[67]. Therefore,

the vocal-motor explanation does not necessarily entail that the FAF fails to participate in other forms of vocal preparation beyond the orchestration of motor programs.

In terms of a cortical network for vocalization, the FAF and AC are engaged in rich information transfer dynamics with functional relationships to vocalization. Moreover, interactions extend to periods of vocal quiescence, when information flows top-down (FAF → AC) in low (δ-α) and high (γ₂) frequencies. Low-frequency top-down influences from higher-order structures (like the FAF) modulate neuronal activity in sensory cortices according to cognitive variables such as attention, also during spontaneous activity[56,68,69]. However, whether and how attentional processes exploit the nature of neural connections in the FAF-AC circuit remains thus far unknown. Our data resonate with the hypothesis of top-down modulation of oscillatory activity in AC, and suggest a strict control of higher-order structures over sensory areas

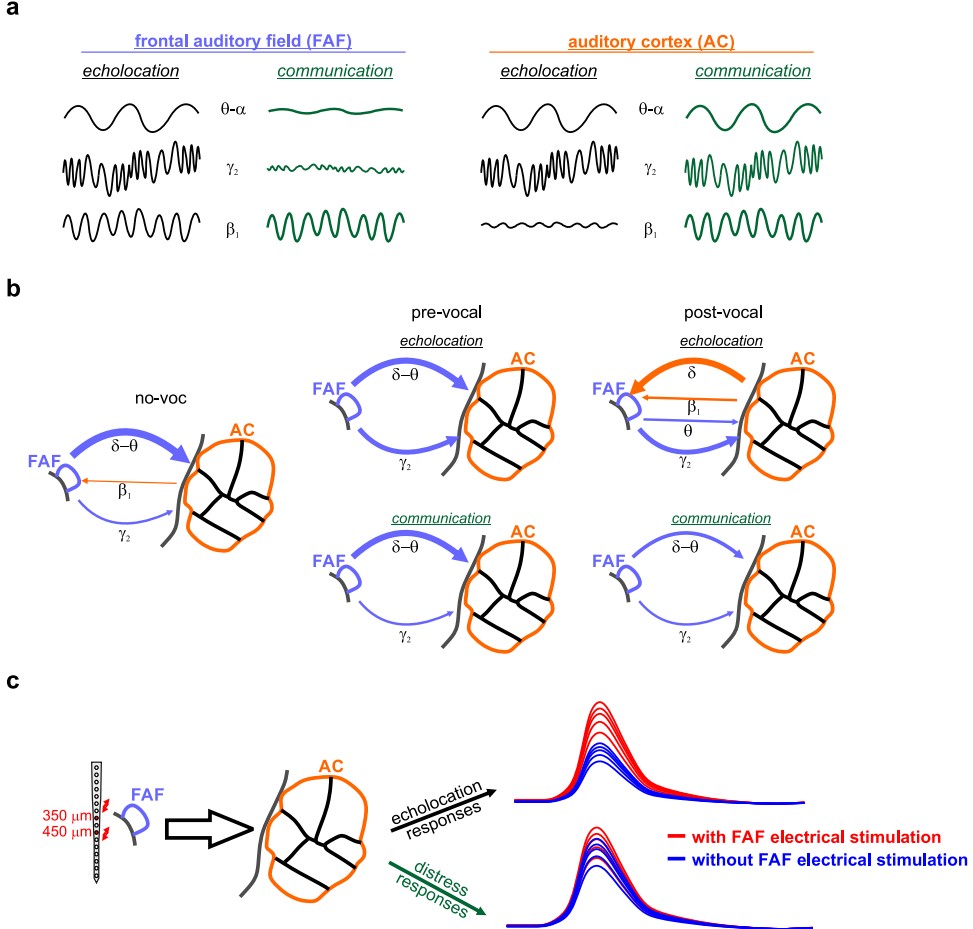

**Fig. 6 The FAF-AC network during vocal production. a** Oscillations in frontal and auditory cortices provide a neural correlate of vocal production, allowing the prediction of ensuing call type. Prediction is possible in complementary frequency bands in each region, and with opposite effects. **b** Schematic representation of causal interactions (within a TE framework) in the FAF-AC network. Strong top-down control, mostly in δ and γ frequencies, occurs during spontaneous activity (no-voc) and prior to vocal utterance. In the δ-band, information flows top-down in the circuit (FAF → AC) during pre-vocal periods, but changes to bottom-up (AC → FAF) information transfer during post-vocal periods. The directionality patterns and the strength of preferential causal interactions depend on the type of call produced, and on the timing relative to vocal onset. **c** Electrical stimulation results provide strong support to the notion that FAF alters the manner in which AC processes acoustic information, preferentially when animals listen to echolocation sounds. In comparison to the listening of distress sounds, auditory cortical responses to echolocation sounds were more reliably enhanced after electrically stimulating the frontal cortex.

reflected in concurrent LFP activity across regions. Such strong top-down control is supported by the fact that FAF microstimulation enhances AC responses to sounds (Fig. 4).

Vocalization-specific changes in power may affect causality estimations, e.g., by creating confounding differences between the vocal conditions studied. However, the dPTE is a causality index that shows robustness to the influence of power, noise, and other variables. In our dataset, the pre-vocal δ-band power increase within each region when animals produced echolocation (call-type specific in FAF, unspecific in AC) was nonetheless accompanied by a decrease of interareal dPTE values. In addition, a δ-band power increase of communication pre-vocal LFPs relative to baseline (Fig. 1) did not result in significant differences of dPTE values during pre-vocal and spontaneous periods. Thus, changes in causality did not necessarily follow changes in power, as reported in previous work[56].

Based on dPTE values associated with spontaneous and pre-vocal activities (Figs. 2, S5), it appears that as animals prepare the production of an echolocation call, the FAF gradually relinquishes control over the AC in the low-frequency (δ) channel. That is, the top-down control wanes during

echolocation pre-vocal periods in δ LFPs. The weakening of preferred top-down information transfer could be taken as a preamble of emerging bottom-up information flow in δ frequencies after an echolocation call is emitted (Fig. 3). This fails to happen in the communication case. Echolocation, the predominant strategy for navigation, is essential for bats. After vocalizing an echolocation pulse, the bat auditory system must be ready to process incoming echoes and use them to construct a representation of surrounding objects[70], potentially involving higher order structures. The observed switch from top-down to bottom-up processing when animals find themselves in echolocation mode could represent the readiness of the bat's auditory machinery for the aforementioned task. Concretely, our data suggest that the former may occur over a continuum encompassing a gradual release of the AC from top-down influences (the FAF), which opens the way for auditory-frontal information transfer supporting the processing and integration of incoming echoes. A reversal in information transfer is also visible (albeit weaker and in a different LFP frequency band) during the production of high frequency communication sounds (Fig. S10). This result could hint towards smooth transition in the way the

FAF-AC network operates, which finds its two extremes in echolocation and low-frequency communication call production.

In all, processing feedback information directly related to navigation appears to have a larger weight in the bottom-up processing of acoustic cues resulting from a self-generated sound. Echolocation pulses are produced to generate echoes that must be listened to. Communication calls are often targeted to an audience as means of transmitting internal behavioral information (e.g., distress, aggressive mood, etc.), not aimed at the emitter itself. For the emitter, in such scenario, feedback processing mostly contributes to the adjustment of vocal parameters such as loudness or pitch[7,71,72]. Since in this study animals vocalized without an audience (i.e., they were isolated in the recording chamber), further research could elucidate whether the presence of conspecifics increases bottom-up information transfer when vocalizing communication calls, as animals could expect a conspecific's response.

The reversal of information flow reported in the δ-band when animals echolocate cannot be solely attributed to passively hearing feedback from their own utterances (Figs. S7–10): active vocalization seems to be necessary to trigger bottom-up information transfer in the FAF-AC circuit. The current data, together with the fact that passive listening fails to significantly alter low-frequency coherence in the FAF-AC network[17], indicate that passive listening alone is not sufficient to significantly alter the dynamics of communication between FAF and AC. Likewise, information flow dynamics associated to echolocation calls could not be attributed solely to their high frequency content, since qualitatively similar variations between echolocation and communication calls were observed when considering only HF-communication utterances, particularly for δ-LFPs (Fig. S10). The act of emitting echolocation pulses therefore triggers unique patterns of information flow reversal in the fronto-auditory network of *C. perspicillata*. Our data suggest that the functional connectivity in this circuit is shaped by the behavioral intent of the vocalization.

The transfer entropy analyses discussed above indicate that the frontal cortex exerts top-down modulation over its auditory cortical counterpart, particularly during spontaneous activity (Fig. 2). Consistent with the top-down modulation perspective, electrical perturbation of the FAF enhanced the strength of responses in AC, echoing known effects of frontal stimulation in other mammals[21]. The precise mechanisms by which FAF microstimulation affects AC response remains to be clarified. It has been suggested that top-down interactions modulate sensory areas by means of inhibitory effects potentially capitalizing on low-frequency interactions (see refs. [73–75]). Given the low-frequency (directed) functional coupling in the FAF-AC network, we hypothesize that top-down projections from the FAF reach the AC and modulate cortical gain by affecting the temporal dynamics of local inhibitory networks (e.g., ref. [76]). These networks might possess intrinsic, low-frequency rhythms, potentially reflected in the quasi-periodic pattern seen in the relationship between sound-onset latency and AC response increase (Fig. 5a). Nevertheless, such observations remain speculative and require empirical validation. Altogether, considering that FAF manipulation most reliably altered responses to echolocation stimuli (Fig. 4), and that the production of echolocation pulses reverses information flow in the fronto-auditory circuit, our data suggest that dynamic interactions in the bat's FAF-AC network are particularly associated with auditory processing for echolocation behavior.

## Methods

**Animal preparation and surgical procedures**. The study was conducted on five awake *Carollia perspicillata* bats (one female). Experimental procedures were in compliance with European regulations for animal experimentation and were approved by the Regierungspräsidium Darmstad (experimental permits #FU-1126 and FR-2007). Bats were obtained from a colony at the Goethe University, Frankfurt. Animals used for experiments were kept isolated from the main colony.

Prior to surgical procedures, bats were anaesthetized with a mixture of ketamine (10 mg*kg$^{-1}$, Ketavet, Pfizer) and xylazine (38 mg*kg$^{-1}$, Rompun, Bayer). For surgery and for any subsequent handling of the wounds, a local anesthetic (ropivacaine hydrochloride, 2 mg/ml, Fresenius Kabi, Germany) was applied subcutaneously around the scalp area. A rostro-caudal midline incision was cut, after which muscle and skin tissues were carefully removed in order to expose the skull. A metal rod (ca. 1 cm length, 0.1 cm diameter) was attached to the bone to guarantee head fixation during electrophysiological recordings. The FAF and AC were located by means of well-described landmarks, including the sulcus anterior and prominent blood vessel patterns (see refs. [17,50,77]). The cortical surface in these regions was exposed by cutting small holes (ca. 1 mm$^2$) with the aid of a scalpel blade on the first day of recordings. In the AC, recordings were made mostly in the high frequency fields[17,50,77])

After surgery, animals were given no less than two days of rest before the onset of experiments. No experiments on a single animal lasted longer than 4 h per day. Water was given to the bats every 1–1.5 h periods, and experiments were halted for the day if the animal showed any sign of discomfort (e.g., excessive movement). Bats were allowed to rest a full day between consecutive experimental sessions.

**Electrophysiological and acoustic recordings**. Electrophysiology was performed chronically in fully awake animals, inside a sound-proofed and electrically isolated chamber. Inside the chamber, bats were placed on a custom-made holder which was kept at a constant temperature of 30 °C by means of a heating blanket (Harvard, Homeothermic blanket control unit). Electrophysiological data were acquired from FAF and AC on the left hemisphere, using two 16-channel laminar electrodes (one per structure; Model A1x16, NeuroNexus, MI; 50 μm channel spacing, impedance: 0.5–3 MΩ per electrode). Probes were carefully inserted into the brain perpendicular to the cortical surface, and lowered with piezo manipulators (one per probe; PM-101, Science products GmbH, Hofheim, Germany) until the top channel was barely visible above the surface of the tissue. The placing and properties of the probes allowed us to record simultaneously at depths ranging from 0 to 750 μm, spanning all six cortical layers (see ref. [78]). Probes were connected to a micro-preamplifier (MPA 16, Multichannel Systems, MCS GmbH, Reutlingen, Germany), and acquisition was done with a single, 32-channel portable system with integrated digitization (sampling frequency, 20 kHz; precision, 16 bits) and amplification steps (Multi Channel Systems MCS GmbH, model ME32 System, Germany). Reference electrodes (silver wires) were used for each recording shank (i.e., in FAF or AC) at a different area of the brain (for FAF: a non-auditory lateral, ipsilateral region; for the AC: a non-auditory occipital, ipsilateral region). Wires were carefully placed to rest between the skull and the dura matter. For each laminar electrode, reference and ground were short-circuited; the ground was however common in the acquisition system (i.e., the ME32). Acquisition was online-monitored and stored in a computer using the MC_Rack_Software (Multi Channel Systems MCS GmbH, Reutlingen, Germany; version 4.6.2).

Vocal outputs were recorded by means of a microphone (CMPA microphone, Avisoft Bioacustics, Glienicke, Germany) located 10 cm in front of the animal. Recordings were performed with a sampling rate of 250 kHz and a precision of 16 bits, using the Avisoft Recorder software (Avisoft Bioacoustics, Glienicke, Germany; versions 4.2.8 and 4.3.01). Vocalizations were amplified (gain = 0.5, Avisoft UltraSoundGate 116Hm mobile recording interface system, Glienicke, Germany) and then stored in the same PC used for electrophysiology. Electrophysiological and acoustic data were aligned using two triggers, an acoustic one (5 kHz tone, 10 ms long) presented with a speaker located inside of the chamber (NeoCD 1.0 Ribbon Tweeter; Fountek Electronics), and a TTL pulse sent to the recording system for electrophysiology (see above). Note that the onsets of the tones were in synchrony with the TTL pulses registered by the acquisition system for electrophysiology.

**Acoustic stimulation**. Two acoustic stimuli were used to evaluate transfer entropy patterns during passive listening. One of them, the high-frequency frequency modulated sound (HF-FM; 2 ms long; downward frequency sweep from 80 to 50 kHz), mimicked the spectrotemporal structure of echolocation pulses; the other, consisted of a distress syllable (distress, 3.8 ms long) typical of *C. perspicillata*'s vocal repertoire. The latter stimulus was embedded in a sequence in which the syllable was presented every 500 ms for 2 seconds (2 Hz rate); other sequences with faster rates were also presented to the animal, but were not considered for this study. Only the first syllable of the 2 Hz sequence was used for analyses. Stimuli for determining frequency tuning consisted of short (10 ms) pure tones at various frequencies (5–90 kHz, in steps of 5 kHz) and levels (15–75 dB SPL, steps of 15 dB). Since the HF-FM and the distress sounds were presented at 70 dB SPL (rms), we focused on the frequency tuning curved obtained with pure tone stimuli presented at 75 dB SPL.

The setup for stimulation has been described in previous studies (see ref. [17]). In short, sounds were digital-to-analog converted using a sound card (M2Tech Hi-face DAC, 384 kHz, 32 bit), amplified (Rotel power amplifier, model RB-1050), and presented through a speaker (description above) inside of the chamber. The

speaker was located 12 cm away from the bat's right ear, contralateral to the cerebral hemisphere on which electrophysiological recordings were made. Prior to stimulation, sounds were downsampled to 192 kHz and low-pass filtered (80 kHz cut-off). Sound presentation was controlled with custom written Matlab softwares (version 8.6.0.267246 (R2015b), MathWorks, Natick, MA) from the recording computer.

**Classification of vocal outputs**. Two sessions of concurrent acoustic recordings (~10 min long) were made per paired penetrations in FAF and AC. Vocalizations were automatically detected based on the acoustic envelope of the recordings. The envelope was z-score normalized to a period of no vocalization (no less than 10 s long), which was manually selected, per file, after visual inspection. If a threshold of 5 standard deviations was crossed, a vocalization occurrence was marked and its start and end times were saved. Given the stereotyped spectral properties of *C. perspicillata*'s echolocation calls, a preliminary classification between echolocation and communication utterances was done based on each call's peak frequency (a peak frequency > 50 kHz suggested an echolocation vocalization, whereas a peak frequency below 50 kHz suggested a communication call). In addition, vocalizations were labeled as candidates for subsequent analyses if there was a time of silence no shorter than 500 ms prior to call production to ensure no acoustic contamination on the pre-vocal period that could affect LFP measurements in FAF or AC. Finally, echolocation and communication candidate vocalizations were individually and thoroughly examined via visual inspection to validate their classification (echolocation or communication), the absence of acoustic contamination in the 500 ms prior to vocal onset, and the correctness of their start and end time stamps. According to the above, and out of a total of 12,494 detected vocalizations, 138 echolocation and 734 communication calls were then used in further analyses.

High-frequency communication calls (HF-communication) were selected according the frequency component of the vocalizations. Specifically, an HF-communication call was an utterance with more than 50% of its power in the 50–100 kHz range. HF-communication calls represented 21.12% of the communication calls used (155/734).

**Extraction of LFP signals and power analyses**. Data analyses were performed using custom-written scripts in MatLab (versions 9.5.0.1298439 (R2018b), and 9.10.0.1684407 (R2021a)), Python (version 2.6 or 3.6), and R (RStudio version 1.3.1073). For extracting LFPs, the raw data were band-pass filtered (zero-phase) between 0.1 and 300 Hz (4th order Butterworth filter; *filtfilt* function, MatLab), after which the signals were downsampled to 1 kHz.

All LFP spectral analyses were done using the Chronux toolbox[79]. Pre-vocal power was calculated with LFP segments spanning -500-0 ms relative to vocal onset, using a TW of 2, and 3 tapers. No-vocalization baseline periods (no-voc) with a length of 500 ms were pseudo-randomly selected and their power spectra calculated in order to obtain baseline power values for spontaneous activity. The total number of no-voc periods matched the total number of vocalizations ($n = 872$), in a way that the number of selected no-voc periods per recording file matched the number of vocalizations found in that particular file. The power of individual frequency bands (i.e., δ, 1–4 Hz; θ, 4–8 Hz; α, 8–12 Hz; $\beta_1$, 12–20 Hz; $\beta_2$, 20–30 Hz; $\gamma_1$, 30–60 Hz; $\gamma_2$, 60–120 Hz; $\gamma_3$, 120–200 Hz) was calculated by integration of the power spectral density accordingly for each case. Finally, the increase of pre-vocal power relative to the baseline periods was calculated as follows (per frequency band, on a call-by-call basis):

$$\text{Relative powerchange} = \frac{\text{BP}_{\text{pre-voc}} - \text{BP}_{\text{no-voc}}}{\text{BP}_{\text{no-voc}}} * 100, \quad (1)$$

where $\text{BP}_{\text{pre-voc}}$ is the pre-vocal power (in the case of either an echolocation or communication vocalization) of the given frequency band and a trial (i.e., a specific call), and $\text{BP}_{\text{no-voc}}$ is the baseline no-voc power associated to the same frequency band and trial.

**Generalized linear model for vocal output prediction**. To determine whether pre-vocal power change relative to baseline was able to predict the type of ensuing vocal output, we used a GLM with a logistic link function (i.e., logistic regression). The model analysis was done in Rstudio with the *lme4* package. In brief, logistic regression was used to predict the probability of a binary outcome (0 or 1; communication or echolocation, respectively) based on the pre-vocal power change as the predictor variable. The probabilities are mapped by the inverse logit function (sigmoid):

$$\sigma(x) = \frac{1}{1 + \exp(-x)}, \quad (2)$$

which restricts the model predictions to the interval [0, 1]. Because of these properties, a logistic regression with GLMs is well suited to compare data (and thus, evaluate predictions of ensuing vocal-output) on a single-trial basis[80].

To estimate the effect size of the fitted models, we used the marginal coefficient of determination ($R^2m$) with the *MuMIn* package. The $R^2m$ coefficient quantifies the variance in the dependent variable (echolocation vs. communication vocalization) explained by the predictor variable (i.e., the relative pre-vocal power change). This value is dimensionless and independent of sample size[80,81], which

makes it ideal to compare effect sizes of different models (e.g., across channels and frequency bands, as in Fig. 1g). Effect sizes were considered small when $R^2m < 0.1$, medium when $0.1 \leq R^2m < 0.4$, and large when $R^2m \geq 0.4$[80].

**Directionality analyses**. Directional connectivity in the FAF-AC network was quantified with the directed phase transfer entropy (dPTE; ref. [56]), based on the phase transfer entropy (PTE) metric[82]. PTE is a data-driven, non-parametric DI that relates closely to transfer entropy (TE; ref. [83]), but is based on the phase time-series of the signals under consideration (here, FAF and AC field potentials). PTE is sensitive to information flow present in broad- and narrowband signals, and is in a large degree robust to the effects of, for example, noise, linear mixing, and sample size[82,84].

In terms of TE, a signal X causally influences signal Y (both of them can be considered as phase times series), if the uncertainty about the future of Y can be reduced from knowing both the past of signal X and signal Y, as compared to knowing the past of signal Y alone. Formally, the above can be expressed as follows:

$$TE_{xy} = \sum p(Y_{t+\delta}, Y_t, X_t) \log\left(\frac{p(Y_{t+\delta}|Y_t, X_t)}{p(Y_{t+\delta}|Y_t)}\right), \quad (3)$$

where δ represents the delay of the information transfer interaction, and $TE_{xy}$ is the transfer entropy between signals X and Y. The estimation of the probabilities for TE quantification requires large computational times and the tuning of various parameters[56]. PTE, on the other hand, converts the time series into a sequence of symbols (binned-phase time series, see below), and is able to estimate TE on the phase series reducing significantly both processing times and the necessity for parameter fitting[82].

Phase time series were obtained after filtering the LFP signals in a specific frequency band (e.g., θ, 4–8 Hz) and Hilbert transforming the filtered data. To avoid edge artefacts, the full ~10 minutes recordings were filtered and Hilbert transformed before chunking segments related to individual trials (i.e., pre-voc: -500-0 ms relative to call onset, post-voc: 0–250 ms relative to call onset, or no-voc baseline periods). According to the condition under consideration (echolocation/communication and pre-voc/post-voc, or baseline periods), we selected 50 trials pseudo-randomly and then concatenated them before quantifying directional connectivity. This process was repeated 500 times and the distribution of dPTE values obtained from each repetition used for further analyses. The former resulted in a distribution of 500 dPTE connectivity matrices; the median value across these was used for constructing connectivity graphs (see below).

Given the phase of the LFP signals, the PTE was calculated according to eq. 3. However, probabilities, in this case, were estimated by constructing histograms of binned phases[82] instead of using the full, continuous time series. Following[85], the number of bins in the histograms was set to:

$$3.49 * \mu\big(\sigma(\phi)\big) * N_s^{-\frac{1}{3}}, \quad (4)$$

where μ and σ represent the mean and standard deviation, respectively, φ represents the phase time series, and $N_s$ denotes the number of samples.

The prediction delay d was set to $(N_s \times N_{\text{ch}})/N_{+-}$[56], where $N_s$ and $N_{\text{ch}}$ are the number of samples and channels ($N_{\text{ch}} = 32$), respectively. The value of $N_{+-}$ corresponds to the number of times the LFP phase changes sign across all channels and times.

The dPTE was calculated from the PTE as follows[56]:

$$\text{dPTE}xy = \frac{\text{PTE}xy}{\text{PTE}xy + \text{PTE}yx} \quad (5)$$

With values ranging between 0 and 1, dPTEs > 0.5 indicate information flow preferentially in the $X \rightarrow Y$ direction, dPTE values below 0.5 indicate preferential information flow in the opposite direction, and dPTE = 0.5 indicates no preferred direction of information flow. In other words, dPTE is a metric of preferred directionality between two given signals. Note that the dPTE analysis among a set of electrodes yields a directed connectivity matrix that can be considered as an adjacency matrix of a directed graph (see below). All PTE and dPTE calculations were done with the Brainstorm toolbox in MatLab[86].

**Connectivity graphs**. A graph-theoretic examination of the connectivity patterns was made by constructing directed graphs based on the results obtained from the dPTE analyses (i.e., the median across the 500 repetitions; see above). For simplicity, channels in the FAF and AC within a range of 150 μm were grouped as follows (in the FAF, as an example): $\text{FAF}_{\text{top}}$, channels 1–4 (0–150 μm); $\text{FAF}_{\text{mid1}}$, channels 5–8 (200–350 μm); $\text{FAF}_{\text{mid2}}$, channels 9–12 (400–550 μm); $\text{FAF}_{\text{bottom}}$, channels 13–16 (600–750 μm). A similar grouping was done for electrodes located in AC. These channel groups were considered as the nodes of a directed graph. A directed edge (u, v) between any two nodes then represents a preferential information flow from node u to node v. The weight of the edge was taken as the median dPTE for the channel groups corresponding to the nodes, according to the dPTE connectivity matrices. For instance, if the groups considered were $\text{FAF}_{\text{top}}$ and $\text{AC}_{\text{bottom}}$, then the weight between both nodes was the median of the obtained dPTE values calculated from channels 1–4 in FAF towards channels 13–16 in AC.

The weight of an edge was quantified as a DI:

$$DI = \frac{\text{median}(dPTE_{uv}) - 0.5}{0.5} * 100, \quad (6)$$

which expresses, in percentage points, the strength of the preference of information flow in a certain direction. Equation 6 is based on the fact that a dPTE of 0.5 corresponds to no preferred direction of information flow[56].

To statistically validate the directionality shown in the graphs we used a bootstrapping approach. Surrogate adjacency matrices were built for the same channel groups (top, mid1, mid2 and bottom), but electrodes were randomly assigned to each group, independently of their depths or cortical location. This randomization was done independently within each of the 500 dPTE matrices obtained from the main connectivity analysis. Then, an adjacency matrix was obtained from these surrogate data in the same way as described above (i.e., using the median across 500 randomized dPTE matrices). Such a procedure was repeated 10,000 times, yielding an equal number of surrogate graphs. An edge in the original graph was kept if the DI of that edge was at least 2.5 standard deviations higher than the mean of the surrogate distribution obtained for that edge (i.e., higher than the 99.38% of the surrogate observations). Edges that did not fulfill this criterion were labeled as non-significant and were therefore not considered for any subsequent analyses.

**Directionality analyses for passive listening conditions**. dPTE values and connectivity graphs for passive listening conditions were quantified using the same methodology described for the cases of active vocalization. Analyses based on responses to acoustic stimulation were made on a trial-by-trial basis. Trials were randomly selected 500 times across all penetrations, depending on whether responses to the HF-FM or the distress sound were considered. We ensured that the number of trials chosen for each penetration, in every randomization run, matched the number of vocalizations taken (in a HF-FM/echolocation or distress/communication scheme) from that particular penetration. With this, we aimed to avoid possible biases in the comparisons across passive listening and active vocalizations conditions.

**Electrical stimulation experiments**. The FAF was electrically stimulated by means of biphasic pulses lasting 410 μs (200 μs per phase, with 10 a μs gap between them) and with an amplitude of 2 μA. Electric pulses were delivered by inserting an A16 Neuronexus shank (same used for recordings) into the frontal cortex, using the channels at depths of 350 and 450 μm as stimulating electrodes. These channels were directly connected to the outputs of an A365 stimulus isolator (World Precision Instruments, Friedberg, Germany). Pulse amplitude was selected based on values used in the literature (e.g., refs. [87–89]), and after empirically establishing that electrical artefacts were undetectable in the AC online during recordings. Recordings in AC were conducted with a second A16 shank (as described above); in the AC no electrical stimulation was delivered.

Precisely, the electrical and acoustic stimulation protocol was as follows. Six biphasic electric pulses were delivered into FAF with inter-pulse intervals of 500 ms (2 Hz, within the δ-band range). After the electrical pulse train, acoustic stimuli were presented to the bats at given latencies relative to the time last electrical pulse (10, 135, 260, 385, 510, 635, 760, and 885 ms; "Estim" condition). Latencies were consistent with sampling four different phases per electrical stimulation cycle (period is considered as the inter-pulse interval), for a total of 2 cycles after the last electrical pulse delivered (see Fig. 4a). Acoustic stimuli consisted of a distress (same syllable used for the passive listening experiments) and an echolocation call (duration, 1.2 ms; spectrum shown in Fig. 4b), both presented at 50 dB SPL (rms). All possible combinations of acoustic stimulus type (i.e., distress or echolocation) and sound onset latency were pseudorandomly presented 25 times each. The interval between a trial block (i.e., electric pulse train and acoustic stimulus) was of 2.5 s. Sounds were also presented without any prior electrical stimulation ("no-Estim" condition), with a variable inter-stimulus interval between 4 and 16 s, a total of 50 times each. Acoustic stimuli were delivered using the same speaker setup described above, but for these experiments a different sound card was used (RME Fireface UC; 16 bit precision, 192 kHz; RME Audio, Haimhausen, Germany).

Comparisons of auditory cortical response strength between Estim and no-Estim conditions were made by calculating the energy of the event-related potential (ERP). Specifically, the response strength was calculated as the area under the curve of the absolute value of the Hilbert-transform of the high-frequency component (25–80 Hz) of the ERP, for the first 150 ms after sound onset. We used the high frequency component of the ERP to avoid biases related with low-frequency prestimulus trends in the LFPs. Because the number of trials differed in the Estim and no-Estim conditions (i.e., 25 and 50, respectively), the response strength of the no-Estim condition was calculated using 25 randomly selected trials out of 50. The ERP is sensitive to the number of trials as it is a trial-average response; the number of trials must therefore be equalized. Only then we compared response strengths between Estim and no-Estim conditions, for each channel and latency relative to the last electrical pulse delivered. Statistical comparisons were made with FDR-corrected Wilcoxon signed-rank tests, with an alpha of 0.05. The effect sizes of these comparisons were calculated using the Cliff's delta metric (a non-parametric approach). Effect sizes are considered negligible when Cliff's delta < 0.147, small when 0.33 ≤ Cliff's delta < 0.33, medium when 0.33 ≤ Cliff's delta < 0.474, and large

when Cliff's delta ≥ 0.474[59]. On account of selecting 25 random trials from the no-Estim condition, the above procedures were repeated 500 times, with the aims of testing whether the outcomes of the statistical comparisons (Estim vs. no-Estim response strength) were reliable and independent of the randomized trial selection.

**Statistical procedures**. All statistical analyses were made with custom-written MatLab scripts. Paired and unpaired statistical comparisons were performed with Wilcoxon singed-rank and rank sum tests, respectively. These are appropriately indicated in the text, together with sample sizes and p-values. All statistics, unless otherwise noted, were corrected for multiple comparisons with the False Discovery Rate approach, using the Benjamini and Hochberg procedure[90]. An alpha of 0.05 was set as threshold for statistical significance. The effect size metric used, unless stated otherwise (as in the GLM case), was Cohen's d:

$$d = \frac{\mu_{D1} - \mu_{D2}}{\sqrt{\left(\frac{(n_1-1)\sigma_{D1}^2 + (n_2-1)\sigma_{D2}^2}{n_1 + n_2 - 2}\right)}}, \quad (7)$$

where D1 and D2 are two distributions, $\mu$ represents the mean, $\sigma^2$ represents the variance, while $n_1$ and $n_2$ are the sample sizes. Effect sizes were considered small when $|d| < 0.5$, medium when $0.5 \leq |d| \leq 0.8$, and large when $|d| > 0.8$[91].

To test differences in the connectivity graphs across conditions (e.g., echolocation vs. communication, or passive listening vs. active vocalization), we obtained adjacency matrices for each of the 500 penetrations (one per dPTE connectivity matrix; see above) and compared the distributions using Wilcoxon signed rank tests. Given that the large sample size ($n = 500$ here) increases the occurrence of significant outcomes in statistical testing, edges were only shown when comparisons were significant and produced large effect sizes ($|d| > 0.8$).

When comparing connectivity graphs between pre-voc and post-voc conditions, we used the exact same trials per repetitions to construct the distribution of dPTE matrices for the pre- and post-voc cases. A certain repetition $m$ for each condition was then treated as paired, and therefore Wilcoxon signed rank tests were used for comparing (as opposed to unpaired statistics above). Again, only edges representing significant differences ($p_{corr} < 0.05$) with large effect sizes were shown.

**Reporting summary**. Further information on research design is available in the Nature Research Reporting Summary linked to this article.

## Data availability

The data generated during and/or analyzed during the current study are available in the G-Node GIN repository, https://doi.org/10.12751/g-node.q6xwhi/. Some data could not be uploaded due to their size; they are available from the authors upon request. Source data are provided with this paper.

## Code availability

Essential code and data are available from the same GIN-Node GIN repository (https://doi.org/10.12751/g-node.q6xwhi/). Further materials are available from the authors upon request.

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

## Acknowledgements

This work was supported by the DFG (Grant No. HE 7478/1-1, to J.C.H.), and the Joachim-Herz Foundation (Fellowship granted to F.G.R.). The authors thank Gisa Prange for assistance with histological procedures.

## Author contributions

F.G.R. and J.C.H. conceived and designed the research. F.G.R. collected the data, analyzed it, produced figures, and wrote the original manuscript. F.G.R., L.L.J., E.G.P., J.W., Y.C.C., A.K., M.K., and J.C.H. discussed analyses and results, interpreted the data, and reviewed figures and text.

## Funding

## Competing interests

The authors declare no competing interests.
