## [Peer Review File · Nature Communications]

Echolocation-related reversal of information flow in a cortical vocalization networkReviewers' comments:

Reviewer #1 (Remarks to the Author):

This is an interesting study with intriguing results, demonstrating directional transition of information, or more accurately, oscillatory activity, between the frontal and auditory cortices that depends on the purpose of a vocalization. The results are obtained using the classical electrophysiology technique of recording local field potentials and simultaneously recording and analyzing temporal and spatial patterns of neural activity within the frontal and auditory cortices. A strength of the study is the detailed analysis of causality of the flow of neural activity within the brain of freely vocalizing bats.

There are several critical problems in the terminology and conceptualization, as written, in the abstract, results, including figures, and a few places in the discussion.

Some of the problems in the abstract, which is the most widely read part of the paper, are listed as follows:

Line 20: "fundamental structures" is not good usage.

Line 22: "Here, we address this issue ..." is a vague statement to include in an abstract, especially since it further relates to a general description relating to "patterns of information exchange" without specifying anything about the patterns.

"... means of electrophysiological recordings..." is also nonspecific and does not need to be included in the abstract.

Line 24: "vocalization type" is used twice without specifying the types of vocalizations. Also, both types of vocalizations are inconsistently referred to as "calls". Among calls, or more accurately "social calls", there are many "call types" (not including "echolocation pulses") for the same purpose of social communication. However, the different types of vocalizations that are referred to in this study have different purposes or "behavioral consequences" as alluded to in the last sentence of the abstract and elsewhere. This is in fact the main functional difference, which leads to differences in information transfer among the frontal and auditory cortices. This needs to be clearly indicated throughout the manuscript as it makes the results more meaningful. "Vocalization types" could be used to stress the acoustic differences; however, it appears that what really differentiates the flow of information between the two vocal categories is the purpose of the vocalization. Therefore, it is better to stress this as the main difference that is reflected in the two different acoustic patterns of the vocalizations.

Line 26: "... in the circuit...", which circuit?

Line 28: it is better to specify "bottom-up" and "top-down", as now indicated in the revised abstract. Similarly, the "frontal cortex" is suddenly referred to as the association network on line 31, which can cause some confusion to the reader.

Line 29: "... imminent post-vocal consequences..." can occur for both echolocation and communication signals. So, this distinction is vague and is also not accurate.

Here is a suggested modified version of the abstract:

"The mammalian frontal and auditory cortices are fundamentally important for supporting vocal behaviour. Here, using local field potential recordings, we demonstrate for the first time that the timing and spatial pattern of oscillations in the fronto-auditory cortical network of freely-vocalizing bats (*Carollia perspicillata*) predict the purpose of vocalization. Transfer entropy analyses of oscillatory activity revealed predominantly top-down (frontal-to-auditory cortex) information flow. The dynamics of information flow depended on the behavioral role of the vocalization and on the timing relative to vocal onset. Remarkably, we observed the emergence of predominantly bottom-up (auditory-to-frontal cortex) information transfer only when animals produced echolocation vocalizations leading to self-

directed acoustic feedback. These results reveal changes in information flow across sensory and frontal (association) cortex driven by the purpose (echolocation vs. social communication) of the vocalization in a highly vocal mammalian model."

If the above version of the abstract with the changes accurately reflects the study results, please feel free to adopt this or incorporate changes in a modified version. These changes also need to be reflected in the rest of the text to make sure there is a consistent use of the terminology and concepts that do not confuse the reader and accurately reflect the findings. If additional specific results can be included in the abstract, that is better.

Most of the introduction is fine, except the following:

Line 54 "... including vocalizations..." appears to be redundant.

Line 90: "... directed information transfer..." is vague. A more specific statement is needed for the hypothesis. Also, "top-down" and "bottom-up" are not really "mechanisms", they simply reflect differences in the timing and flow of neural activity.

Line 114: The authors start to use the term "sonar and non-sonar calls" here. This should be avoided. To be consistent it is best to avoid the "sonar and "non-sonar" terms as distinction, as among other reasons, sonar includes an echo and not just the pulse, which is being produced and investigated here. Was there an echo produced from the pulses emitted by the bat and did the bats hear them? Please clarify this issue.

The terms echolocation and communication indicate functional differences, or "echolocation pulses" or simply "pulses" and "social calls" or simply "calls" indicate acoustic distinction. Also, please provide some information on the spectrograms of the social call vocalizations. Are these of one type or a few different types or very diverse? Some information on the spectral distribution is needed to categorize this class of sounds as social calls given the increasing body of acoustic, behavioural, and neuroscientific literature on this class of vocalizations in echolocating bats.

Does the physiological state of the animal affect the oscillatory activity or its directional transition? How was the physiological state of the animals controlled?

Line 117: Does the "stereotypical design" apply to echolocation pulses or also to social calls? Usually social calls are not very stereotypic. Note as well that once again echolocation vocalizations are categorized as "calls", which they are not and this usage here, on line 436 and elsewhere, muddies the entire context of the differences in information flow.

Line 566: "... preparatory signals in AC ... "?

Line 689: "... makes it possible ..." delete "of" .

Line 725: change "sensible" to "sensitive".

Line 775: What was the basis, if any, of grouping channels? What other combinations were examined, and would that change the results?

Line 1132: Change "sonar calls" to "echolocation vocalizations". " ... if two" or should it be "if three"?

Line 1150: "... neural correlate of..."

Line 1158: Change "... Type of call produced..." to "purpose of vocalization"

Please modify "sonar" and "nonsonar" labels in all figures to possibly "pulse" and "call".

Please provide some additional examples of spectrograms particularly of calls given the wide range of their duration.

Please clarify the time axis in figure 1B. Are there some echolocation pulses that are 30 ms long? Are these single pulses or trains of echolocation pulses or single pulses? How does the spectrogram appear of such a pulse? Change "call duration" to "sound duration".

Figure 1D should read "time from vocalization onset" instead of " from call onset" for reasons stated above.

Figures 4a and 5a may be enlarged and combined to show only "echolocation vs. communication" differences. This will improve clarity. The 'b' and 'c' parts of these figures could be moved to the supplementary section as these figures appear cluttered with too much and possibly not-so-relevant information. This may improve the readability of the paper, but the authors are in the best position to decide to do so.

Figure 7a is unclear. To which regions do the black and gray traces correspond? Change sketches in 7b to names FAF and AC to simplify.

Reviewer #2 (Remarks to the Author):

In this manuscript, the authors examined the relationship between ongoing neural activity, measured as LFPs, in the FAF and AC of an awake, retrained echolocating bat, during pre/post/non-vocalization periods for both echolocating calls and presumed social calls. They report that pre-vocalization, there is a net flow of information in the "top-down" direction from FAF to AC, primarily in the delta range of frequencies. Post-vocalization, there is a reversal in the direction of this information flow, from AC to FAF. This reversal is more prominent after sonar vocalizations. They also characterize some of the waveform details in FAF and AC and hypothesize that there are separate neural generators for the oscillatory patterns seen in these two structures.

Overall, this was a very interesting study that appears to be carefully-done and the manuscript is well written and illustrated. My main concern here is that the impact of the work is incremental and provides very little mechanistic information. In addition, the differences between sonar and non-sonar calls was relatively small and could potentially be explained by different tuning characteristics of AC (see below). The differences in waveforms in fig 2 is interesting, but on its own is not a particularly convincing demonstration of different mechanisms in the two regions. My specific concerns are laid out below:

Major:

1. It is challenging with these species to conduct mechanistic studies, but this study essentially raises an interesting hypothesis about the role of FAF in prepping AC to receive incoming information, and then AC "updating" FAF once that information is received. Thus, it would be important to perturb or manipulate the nature of the FAF-AC information flow to determine if such a perturbation would then alter how AC updates FAF, or how AC processes the incoming information.
2. Although bats are used here because of their ability to connect brain physiology to behavioral changes, the setup used here was a very non-ethologically relevant one. The bats are echolocating while stationary and emitting putative communication calls in an environment without conspecifics. Thus, it is not clear how such findings would correlate with neural changes that would occur had these events been recorded in a natural environment. Further, it would be useful to know how similar each of the emitted calls are to those emitted naturally. For example, most bats vary the parameters of their sonar calls while hunting – which calls were the recorded sonar calls in this study most similar to? Likewise for the putative social calls. What kinds of calls were these? Were they distress calls? Did the neural pre/post activity differ based on the particular putative communication call being emitted?
3. What portion the AC is each electrode in? Does it matter? As the authors know well, different subfields of the AC or subregions within the primary AC will be differentially involved in processing

low- vs. high- frequency sounds (which differentiates sonar vs. non-sonar calls), which may potentially be responsible for the sonar/non-sonar differences seen here. Can the authors provide more information about the location of their AC electrode and discuss the implications of its location further?

4. The authors are reporting p-values as low as $10E-115$. Those are not reasonable p-values for a biological system. The p-values come from a model that only approximates biological reality, and thus it is likely such tiny p-value estimates are not accurate. $P=10E-115$ suggests that one would need to repeat the experiment $10E115$ times to come across the same value by chance. I would suggest choosing a more reasonable value, like 0.001 or 0.0001 and stating that p is less than that number.

5. In portions of the manuscript, the authors are careful in their use of the word “causal” and in other portions they are not. One cannot infer causality based on any of the approaches used here. Please insert modifiers (e.g. “potentially causal” in line 219) when using the word “causal” in this manuscript.

Minor:

1. Why were beta 2 and gamma 1 frequencies not examined?
2. Line 415 mentions LFP changes “during” vocal production, but no assessments during vocal production were made
3. Line 423 should be “such a position”
4. Line 634 – what is meant by “posterior analyses”? Perhaps rewrite as “later analyses”?
5. Line 685 – I believe it should state that medium effect sizes should be ≥ 0.1 and < 0.4

Reviewer #3 (Remarks to the Author):

Garcia-Rosales et al., Echolocation reverses information flow in a cortical vocalization network, NCOMMS-21-19228

The experiments described in the manuscript investigate the patterns of information exchange between two cortical regions, the frontal auditory field (FAF) and the auditory cortex (AC) in awake and spontaneously vocalizing bats (*Carollia perspicillata*). Local field potentials (LFPs) were simultaneously recorded with multi-channel electrodes in both cortical regions. A number of elaborate statistical analysis methods is applied to reveal a predominantly top-down direction of information flow between the FAF and the AC. The direction of information flow was reversed after the emission of sonar calls. Generally, oscillations in the FAF and AC predicted the type of vocalization (sonar or non-sonar).

The experiments are technically sound and the manuscript is well written. The results are new and interesting and should be of general interest for a broader scientific audience, but especially important for scientists interested in neural coordination of vocal communication and cortico-cortical network interactions.

However, shortcomings of the manuscript result mainly from incongruities in the interpretation of the data and some technical/methodological concerns. The following comments might help to improve the manuscript and make the functional interpretation of the results more convincing.

Major comments

1) Line 29/Line 554ff. Of course, it is clear that during echolocation processing of the reflected echoes is of major importance. However, social communication can also have “imminent post-vocal consequences”, as conspecifics might respond to the calls (i.e. during mother –infant communication).

The time-scale on which acoustic information as a response to an emitted call is received might be different in a social communication context, but information contained in a vocal response should be important. The authors should comment on this and explain why a reversal of information flow could not also be beneficial in vocal communication and not just only in the context of echolocation. Especially in the context of vocal learning bottom-up feedback control would also be important.

2) Line 96/Line 239/Line 504ff. The authors report causal interactions between the FAF and the AC during spontaneous activity. I was a bit surprised by that, because it sounds counter-intuitive to me. If the information-flow between FAF and AC is already directional during non-vocal periods, how can this information flow be considered to be specific in a vocal context?

If causal interactions during spontaneous activity are due to attention (as suggested by the authors in the discussion), it is not clear why top-down attentional effects are not observed during emission of sonar calls (information flow is reversed from AC to FAF). Especially during an echolocation context, attention should be important.

In this context, the line of argumentation of the authors in the discussion (second last paragraph, line 535-561) is not coherent: The authors state that in an echolocation context, the bat auditory system must be “ready” to process echo information. I think, “readiness” in this context would just be another word for attention.

Furthermore, if it would be true that, as the authors state in line 554 “communication calls information not aimed at the emitter itself”, the line of argumentation putting the FAF to AC directed information flow in the framework of attention would not be convincing. The authors should comment on this

3) The authors must show that differences in LFP oscillations (and thus predictability of call type from LFP) cannot simply be attributed to the large differences in the frequency content of sonar and non-sonar calls. Thus, the differences would be an epiphenomenon, not to be assigned to higher cognitive functions in vocal-motor control. Were there other communication calls emitted during experiments which had a spectral content more similar to echolocation calls, that can be used to check for the influence of the spectral content on LFPs? I know that this question was addressed in Weineck et al. 2020, Fig.5. However, I found the data presented there not very convincing, as the spectral structure of the LHF communication calls still was very much different from the echolocation call spectral content, especially in the low-frequency range.

Also, the influence of the acoustic stimulation by the calls should therefore be clarified (most important during the post-voc periods), especially as the authors state, that in an earlier paper (Garcia-Rosales et al. 2020) passive acoustic stimulation also had a profound effect on oscillations in the FAF and AC. In this respect, it might be important that the recordings in the AC were derived from the high-frequency region. However, the frequency-preference of neurons in the FAF in the region the electrode was inserted is not described in the manuscript. Furthermore, sonar and non-sonar call are typically emitted differently in phyllostomid bats: sonar calls are emitted thru the nostrils, typically associated with movements of the nose leaf, while non-sonar calls are often emitted through the mouth. A possible differential influence of motor-activity on the LFPs should be discussed.

4) To evaluate the recording position within the cortical layers, it would be helpful to indicate layers borders in Figure 1d for FAF and AC separately. It is important to consider that the cortical thickness in the AC and FAF might be different. This is especially important as data from different electrode channels were pooled for some of the analyses described in the manuscript (line 229).

In the methods section, (line 590) it is stated that “In the AC, recordings were made mostly in the high-frequency field”. What does “mostly” mean? This should be quantified. How many electrode penetrations were located in other regions? How were recording sites verified after the experiments had been finished? How large was the variation of positions of recording site within the AC and the FAF in the three bats?

5) In a previous paper using similar methods (Weineck et al., 2020), in the FAF high-frequency power only predicted the vocal output (Weineck et al., 2020, Fig 6). In the present manuscript low frequency power in the FAF predicted vocal output, too (line 146, Fig1g). As the methods to record LFPs seem to be basically the same in both manuscripts, this difference should be commented to support reliability of analyses and methods. If activity in different frequency bands would not be reproducible within different data sets, then much of the detailed analyses presented in the current manuscript would not be very meaningful.

Minor comments

Line 52/451. Suppression of neural response to self-produced sounds should not occur during echolocation, as the emitted call is an important marker for echo-delay computation. Although the data by Li et al., 2020 might indicate that corollary discharges might also enhance processing of self-generated sounds, it is not clear if the time-scale on which the dynamics of pre-vocal power described in the manuscript occur would fit the requirement for fast processing of echolocation sequences. Please, comment on this.

Line 112 and Figure 1b. Although the difference was not significant, sonar calls should be generally shorter, compared to the non-sonar calls, as indicated in figure 1a. However, the distribution of sonar-call lengths seems to be double-peaked. Please check.

As far as I know, the Kolmogorov-Smirnov-Test assumes a normal distribution of data whereas the Wilcoxon Rank Sum Test is a non-parametrical test not assuming normal distribution of data. I think both tests should not be used in combination.

Line 120ff, Fig. 1d: To illustrate base line activity, it would be helpful to also show LFP traces which are totally unrelated to vocal activity (say, 1-2 seconds before/after vocalizations, or show the time period used to determine no-voc activity in Fig.1e).

Line 120, Fig. 1d. A scale bar for the y-axis should be added. It must be clear that all traces are shown on the same scale.

Line 154. As the FAF receives sensory input via two different pathways (the classical ascending auditory pathway via the AC, but also via the extra-lemniscal pathway by-passing the AC), it should not be exclusively labelled "association cortex".

Line 414/435. While the authors state that "LFPs in FAF and AC are causally related (within a TE framework) during vocal production..." they also claim that there is no causal role of LFPs for initiation or planning of sonar or non-sonar calls. This sounds contradictory. If LFPs in FAF and AC are causally related during vocal production, and LFPs predict vocal output, why is there no causal role of LFPs for initiation or planning of sonar or non-sonar calls? The authors should comment on this in more detail.

The same hold true for line 501: "...causal interactions (within a TE framework)...with functional relationship to vocalization". Here again, a causal relation to vocalization is at least implied. This statement should be avoided if there is no causal role of LFPs for initiation or planning of sonar or non-sonar calls, as the authors claim in line 435.

Line 440. "...as previously reported..." Does this refer to Tsunada and Eliades, 2020? Please clarify.

Fig. 4,5,6. The figures are very difficult to read as they look very much cluttered, especially the line plots in top rows. I would recommend simplifying the figures. Maybe only show the most important information.

Line 632/638: "...a preliminary classification ...was done based on each call's peak frequency...". "Finally, vocalizations were examined via visual inspection to validate their classification". Were additional parameters other than peak frequency used for classification? Were the non-sonar-calls a homogenous group, or would a further classification into sub-classes have been possible?

Line 1060, Legend to Figure 1: the sub-figure labels are not correct (should be e,f,g not e,e,f)

We thank the Editors and the three anonymous referees for their comments and review of the original version of our study. The manuscript, in the form presented here, is a revised and corrected version of the original. Many changes have been made in accordance to the reviewers' comments. As a result, a more transparent and straightforward study is submitted for evaluation. An overview of the modifications is given below:

- General revision of the text to improve readability and transparency, taking into account suggestions and comments made by the reviewers.
- Based on the concerns of Reviewer 2, we decided to remove the data shown in the original Fig. 2 (i.e. differences between frontal and auditory cortices in terms of oscillatory shape). We agreed with the reviewer and further considered that the data did not contribute to the main message of our paper. We believe that this decision has led to a more straightforward and engaging work.
- New data have been included to support the main findings of the study. For example, we evaluated whether the tuning characteristics of the AC would explain our original results, which advance unique information exchange dynamics in the FAF-AC network during echolocation. These analyses were done with data from passive-listening animals, and using data of animals vocalizing high-frequency communication sounds. We show that the tuning properties of AC, or the production of high-frequency sounds and their auditory feedback, do not fully account for the main results of the study (Supplementary Figures S7-10; Supplementary Text).
- Inspired by the first major comment of Reviewer 2, we performed further experiments in 2 new animals in order to explore whether a perturbation of the frontal cortex alters in any way responses in the AC. We show that electrical stimulation of the FAF results in an increased response strength in AC, much more reliably when animals listen to echolocation sounds as compared to when communication sounds were presented. These new results strongly support the notion that FAF modulates auditory cortical responses, in a manner that appears to favour echolocation processing. These data are presented in the new Fig. 4 of the main text, together with a new sub-section in the Results.
- Following recommendations made by the reviewers, some figures were merged and comparisons that did not directly contribute to the main conclusions of the manuscript were moved to the supplementary materials. We believe that this largely improves clarity and readability. In addition, such changes allowed us to present a revised text which complies with the length restrictions of the Journal.
- Supplementary figures were included to illustrate recording positions, and complementary analyses.

In the following, the reviewers and the Editor will find a point-by-point response to each comment and a more nuanced description of the revisions made to the original manuscript.

Reviewer #1 (Remarks to the Author):

This is an interesting study with intriguing results, demonstrating directional transition of information, or more accurately, oscillatory activity, between the frontal and auditory cortices that depends on the purpose of a vocalization. The results are obtained using the classical electrophysiology technique of recording local field potentials and simultaneously recording and analyzing temporal and spatial patterns of neural activity within the frontal and auditory cortices. A strength of the study is the detailed analysis of causality of the flow of neural activity within the brain of freely vocalizing bats.

There are several critical problems in the terminology and conceptualization, as written, in the abstract, results, including figures, and a few places in the discussion.

We would like to thank the reviewer for the insightful comments on the original version of the manuscript. Below, we address the raised concerns point by point.

Some of the problems in the abstract, which is the most widely read part of the paper, are listed as follows:

Line 20: "fundamental structures" is not good usage.

Thank you. As we adopted the reviewer's version of the abstract for the most part in the current revision, this has been addressed.

Line 22: "Here, we address this issue ..." is a vague statement to include in an abstract, especially since it further relates to a general description relating to "patterns of information exchange" without specifying anything about the patterns.

Thank you. As mentioned above, this has now been addressed in the new abstract; the vague statement was removed.

"... means of electrophysiological recordings..." is also nonspecific and does not need to be included in the abstract.

Thank you. This was removed.

Line 24: "vocalization type" is used twice without specifying the types of vocalizations. Also, both types of vocalizations are inconsistently referred to as "calls". Among calls, or more accurately "social calls", there are many "call types" (not including "echolocation pulses") for the same purpose of social communication. However, the different types of vocalizations that are referred to in this study have different purposes or "behavioral consequences" as alluded to in the last sentence of the abstract and elsewhere. This is in fact the main functional difference, which leads to differences in information transfer

among the frontal and auditory cortices. This needs to be clearly indicated throughout the manuscript as it makes the results more meaningful. "Vocalization types" could be used to stress the acoustic differences; however, it appears that what really differentiates the flow of information between the two vocal categories is the purpose of the vocalization.

Therefore, it is better to stress this as the main difference that is reflected in the two different acoustic patterns of the vocalizations.

We appreciate the reviewer's comment. We made several changes to the main text related to it, the most direct of them being the substitution of terms "sonar" and "non-sonar" for "echolocation" and "communication". We further agree with the reviewer that there are various ways in which one can categorize the vocal outputs, and for this the construct "vocalization type" could be misleading. Indeed, different vocalization types could refer to echolocation/communication or various types of communication utterances (even when differentiated merely by acoustic structure, let alone behavioural meaning). In the paper we stress as much as possible, while still trying not to overwhelm or patronize the reader, that "vocalization type" in this context refers to either echolocation or communication (denoting of course the ethological differences of these). On occasions, for better readability, we favour using "vocalization types". However, in the revised version of the paper we also strove to remind the reader that these two "types" have indeed distinct purposes (as we have done, for example, by adopting in the abstract the words kindly shared by the reviewer). In light of the reviewer's comment, we hope we have struck an appropriate balance in the new version of our study.

Line 26: "... in the circuit...", which circuit?

Thank you for noticing this. We intended to refer to the FAF-AC circuit, although without making this explicit beforehand. It was a mistake that has been corrected.

Line 28: it is better to specify "bottom-up" and "top-down", as now indicated in the revised abstract. Similarly, the "frontal cortex" is suddenly referred to as the association network on line 31, which can cause some confusion to the reader.

Thank you the remark. We adopted these changes. With "association network" we referred in line 31 to a network formed by sensory and association cortices. We apologize for the lack of clarity; this has been corrected.

Line 29: "... imminent post-vocal consequences..." can occur for both echolocation and communication signals. So, this distinction is vague and is also not accurate.

Indeed. Thank you.

Here is a suggested modified version of the abstract:

"The mammalian frontal and auditory cortices are fundamentally important for supporting vocal behaviour. Here, using local field potential recordings, we demonstrate for the first time that the timing

and spatial pattern of oscillations in the fronto-auditory cortical network of freely-vocalizing bats (Carollia perspicillata) predict the purpose of vocalization. Transfer entropy analyses of oscillatory activity revealed predominantly top-down (frontal-to-auditory cortex) information flow. The dynamics of information flow depended on the behavioral role of the vocalization and on the timing relative to vocal onset. Remarkably, we observed the emergence of predominantly bottom-up (auditory-to-frontal cortex) information transfer only when animals produced echolocation vocalizations leading to self-directed acoustic feedback. These results reveal changes in information flow across sensory and frontal (association) cortex driven by the purpose (echolocation vs. social communication) of the vocalization in a highly vocal mammalian model."

If the above version of the abstract with the changes accurately reflects the study results, please feel free to adopt this or incorporate changes in a modified version. These changes also need to be reflected in the rest of the text to make sure there is a consistent use of the terminology and concepts that do not confuse the reader and accurately reflect the findings. If additional specific results can be included in the abstract, that is better.

We thank you for the support in sharing this version of the abstract. We took the liberty of adopting it with some minor changes.

Most of the introduction is fine, except the following:

Line 54 "... including vocalizations..." appears to be redundant.

We appreciate this remark. However, given that self-produced sounds are not necessarily vocalizations (e.g. sounds as the animal steps, walks, flies, lands on a platform, etc.), we intended to stress that the production of those sounds *and also* vocalizations triggers suppression in the auditory cortex. In that sense, our choice of words is somewhat redundant, although we believe it's a stylistic redundancy that stresses the point of vocalization and cortical suppression.

Line 90: "... directed information transfer..." is vague. A more specific statement is needed for the hypothesis. Also, "top-down" and bottom-up" are not really "mechanisms", they simply reflect differences in the timing and flow of neural activity.

Thank you. In the new version of the manuscript the hypothesis is more clearly stated, and the ambiguities highlighted by the reviewer have been corrected. The corrected version of the paragraph to which the Reviewer refers here can be read below:

The goal of this study was to unravel the nature of information exchange in the bat's FAF-AC network, and to understand whether information between these structures flows in accordance with the canonical roles of the frontal cortex for vocal coordination, and of the AC for feedback control. We further aimed to address whether the distinct behavioural contexts of echolocation and communication affect the dynamics of information transfer in the fronto-auditory circuit. We found complex causal interactions (within a transfer entropy framework) between frontal and auditory cortices during spontaneous activity and periods of vocal production. Directed connectivity in the

FAF-AC network varied dynamically according to whether animals produced communication or echolocation calls, and to the timing relative to vocal onset. For echolocation the changes were drastic, resulting in a reversal of information flow from pre-vocal to post-vocal periods. Altogether, our data suggest that dynamic information transfer patterns in large-scale networks involved in vocal production, such as the FAF-AC circuit, are shaped by the behavioural consequences of produced calls.

Line 114: The authors start to use the term "sonar and non-sonar calls" here. This should be avoided. To be consistent it is best to avoid the "sonar and "non-sonar" terms as distinction, as among other reasons, sonar includes an echo and not just the pulse, which is being produced and investigated here. Was there an echo produced from the pulses emitted by the bat and did the bats hear them? Please clarify this issue.

We appreciate the reviewer's suggestion. Terms "sonar" and "non-sonar" were substituted with the terms "echolocation" and "communication". The change also aims to highlight the main difference in the purpose of the vocalization, as the reviewer expressed in previous remarks. Sonar vocalizations emitted by the bats likely resulted in returning echoes from objects in the chamber. Although the walls of the cage were padded with non-reflecting foam, the microphone placed directly in front of the animals would have been a source of echoes from echolocation pulses (with a microphone distance of 10 cm, and a speed of sound of 343 m/s at 20°C, the echoes would return to the animal after $\sim 5.8 \times 10^{-5}$ seconds). Other objects inside of the chamber in principle could have reflected the pulses, but bats were head-restrained and only faced the direction of the microphone. It is also reasonable to assume that objects would have reflected echoes associated to the production of communication calls (both low- and high-frequency).

The terms echolocation and communication indicate functional differences, or "echolocation pulses" or simply "pulses" and "social calls" or simply "calls" indicate acoustic distinction. Also, please provide some information on the spectrograms of the social call vocalizations. Are these of one type or a few different types or very diverse? Some information on the spectral distribution is needed to categorize this class of sounds as social calls given the increasing body of acoustic, behavioural, and neuroscientific literature on this class of vocalizations in echolocating bats.

Thank you for the suggestions. To improve transparency, we now make available a PDF containing spectrograms of all 872 vocalizations used in this study (link: [https://gin.g-node.org/fgrs092/FAF AC vocs GarciaRosales et al 2021/src/master/CallsUsed.pdf](https://gin.g-node.org/fgrs092/FAF_AC_vocs_GarciaRosales_et_al_2021/src/master/CallsUsed.pdf)). This PDF will be accessible to readers after publication, together with other data already associated to the paper upon submission.

Regarding the second part of the reviewer's comment, we observed diverse spectrotemporal patterns of communication calls. However, we could not associate these patterns to specific social/behavioural contexts as we did not have a behavioural readout from the bats. We followed the methodology described in (Hechavarria et al., 2016) and clustered the bat vocalizations according to their spectrotemporal dynamics; these results are illustrated in **Fig. R1**. The figure illustrates that although

communication utterances were very similar among each other, communication calls were more variable than echolocation calls. For example, ~81% of echolocation calls (112/138) were grouped in 3 templates that did not differ largely from each other (**Fig. R1a, b**; their main differences consisted on the strength of the high-frequency harmonics of the bats' utterances). However, in the case of communication calls, ~80% of them (586/734) were associated to 12 clusters (**Fig. R1c**) which did exhibit some variability in the vocalizations' spectrotemporal patterns. The most populated 8 clusters (making up 502 out of the 734 calls, ~68%) are depicted in **Fig. R1d**. At the reviewer's discretion, we can add the information depicted in **Fig. R1** to the supplementary materials of the manuscript.

Fig. R1. Consistency and variability of echolocation and communication calling. (a) Cumulative probability distribution of number of echolocation vocalizations throughout call templates. The three most populated templates account for ~80% of the echolocation calls. (b) Spectrograms of one representative call for each of the three most populated echolocation templates. Horizontal bars indicate 1 ms. (c) Same as in a, but considering communication calls. Note that the 12 most numerous templates account for ~80% of the communication utterances recorded. (d) Representative vocalizations of each of the 8 most populated templates (one per template; 502/734 calls in total, ~68%). The number of calls associated to each template is indicated in the figure.

Does the physiological state of the animal affect the oscillatory activity or its directional transition? How was the physiological state of the animals controlled?

A large body of literature indicates that oscillatory activity, mostly in frontal regions, is affected by the physiological or emotional state of the animal (for reviews, see (Arnsten et al., 2015; Okonogi and Sasaki, 2021)). Exposure to stress, for example, not only disrupts oscillations in the prefrontal cortex but also appears to weaken the top-down cognitive control exerted from this area. To date, it is unknown if the same would happen in bats, particularly in the case of *C. perspicillata* (the animal model of our study). Unfortunately, we did not monitor physiological variables (heart rate, respiration rate, cortisol levels, etc.), and therefore cannot address whether the animal's state affects our results. We did have a readout from the microphone that allowed us to monitor if animals were moving too excitedly (also detectable

from a deterioration of the signal-to-noise ratio in the electrophysiological signals). However, whenever this happened the experimental session was terminated for the day following regulations for animal experimentation in Germany.

Line 117: Does the "stereotypical design" apply to echolocation pulses or also to social calls? Usually social calls are not very stereotypic. Note as well that once again echolocation vocalizations are categorized as "calls", which they are not and this usage here, on line 436 and elsewhere, muddies the entire context of the differences in information flow.

The reviewer is right. The wording was not clear and slightly misleading. We have corrected this sentence. However, we would like to bring the attention of the Reviewer to **Fig. R1** and our answer associated with it. According to our analysis, communication calls measured in our experiments are actually relatively stereotyped, even though they are still more variable than echolocation calls.

Line 566: "... preparatory signals in AC ... "?

Thank you for noticing this. It was not clear in the original text what was meant by “preparatory signals in AC”. This statement also suggests that signals in AC have some degree of involvement in the preparation of vocal outputs, which would have been an unsubstantiated claim. In the revised version of the manuscript the Discussion has been rewritten and this phrase has been removed.

Line 689: "... makes it possible ..." delete "of" .

Thank you. Done.

Line 725: change "sensible" to "sensitive".

Done.

Line 775: What was the basis, if any, of grouping channels? What other combinations were examined, and would that change the results?

Channels were grouped according to the laminar architecture of the AC. In the new Supplementary Figure 1, we show how channel groups relate to cortical layers. For example, channels grouped as “superficial” consisted of electrodes situated in layers I-II; channels in the middle groups (mid1 and mid2) spanned depths corresponding to mid layer III, down to mid layer V; channels in the “deep” group consisted mostly of depths spanning the bottom of layer V, and layer VI. In the FAF, layers are less anatomically and functionally demarcated, making it harder to establish a proper grouping based on laminar architecture.

In the paper (now Supplementary Figure 4) we show the full directed phase-transfer entropy (dPTE) matrices. These matrices are the foundations of the graph representation of the FAF-AC directed connectivity. There is no grouping done in the matrices shown in Fig. S4; rather, connectivity values are

shown on a channel-per-channel basis in both structures. The patterns described in, for instance, Figures 2 and 3, are clearly visible in the dPTE matrices, and thus the channel grouping is not a determining factor for our main results. Indeed, it is apparent that matrices are rather poorly resolved spatially (which is yet another argument for why grouping does not affect the outcomes). Poor spatial resolution may be a consequence of the nature of LFP signals, which are not local to a single electrode in the shank (in either FAF or AC), being well-correlated hundreds of micrometres away from the recording site.

Line 1132: Change "sonar calls" to "echolocation vocalizations". "... if two" or should it be "if three"?

Indeed, it should be "three". This has been fixed. Also, "sonar" has been changed to "echolocation" throughout the manuscript.

Line 1150: "... neural correlate of..."

Fixed.

Line 1158: Change "... Type of call produced..." to "purpose of vocalization"

Thank you. Done.

Please modify "sonar" and "nonsonar" labels in all figures to possibly "pulse" and "call".

We appreciate the reviewer's suggestion. These labels were modified to "echolocation" and "communication". However, we would like to point out that the word "calls" is also used in the literature to refer to echolocation vocalizations. Therefore, at times we speak of both vocalizations as "echolocation and communication calls" in the text. This is meant to avoid repeating "echolocation and communication vocalizations" or equivalent phrases throughout. For the sake of readability, we would like to keep open the possibility of using "calls" to refer to the two vocal categories. However, if the reviewer considers this inaccurate, we are willing to correct it.

Please provide some additional examples of spectrograms particularly of calls given the wide range of their duration.

Done. See **Fig. R1** and the response to the comment associated with it. Please also note that the PDF with all the calls considered in the study (see above) will be available with the published article.

Please clarify the time axis in figure 1B. Are there some echolocation pulses that are 30 ms long? Are these single pulses or trains of echolocation pulses or single pulses? How does the spectrogram appear of such a pulse? Change "call duration" to "sound duration".

Thank you for noticing this. There were 9 communication calls that were classified as echolocation calls, because they had high energy in the 50-100 kHz range, and because, in the manual curation stage, the misclassification was not detected. We apologize for this mistake. Classification was re-checked and these calls were moved to the “communication” pool. In the current dataset, therefore, we have 138 echolocation and 734 communication calls (as opposed to the 147 / 725 of the original manuscript). All analyses were rerun with the corrected dataset; every figure was updated accordingly. The results did not change in a meaningful way, and therefore our conclusions remain unaltered.

In Fig. 1, we made the suggested change in the axis label.

Figure 1D should read "time from vocalization onset" instead of "from call onset" for reasons stated above.

Done.

Figures 4a and 5a may be enlarged and combined to show only "echolocation vs. communication" differences. This will improve clarity. The 'b' and 'c' parts of these figures could be moved to the supplementary section as these figures appear cluttered with too much and possibly not-so-relevant information. This may improve the readability of the paper, but the authors are in the best position to decide to do so.

Thank you for this suggestion. We combined the original Figs. 4 and 5 into one figure, which was moved to the Supplementary materials for the sake of readability, and to comply with length restrictions of the Journal. These data, along with others, are now presented in Figs. S5, S6. A quantification and a detailed description of the data to which the reviewer refers is given in the Supplementary Text. Thanks to the reviewer’s recommendation, we believe that the paper is now much more concise and straightforward.

Figure 7a is unclear. To which regions do the black and gray traces correspond? Change sketches in 7b to names FAF and AC to simplify.

We apologize for the lack of clarity in the figure. The summary figure (Fig. 5 in the current revision) has been re-worked to address the reviewer’s suggestion. We also added a summary for the new FAF electrical stimulation experiments, ran in response to the first major comment of Reviewer No. 2.

Reviewer #2 (Remarks to the Author):

In this manuscript, the authors examined the relationship between ongoing neural activity, measured as LFPs, in the FAF and AC of an awake, retrained echolocating bat, during pre/post/non-vocalization periods for both echolocating calls and presumed social calls. They report that pre-vocalization, there is a net flow of information in the “top-down” direction from FAF to AC, primarily in the delta range of frequencies. Post-vocalization, there is a reversal in the direction of this information flow, from AC to FAF.

This reversal is more prominent after sonar vocalizations. They also characterize some of the waveform details in FAF and AC and hypothesize that there are separate neural generators for the oscillatory patterns seen in these two structures.

Overall, this was a very interesting study that appears to be carefully-done and the manuscript is well written and illustrated. My main concern here is that the impact of the work is incremental and provides very little mechanistic information. In addition, the differences between sonar and non-sonar calls was relatively small and could potentially be explained by different tuning characteristics of AC (see below). The differences in waveforms in fig 2 is interesting, but on its own is not a particularly convincing demonstration of different mechanisms in the two regions. My specific concerns are laid out below:

Major:

1. It is challenging with these species to conduct mechanistic studies, but this study essentially raises an interesting hypothesis about the role of FAF in prepping AC to receive incoming information, and then AC “updating” FAF once that information is received. Thus, it would be important to perturb or manipulate the nature of the FAF-AC information flow to determine if such a perturbation would then alter how AC updates FAF, or how AC processes the incoming information.

We agree with the reviewer: given our original data and observations, it would be very interesting to determine whether a manipulation of the FAF alters the manner in which the AC responds to incoming acoustic information. We conducted new experiments to test this hypothesis; results are shown in the new Figures 4, S11 and S12, and in the new subsection of the Results: “Electrical stimulation of the FAF enhances auditory cortical responses” (to facilitate the reviewer’s task, Figure 4 was replicated below this answer as **Fig. R2**). Specifically, we delivered electrical stimulation to the FAF with biphasic pulses lasting 410 μ s (200 μ s per phase, 10 μ s gap between phases) and with an amplitude of 2 μ A, while simultaneously recording responses from the AC to externally presented acoustic stimulation. The electric pulses were delivered into the FAF by means of an A16 shank, from which two channels at depths of 350 and 450 μ m were used as stimulating electrodes. Stimulating channels received electric signals from a stimulus isolator (A365, World Precision Instruments). We initially intended to record from the remaining electrodes in this shank, but unfortunately signal acquisition in the FAF was not reliable enough (e.g. we lost signal from channels during recordings several times) to allow us to use data from this structure in the analyses, beyond calculating grand averages of the stimulation periods for illustrative purposes. Pulse amplitude was selected according to values used in the literature (e.g. (Atencio et al., 2014; Takahashi et al., 2019; Yazdan-Shahmorad et al., 2011)), and after empirically establishing that electrical artefacts were undetectable in the AC online recordings (see, for example, the panel **c** of the new Fig. 4).

The precise electrical and acoustic stimulation was as follows. Six biphasic electrical pulses were delivered with an inter-pulse interval of 500 ms (2 Hz, within the δ -band range). After the electrical pulse train, acoustic stimuli were presented to the bats at given delays relative to time of the last electrical pulse (10, 135, 260, 385, 510, 635, 760 and 885 ms; henceforth “Estim” condition). These delays were consistent to sampling at four equidistant phases per electrical stimulation cycle (a cycle considered

from pulse onset to pulse onset), for a total of 2 cycles after the last electrical pulse was delivered (see Fig. 4a). As acoustic stimuli, we used natural distress (a type of communication signal) and echolocation calls (one of each, see Fig. 4b) presented at 50 dB SPL (RMS). Each possible combination of acoustic stimulus type (i.e. distress or echolocation) and delay was presented 25 times, in a pseudorandom manner. Sounds were also presented without any prior electrical stimulation (“no-Estim” condition), with a variable inter-stimulus interval ranging from 6-10 seconds, a total of 50 times each. Responses from a total of 20 columns were recorded in the high frequency fields of the AC of 2 bats using a 16-channel laminar probe (A16, Neuronexus).

To test whether electrical stimulation in the frontal cortex altered the response properties of the AC, we directly compared auditory cortical responses to sounds (either the distress or the echolocation call) between Estim and no-Estim conditions. Response strength was measured as the area under the curve of the Hilbert transform of the high-frequency component (25-80 Hz) of the event-related potential (ERP) in AC, for the first 150 ms after acoustic stimulus onset. We used the high-frequency component of the ERP to avoid biases related to low-frequency pre-stimulus trends in the LFPs, which were prominent in the recording and did not disappear after averaging 25 trials (in the Estim condition). Because the number of times sounds were presented differed between the Estim and no-Estim conditions (25 and 50 trials, respectively), the response strength of the no-Estim condition was calculated using 25 randomly selected trials out of the total of 50. Note that the ERP is sensitive to the number of trials used because it is a trial-averaged response, and as such the number of trials for each condition must be equalized. Only then we compared the ERP energy across channels and acoustic latencies relative to the last electrical pulse delivered (FDR-corrected Wilcoxon signed ranks tests, paired; significance when $p_{\text{corrected}} < 0.05$). To evaluate the reliability of these comparisons, we repeated the above procedure 500 times.

Electrically stimulating the FAF prior to sound presentation led to stronger responses in the AC, as compared to the no-Estim condition. This effect was statistically significant, but was dependent on the sound that was being considered, on the depth of channels in the AC, and on the delay with which the sound was presented relative to the last electrical pulse delivered into FAF. For example, when the distress syllable was presented, statistically significant changes between responses related to Estim and no-Estim were confined to AC depths $> 300 \mu\text{m}$, but were not very reliable across all 500 iterations (i.e. only for $\sim 40\%$ of them) for sound delays of 135, 510, and 760 ms (see Fig. 4e, f, and the left portion of panel i). However, when the sound presented was an echolocation pulse, changes between Estim and no-Estim conditions were evident and robust (Fig. 4g, h, and the right portion of panel i). Auditory cortical responses to the echolocation sound were significantly larger for the Estim condition, particularly when the delay of the sound was 135 ms after the electrical stimulation, and at superficial channels ($50 - 300 \mu\text{m}$). Differences in this case were highly reliable, occurring up to 90% of the time (out of a total of 500 iterations). The fact that these effects were stimulus-specific, spatially resolved, and not strongest for a very short delay (10 ms), denotes that they were not merely a consequence of electrical artefacts in AC. We confirmed the former by running the same analysis on a window directly before acoustic stimulation.

These data indicate, first, that perturbation of the FAF leads to changes in the response properties of the AC and, second, that the manners in which auditory cortical responses to distress and echolocation sounds are affected by the distant electrical stimulation are different. The first finding supports the

conclusion that the FAF might play an important role in modulating how primary sensory areas process incoming acoustic information. The second finding suggests that only responses to echolocation are reliably affected by the perturbation of the frontal field. The latter might be related to our original observations that echolocation production uniquely modifies the nature of information exchange in the FAF-AC network. However, production and passive listening of echolocation calls are of course two very different circumstances, so therefore the above should be considered speculative. In all, we show that vocalizing echolocation signals changes the functional connectivity in the fronto-auditory network and, moreover, that listening to echolocation is particularly sensitive to perturbations in the frontal cortex (as compared to listening to distress calls). One could argue from these results that the FAF-AC circuit is specialized in dealing with echolocation-related behaviour, whether being during active sensing (i.e. vocalizing) or during passive listening. With that, we do not imply that the fronto-temporal network is not suited for communication and socialization tasks. Future research could unravel in detail how frontal and auditory cortices contribute to either of these behaviours.

Fig. R2 (Replicated from Fig. 4 of the revised main text). Electrical stimulation of the FAF increases response strength in AC. (a) Schematic representation of the paradigm for electrical and acoustic stimulation. The timestamps for acoustic stimulation (coloured according to delay; see also panels e-g) represent the delay of sound onset relative to the end of electrical stimulation train. (b) Oscillograms of the natural distress syllable and echolocation pulse used for acoustic stimulation. On

the right, the normalized power spectra of both calls are shown (orange, distress; blue, echolocation). (c) Broadband (0 – 10kHz), raw data recorded simultaneously from FAF and AC (at representative depths of 50, 250 and 600 μ m) illustrating a single trial of electrical stimulation. Note that no electrical artefacts are visible in AC. (d) Auditory cortical LFPs (left column), and time-course of their energy (right), in response to either the distress syllable (top) or the echolocation pulse (bottom). Responses corresponding to the no-Estim conditions shown in blue; responses related to the Estim condition, in red. (e) Strength of auditory cortical ERPs in response to the distress syllable, across all recorded columns ($n = 20$) and depths. In blue, responses associated to the no-Estim condition; in red, those associated to the Estim condition (data as mean \pm s.e.m). (f) Corrected p-values obtained after statistical comparisons between response strengths related to Estim and no-Estim conditions, across all channels and delays (paired, FDR-corrected Wilcoxon signed rank tests, $\alpha = 0.05$). (g-h) Same as in e-f, but dealing with responses to the echolocation pulse. (i) Proportion iterations (out of 500) in which responses associated to the Estim condition were significantly larger than those associated to the no-Estim condition (same test as above). Data are presented across all channels and delays analysed, for responses to the distress and echolocation sounds. (j) Median effect size (Cliff's delta) for the same comparisons summarized in i.

2. Although bats are used here because of their ability to connect brain physiology to behavioral changes, the setup used here was a very non-ethologically relevant one. The bats are echolocating while stationary and emitting putative communication calls in an environment without conspecifics. Thus, it is not clear how such findings would correlate with neural changes that would occur had these events been recorded in a natural environment. Further, it would be useful to know how similar each of the emitted calls are to those emitted naturally. For example, most bats vary the parameters of their sonar calls while hunting – which calls were the recorded sonar calls in this study most similar to? Likewise for the putative social calls. What kinds of calls were these? Were they distress calls? Did the neural pre/post activity differ based on the particular putative communication call being emitted?

The reviewer raises an interesting point. Bats, and particularly *C. perspicillata*, constitute a highly vocal mammalian model, which is excellent to study the neural circuitry underlying vocalization. Although animals are stationary and held in our preparation, echolocation continues to be useful to sample the chamber and the environment in which the animal is set. Therefore, we argue that echolocation serves the same basic purpose served in the wild: to form an acoustic image of the bat's surroundings. The former is supported by the fact that bats vocalized at their own volition, as they would in a natural scenario, putatively in order to update their spatial representation of their environment. The echolocation calls recorded in this study do not differ markedly from calls recorded when bats are freely flying or attached to a swing (Beetz et al., 2016a; Brinklov et al., 2011; Thies et al., 1998). It is possible to hypothesize that the dynamics associated to echolocation production described in this study for the fronto-temporal network would not change drastically under different, fully natural conditions. Of course, such hypothesis still requires empirical validation; the same goes for the opposite contention.

Tackling this question is admittedly difficult and technically challenging. At this point, studies on vocalizing bats, and naturally-behaving bats in general, are scarce. Notably, research done in the Ulanovsky lab on bat navigation, as well work from Moss's and Yartsev's laboratories, have accomplished recordings in unrestrained animals. The main constrain for freely-behaving animals is the weight and size of the recording apparatus. Many of the navigation studies use bats such as *Rousettus aegyptiacus*, a megachiroptera, that weigh over 150 g (e.g (Zhang and Yartsev, 2019)), and can therefore carry most wireless data loggers with ease. In a different study, Kothari and colleagues recorded from the inferior colliculus of flying *Eptesicus fuscus* bats, which weigh between 18 and 21 g (Kothari et al., 2018). However, the latter approach used a single electrode complex in one brain region. In our work, we

record with electrode arrays in two areas that do not neighbour one another (i.e. the FAF and the AC). An additional electrode in a distant region doubles the weight of the electrode shanks, adds weight to the recording device, and makes for a larger implant. Thus, with a species such as *C. perspicillata* (typical adult weight: 18-22 g), simultaneous recordings in the fronto-auditory circuit with laminar electrodes is extremely hard to accomplish if electrodes are to be implanted. The sheer weight of the implants, we believe, would also affect the behaviour and well-being of our animals.

With our above-stated hypothesis, however, we do not aim to disregard a growing body of literature supporting the notion that natural behaviour affects the manner in which neural circuits operate and interact in the brain. Our argument is rather that the nature of echolocation in our experimental setup resembles the nature of echolocation in the wild. Without a question, echolocation in the wild often occurs paired with active navigation, which might entail flying and other movements. The aim of the current study was to examine the dynamics of information exchange between the FAF and the AC associated to vocal behaviour. With this in mind, animal movement (e.g. wingbeat, head movements, etc.) could add confounding effects in the electrophysiological ensembles that would make hard to disentangle vocalization-related signals from more general movement-related activity. Therefore, a more controlled scenario where movement is minimized allows to more clearly identify the dynamics pertaining to vocal behaviour. Nevertheless, the contribution of movement related activity cannot be completely ruled out in our dataset. For example, as Reviewer #3 kindly notices, the differences in the way in which echolocation and communication calls are emitted (either through the nostrils or through the mouth, respectively) could have an impact in the LFP activity reported in the manuscript. This possible contribution of motor-related activity has been added to the discussion. Similarly, it can be argued that the patterns observed in our data stem from the different motor-related activity associated with producing vocalizations which differ strongly in their frequency content. Notwithstanding, we were able to demonstrate that the frequency content of the calls, and hence the motor-related activity associated with generating it, does not suffice to explain our main results (see below).

The matter of communication calling is an interesting and puzzling one. Animals in our study vocalize these calls freely, even when there are no conspecifics around. A reason for this could be that animals occasionally check for the presence of conspecifics by emitting calls, perhaps expecting a reply from another bat once they are heard. Another possible reason might be that the emission of such communication calls is a consequence of the combination of a particular set of internal factors that drive, with or without the presence of conspecifics, vocalization. This would be the case if the animals were emitting distress utterances. Because the animals were not explicitly placed under stressful circumstances beyond those inevitably associated to the experimental setup (i.e. being placed in the holder, inside a chamber, without the ability to move), these calls cannot be rigorously categorized as purely distress signals. In fact, the communication calls recorded by us differ from distress calls reported in other studies from our lab, i.e. they lack the temporal modulation characteristic of distress calls and are lower in peak frequency (10 kHz vs. 22 kHz distress; (Hechavarría et al., 2016)). We have observed similar types of social calls in the colony suggesting that they are not unique to isolated bats, although we cannot pinpoint in which behavioural contexts (other than isolation) they are emitted. It is difficult to attribute the vocalizations to specific behaviours, given that there was no behavioural data for cross-referencing.

It remains open whether the presence of an audience would alter our results on LFP spectral and connectivity dynamics, something that must be studied rigorously in the future. However, we are not fully convinced that the presence of other bats would affect our observations significantly, especially when considering oscillatory activity so close to the moment of call emission (+- 500 ms around vocal onset). Nevertheless, a response from a conspecific could alter the patterns of information flow seen during passive listening, because listening to a conspecific responding to an own vocalization might be of elevated importance to the bat. The same could be when animals are aware of an audience at the point of vocalization. These are interesting questions that we aim to address in future research.

By dividing our communication pool into low- and high-frequency groups (LF- and HF-communication), we were able to test whether call type significantly altered our results. By call type here we take the acoustic (spectral) signature of a vocalization. Indeed, and as shown in response to the next comment from the reviewer, we observed that the patterns of pre-vocal LFPs associated to low- and high-frequency communication calls were not considerably different with one another. That is, models performed poorly when predicting whether an LF- or an HF-communication call was produced, based on pre-vocal LFP power. However, models significantly predicted ensuing vocal type when considering pre-vocal power of echolocation and any type of communication call (see below). Similarly, differences in predominant information flow dynamics between HF-communication and echolocation supported our main conclusion that echolocation triggers unique information flow reversal in the FAF-AC circuit. In light of the above, it does not seem that call type, indexed here by the frequency content of a communication utterance, significantly affects our main results.

3. What portion the AC is each electrode in? Does it matter? As the authors know well, different subfields of the AC or subregions within the primary AC will be differentially involved in processing low- vs. high-frequency sounds (which differentiates sonar vs. non-sonar calls), which may potentially be responsible for the sonar/non-sonar differences seen here. Can the authors provide more information about the location of their AC electrode and discuss the implications of its location further?

We appreciate the reviewer's concern. Indeed, the location of recordings in the AC does matter, particularly when considering the processing of vocalization feedback as bats hear themselves. In the revised version of the manuscript, we directly tackled this possible confounding effect and addressed the reviewer's comment. In a first step, for the sake of transparency, we included a figure (Supplementary Figure 1; see also **Fig. R3** in this document) depicting the approximate locations of the 30 penetrations made, according to landmarks specified in the literature (see (Esser and Eiermann, 1999; Hagemann et al., 2010)).

Fig. R3. Schematic of penetration sites in the AC. Penetrations per animal are colour coded. Subfields in the AC as follows: HF1-2, high frequency fields 1 and 2; AAF, anterior auditory field; DP, dorsoposterior auditory field; A1-2, primary and secondary auditory cortices.

We then examined whether the spectral properties of the produced calls accounted for the difference observed in the neural activity between the cases of echolocation and communication production (Supplementary Figures 7-10 in the revised manuscript). Initially, we recorded responses to two acoustic stimuli to which the animals listened passively: one high-frequency frequency-modulated sound (“HF-FM”, 2 ms long, downward frequency sweep from 80 to 50 kHz), and one distress syllable (“distress”, 3.8 ms long). Both sounds are depicted in the revised version, Supplementary Figure 7. Passive listening to these two sounds did not account for the patterns observed during vocal production, and especially did not explain the differences observed when comparing across vocalization types. Neither stimulation with the “HF-FM” sound, nor stimulation with the distress syllable, resulted in the predominance of bottom-up information flow observed for echolocation-related post-vocal periods. In fact, the information transfer patterns observed were very similar across acoustic stimuli, and also very similar to those observed during spontaneous activity. This suggests that passive listening fails to alter significantly the information flow in the FAF-AC network for low LFP frequencies. Such observation resonates with the lack of significant changes in coherence between frontal and auditory cortices in low LFP frequencies when animals listen passively to naturalistic acoustic sequences (see (García-Rosales et al., 2020)). The above indicates that, in order to alter the connectivity dynamics in the FAF-AC network, more than passive listening of acoustic inputs is required. The contrast becomes evident if one considers the case of echolocation call production. Most importantly, however, these new analyses indicate that mere acoustic feedback does not constitute an alternative explanation for our original conclusions.

We also explored the possibility that the frequency content of the produced vocalizations would suffice to explain our original results (active feedback vs. passive listening). We separated communication calls into two sub-groups: HF-communication ($n = 155$; calls with more than half their power in the 50-100 kHz range), and LF-communication (communication calls that did not fall in the HF-communication group). The spectra of these calls are shown in Supplementary Fig. 3. We were able to demonstrate that the pre-vocal spectral power in both FAF and AC field potentials also predicts whether animals produce echolocation or HF-communication utterances. Power change patterns relative to baseline and dependent on vocal type were the same observed in our original results. However, spectral changes in

frontal and auditory cortices were poor predictors of whether animals vocalized HF- or LF-communication calls. These data, shown in Fig. S3, illustrate that the frequency content of the produced calls does not explain our original observations.

When considering the dynamics of information exchange in the FAF-AC circuit, we did observe bottom-up information transfer for post-vocal periods related to HF-communication calls (new Fig. S10). This occurred, however, in the θ and α bands of the LFP, and not in the δ band as shown for the case of echolocation production. The change in α for the HF-communication case did not constitute an information reversal when compared to pre-vocal periods (Fig. S10b); the change in θ , albeit consistent with an information reversal, resulted in values of bottom-up information transfer no different to those observed for post-vocal echolocation periods in the same LFP frequencies. Thus, there were changes in information transfer dynamics that were unique to echolocation production, especially taking into account the reversal of information in δ frequencies. These new results were taken into consideration and discussed in the revised version of the manuscript. Importantly, these data corroborate that the differences in the frequency content of the produced calls fail to fully explain our original observations.

4. The authors are reporting p-values as low as $10E-115$. Those are not reasonable p-values for a biological system. The p-values come from a model that only approximates biological reality, and thus it is likely such tiny p-value estimates are not accurate. $P=10E-115$ suggests that one would need to repeat the experiment $10E115$ times to come across the same value by chance. I would suggest choosing a more reasonable value, like 0.001 or 0.0001 and stating that p is less than that number.

The reviewer is right. The p-values to which the reviewer refers, on their own, could paint a misleading picture given that they are very low. We note that these values arise from the relatively large sample sizes, in conjunction with very clear differences, that occur because of the number of repetitions we made during bootstrapping (500). Therefore, we not only considered significant p-values but also took into account the effect sizes of the comparisons (Cohen's d). Only large effect sizes ($|d| > 0.8$) were deemed sufficient to draw conclusions in most of the comparisons of information transfer values (i.e. differences in the connectivity graphs). As the reviewer suggests, we could in principle state that the p-value is smaller than, for example, 10^{-3} or 10^{-5} . However, we did not do so because of Nature's policy to include, whenever possible, the most accurate value of this parameter. At the discretion of the reviewer and the Editor, we are fully willing to change the main text to accommodate the recommendations in this point.

5. In portions of the manuscript, the authors are careful in their use of the word "causal" and in other portions they are not. One cannot infer causality based on any of the approaches used here. Please insert modifiers (e.g. "potentially causal" in line 219) when using the word "causal" in this manuscript.

Thank you for this comment. Indeed, "true" causality cannot be asserted via any purely statistical/numerical approach. Therefore, throughout the manuscript we attempted to always be clear that, when referring to "causality", we did so within the transfer entropy framework. We revised the paper and tried to be this specific as often as possible in the text. In the specific sentence to which the reviewer refers in the comment, we specified that causal interactions in the circuit were quantified

within the TE framework. We apologize for the lack of clarity. The sentence has been rewritten as follows:

To study FAF-AC oscillatory dynamics during vocal production, we looked beyond phase correlations and examined causal interactions (within a transfer entropy framework) in the fronto-auditory circuit.

Minor:

1. Why were beta 2 and gamma 1 frequencies not examined?

We appreciate the reviewer’s concern. Beta 2 and gamma 1 frequencies were examined, but the results were not included in the main manuscript for two reasons. First, as mentioned in the main text, these frequencies only weakly predicted ensuing call type when considering LFP power (Fig. 1). Second, transfer entropy dynamics in these frequencies did not bring additional information when compared to other frequency bands. The combination of these two factors led us to focus on the frequency ranges included in the main text, for the sake of readability and to avoid cluttering figures with information that was of no consequence to the main conclusions of the study.

Below (**Fig. R4**), the reviewer can see the directed phase-transfer entropy matrices of LFP bands β_2 , γ_1 , and γ_2 . This information can be included as supplementary figure at the reviewers discretion.

Fig. R4. dPTE matrices for LFP bands β_2 , γ_1 , and γ_2 . Data in the figure are illustrated following the structure of Fig. S4. On the left, shaded in grey, a schematic representation of channel depths and cortical region associated to channel numbers in the dPTE matrices. On the right, mean dPTE matrices are shown for frequency bands β_2 , γ_1 , and γ_3 . Matrices are grouped according to

vocal conditions: related to echolocation and communication call production (top and bottom in each group, respectively), as well as to pre-vocal, post-vocal and no-voc periods (top, middle and bottom groups, respectively). Each matrix in the figure illustrates the average dPTE across 500 repetitions calculated using 50 trials corresponding to echolocation, communication (both pre- and post-vocal), or no-voc LFP segments. A cell (i, j) in a matrix shows the average dPTE value related to the information flow between channels i and j , which occurs in the $i \rightarrow j$ direction for dPTE values > 0.5 (red colours), and in the $j \rightarrow i$ direction for dPTE values < 0.5 (blue colours).

2. Line 415 mentions LFP changes “during” vocal production, but no assessments during vocal production were made

The reviewer is right. We changed “during” to “in periods associated to”.

3. Line 423 should be “such a position”

Thank you. Corrected.

4. Line 634 – what is meant by “posterior analyses”? Perhaps rewrite as “later analyses”?

Thank you for noticing this. To avoid misunderstandings, we changed the text to “subsequent analyses”.

5. Line 685 – I believe it should state that medium effect sizes should be ≥ 0.1 and < 0.4

Thank you for this. We corrected the statement for clarity and precision.

Reviewer #3 (Remarks to the Author):

Garcia-Rosales et al., Echolocation reverses information flow in a cortical vocalization network, NCOMMS-21-19228

*The experiments described in the manuscript investigate the patterns of information exchange between two cortical regions, the frontal auditory field (FAF) and the auditory cortex (AC) in awake and spontaneously vocalizing bats (*Carollia perspicillata*). Local field potentials (LFPs) were simultaneously recorded with multi-channel electrodes in both cortical regions. A number of elaborate statistical analysis methods is applied to reveal a predominantly top-down direction of information flow between the FAF and the AC. The direction of information flow was reversed after the emission of sonar calls. Generally, oscillations in the FAF and AC predicted the type of vocalization (sonar or non-sonar).*

The experiments are technically sound and the manuscript is well written. The results are new and interesting and should be of general interest for a broader scientific audience, but especially important for scientists interested in neural coordination of vocal communication and cortico-cortical network

interactions.

However, shortcomings of the manuscript result mainly from incongruities in the interpretation of the data and some technical/methodological concerns. The following comments might help to improve the manuscript and make the functional interpretation of the results more convincing.

Major comments

1) Line 29/Line 554ff. Of course, it is clear that during echolocation processing of the reflected echoes is of major importance. However, social communication can also have “imminent post-vocal consequences”, as conspecifics might respond to the calls (i.e. during mother –infant communication. The time-scale on which acoustic information as a response to an emitted call is received might be different in a social communication context, but information contained in a vocal response should be important. The authors should comment on this and explain why a reversal of information flow could not also be beneficial in vocal communication and not just only in the context of echolocation. Especially in the context of vocal learning bottom-up feedback control would also be important.

We agree with the reviewer and thank her/him for this comment. Communication calls can indeed carry post-vocal consequence of importance to the emitter, as is the case with mother-infant communication. However, as the reviewer notices, the exchange of acoustic information when, for example, two bats communicate with one another, might have different timescales than those seen during echolocation (e.g. the time between an echolocation call and the arrival of its echo in the bat's ear). In other words, a response from a conspecific will (almost certainly) arrive later than the echo of an echolocation call. This may be one of the main reasons why we see little evidence supporting the notion of predominant bottom-up information flow, *at least* during the first 500 ms after call onset. Moreover, the lack of a response from a conspecific could play an important role in describing the patterns we observe. Indeed, both echolocation and communication calls generate echoes from the environment after being produced, yet only echoes are relevant when animals are echolocating. When animals find themselves in “communication mode”, one could argue that an immediate response from a fellow bat could trigger significant bottom-up processing in the FAF-AC circuit. Thus, perhaps an audience effect is very important when considering what happens in the fronto-auditory network during communication production. In the original Discussion we had written as follows:

Since in this study animals vocalized without an audience (i.e. they were isolated in the recording chamber), further research could elucidate whether the presence of conspecifics increases bottom-up information transfer when vocalizing communication calls, as animals could expect a response.

Finally, we would like to stress that the directed phase transfer entropy (dPTE) metric quantifies the *predominant* directionality of information flow. In consequence, a *predominance* of top-down or bottom-up information transfer does not mean that information does not flow in the opposite direction as well. Therefore, we cannot discard the possibility that during post-vocal communication periods or others, information flows also in the bottom-up direction. However, this bottom-up transfer is not the predominant one and, as seen, putative changes in the relative strength of bottom-up and top-down information flows do not result in pre-vocal to post-vocal information flow reversals. The former is the

reason why throughout the manuscript's text we are careful to always make clear to the reader that the patterns shown are those of predominant directionality.

2) Line 96/Line 239/Line 504ff. The authors report causal interactions between the FAF and the AC during spontaneous activity. I was a bit surprised by that, because it sounds counter-intuitive to me. If the information-flow between FAF and AC is already directional during non-vocal periods, how can this information flow be considered to be specific in a vocal context?

We appreciate the reviewer's concern. Surely, our data reveal that information flows with preferred directionality between frontal and auditory cortices both during spontaneous activity and at periods related to vocal production. We do not claim that the preferred directionality of information flow is unique to vocal contexts. Rather, we demonstrate that the patterns of such directionality are contingent on several aspects of vocalization, such as time relative to vocal onset and type of call produced. For example, while information flow in the δ LFP band is predominantly top-down during spontaneous periods, bottom-up information transfer occurs predominantly in the bottom-up direction during post-vocal echolocation periods.

If causal interactions during spontaneous activity are due to attention (as suggested by the authors in the discussion), it is not clear why top-down attentional effects are not observed during emission of sonar calls (information flow is reversed from AC to FAF). Especially during an echolocation context, attention should be important.

The reviewer raises an important concern. Given the current dataset, we cannot support the claim that top-down attention is the reason why information flows predominantly in the FAF→AC direction, most strongly at lower LFP frequencies during spontaneous activity. This would first require to establish that the FAF is an attentional hub, i.e. a structure that orchestrates attentional processes in the bat's brain. To the best of our knowledge, the former remains speculative, even though we do know that the frontal cortex is a source of cognitive control in mammals. Were a top-down relationship based on attentional processes exist in the FAF-AC network, our functional, directed connectivity results show that a neural substrate based on oscillatory activity could be capitalized upon for the benefit of such FAF-AC interaction. Similar results have been obtained by studying the human brain, where the directionality of information flow in resting state networks is preferentially top-down for low LFP frequencies (θ), and bottom-up in higher frequencies such as β (Hillebrand et al., 2016). Notice how the spontaneous data we present in bats echoes the findings of Hillebrand et al.

The concept of attention is a contentious one. In the literature, attempts to describe attention go from, for example: "Attention is the important ability to flexibly control limited computational resources" (Lindsay, 2020), all the way to: "No one knows what attention is" (Hommel et al., 2019). However, there is some consensus in that there appear to be two main types of attention: top-down and bottom-up. The earlier refers to the internal guidance of attention; the latter speaks of attention driven from external factors (see (Katsuki and Constantinidis, 2014)). Thus, it is imaginable that echolocation could drive a form of bottom-up attention, where the salience of the information in echo-delay (which derives from

external factors such as object distance) is the main driving force. In that sense, bottom-up information flow from the auditory to the frontal cortex might also provide a neural substrate for such a process. This discussion continues to be speculative, particularly in our dataset where attention is not a variable that lends itself to be quantified with ease. Altogether, however, we do agree with the reviewer in that attention during echolocation should be important. We wonder what type of attention that would be, and how it could manifest in electrophysiological data. Future research should focus on these important questions. The section of the discussion to which the reviewer refers was corrected to the following (note the text in italics):

In terms of a cortical network for vocalization, the FAF and AC are engaged in rich information transfer dynamics with functional relationships to vocalization. Moreover, interactions extend to periods of vocal quiescence, when information flows top-down (FAF \rightarrow AC) in low (δ - α) and high (γ_2) frequencies. Low-frequency top-down influences from higher-order structures (like the FAF) modulate neuronal activity in sensory cortices according to cognitive variables such as attention, also during spontaneous activity (Fox et al., 2006; Hillebrand et al., 2016; Sang et al., 2017). *However, whether and how attentional processes exploit the nature of neural connections in the FAF-AC circuit remains thus far unknown.* Our data resonate with the hypothesis of top-down modulation of oscillatory activity in AC, and suggest a strict control of higher-order structures over sensory areas reflected in concurrent LFP activity across regions. Such strong top-down control is supported by the fact that FAF microstimulation enhances auditory cortical responses to sounds (Fig. 4).

In this context, the line of argumentation of the authors in the discussion (second last paragraph, line 535-561) is not coherent: The authors state that in an echolocation context, the bat auditory system must be “ready” to process echo information. I think, “readiness” in this context would just be another word for attention.

We apologize for this misunderstanding. By readiness we did not necessarily mean attention, we meant that it appears that the system primes itself for the processing of important echolocation-related information. In light of the arguments we give in response to the reviewer’s previous point, even when “readiness” is taken as attention (which was not our intention when writing these lines), a form of bottom-up attention could predominate during echolocation, which may take advantage of the predominant bottom-up directionality of information flow. The importance of arriving echoes for the animal during echolocation, which constitute external factors (see above), can hardly be overstated. However, in the manuscript and in this response we refrain from asserting that bottom-up information transfer is unequivocally a sign of bottom-up attentional processes.

Furthermore, if it would be true that, as the authors state in line 554 “communication calls information not aimed at the emitter itself”, the line of argumentation putting the FAF to AC directed information flow in the framework of attention would not be convincing. The authors should comment on this

We apologize for this confusion. We did not intend, in the original paper, to state that the preferential information transfer patterns reported were a cause (or a substrate) of attention (see quoted text above). Within the framework of attention, bottom-up information transfer during echolocation and not

during communication could result in bottom-up attention triggered by incoming echoes, and a lack of the same type of attentional processes in the other condition (i.e. communication). However, this is speculative and requires empirical support. We hope that the modifications in the new version of the manuscript help clarify that attentional processes are but one possible reason why top-down modulation in the FAF-AC circuit exists. We do not assume an attentional framework in the Discussion of our work.

3) The authors must show that differences in LFP oscillations (and thus predictability of call type from LFP) cannot simply be attributed to the large differences in the frequency content of sonar and non-sonar calls. Thus, the differences would be an epiphenomenon, not to be assigned to higher cognitive functions in vocal-motor control. Were there other communication calls emitted during experiments which had a spectral content more similar to echolocation calls, that can be used to check for the influence of the spectral content on LFPs? I know that this question was addressed in Weineck et al. 2020, Fig.5. However, I found the data presented there not very convincing, as the spectral structure of the LHF communication calls still was very much different from the echolocation call spectral content, especially in the low-frequency range.

Also, the influence of the acoustic stimulation by the calls should therefore be clarified (most important during the post-voc periods), especially as the authors state, that in an earlier paper (Garcia-Rosales et al. 2020) passive acoustic stimulation also had a profound effect on oscillations in the FAF and AC. In this respect, it might be important that the recordings in the AC were derived from the high-frequency region. However, the frequency-preference of neurons in the FAF in the region the electrode was inserted is not described in the manuscript. Furthermore, sonar and non-sonar call are typically emitted differently in phyllostomid bats: sonar calls are emitted thru the nostrils, typically associated with movements of the nose leaf, while non-sonar calls are often emitted trough the mouth. A possible differential influence of motor-activity on the LFPs should be discussed.

The reviewer raises interesting points. In the revised version, we addressed these possibly confounding issues in several steps (see also the response to Reviewers #2's third major comment). Indeed, there were communication calls emitted by the animals that had a spectral content similar to that of the echolocation pulses. Thus, we divided the group of communication vocalizations into two-subgroups: HF-communication (n = 155; calls with more than half their power in the 50-100 kHz range), and LF-communication (communication calls that did not fall in the HF-communication group; n = 579). The frequency content of HF- and LF-communication calls is shown in Fig. S3. The reviewer may note that differences between HF- and LF-communication calls in the current study were considerably larger than in (Weineck et al., 2020). Although the energy of HF-communication vocalizations was concentrated in high frequencies, pre-vocal LFP power differences between HF-communication and echolocation calls were still substantial, and occurred with the same patterns observed in the original dataset (shown in Fig. 1). Importantly, spectral differences in the LFP between HF- and LF-communication calls were negligible, with models performing poorly when attempting to predict ensuing call type (new Fig. S3). These results indicate that the frequency content of the produced vocalizations did not account for the conclusions drawn from the data shown in Fig. 1.

We also explored how the influence of acoustic inputs shapes the patterns of information exchange in the FAF-AC circuit. To that effect, we first quantified frequency tuning curves of the LFPs across

penetrations (recording sites in AC are depicted in Fig. S1; frequency tuning curves can be seen in Fig. S7a). The outcomes of these analyses are as follows (subsection “Passive listening of high- or low-frequency natural sounds does not explain information flow patterns of active vocalization” in the supplementary materials):

In a first step, we quantified frequency tuning in the AC (and FAF, see Methods) and observed the tuning of recorded LFPs did not favour the frequency range of echolocation calls (i.e. > 60 kHz), as it peaked at 20-40 kHz for most recording sites (Fig. S7 shows LFP frequency tuning curves measured with 75 dB SPL, 10 ms tones, across penetrations). Thus, LFP responses in the AC, at least based on frequency tuning alone, would not elicit on average a stronger response and therefore a stronger bottom-up transfer towards the FAF.

Further, we examined how the spectral content of naturalistic sounds affected the information flow patterns in the network. We recorded responses to two acoustic stimuli to which the animals listened passively: a high-frequency frequency-modulated sound (“HF-FM”, 2 ms long, downward frequency sweep from 80-50 kHz), and one distress syllable (“distress”, 3.8 ms long; both depicted in Fig. S7b). Analyses were made on LFP segments spanning 0-500 ms after sound presentation. The passive listening of these acoustic stimuli failed to reproduce the patterns observed during post-vocal periods. Neither stimulation with the HF-FM nor with the distress stimuli resulted in a predominance of bottom-up information transfer, as observed during post-vocal periods of active echolocation utterance. Predominant information flow patterns were in fact very similar in response to both acoustic inputs (HF-FM and distress), and also similar to those observed during spontaneous activity (Figs. S7-S9). The former suggests that passive listening fails to alter significantly the connectivity in the FAF-AC circuit, in particular for low LFP frequencies, in line with our previous report in (García-Rosales et al., 2020), where interareal coherence was quantified. Jointly, our data suggest that, in order to alter significantly the FAF-AC functional connection, more than passive listening is required. Such is the case when considering situations of active vocalization, especially when animals produce echolocation calls. These data indicate that acoustic feedback alone does not fully explain the connectivity dynamics associated to vocalization.

Finally, we explored the possibility that the frequency content of active vocalizations would account for our original results. When considering the dynamics of information flow in the FAF-AC network, there was some bottom-up information transfer for post-vocal periods related to HF-communication calls (Fig. S10). However, the former occurred in the θ and α bands of the LFP, and not in δ as was the case with echolocation production. Changes in α for the post-vocal HF-communication case did not constitute an information reversal when compared to pre-vocal periods (Fig. S10b); the change in θ , albeit consistent with an information reversal, resulted in values of bottom-up information transfer no different to those observed for post-vocal echolocation periods in the same LFP frequencies. Thus, there were changes in information transfer dynamics that were unique to echolocation production, especially taking into account the reversal of information in δ frequencies. Importantly, these data corroborate that the differences in the frequency content of the produced calls fail to fully explain our original observations.

These results are now a part of the revised manuscript: new subsections “Passive listening of high- or low-frequency natural sounds does not explain information flow patterns of active vocalization” and “Information transfer patterns related to HF-communication calling differ to those associated with

echolocation” in the Supplementary Materials. The results are integrated into the new version of the Discussion.

As the reviewer points out, influences of motor activity related to echolocation or communication call production may affect differentially the LFPs. Although phyllostomid bats were previously thought to echolocate only through their nostrils (notably, for example, bats belonging to the *Phyllostomus* genus), recent evidence suggests that *C. perspicillata* also emits sonar pulses through their mouth (Gessinger et al., 2021). This does not mean that echolocation through the nostril does not occur in *Carollinae* bats, whose emission through the mouth may help narrowing the sonar beam in the vertical direction. However, even when echolocation and communication calls are generated in the larynx and emitted through the mouth (and nostrils, for echolocation), the motor plan for such emissions may differ. Our own data indicate that the frequency content of the emitted signals does not account for pre-vocal LFP activity (see above). Rather, specific differences in the motor program for different vocal types could be associated with the particular requirements of echolocation and communication call production that go beyond the frequency of the emission. This could include nose-leaf movements, unique larynx dynamics for producing short downwards frequency modulated sonar pulses, among other factors. In consequence, we added a paragraph to the Discussion where the possible contribution of motor-related activity in the pre-vocal LFPs is clarified. The following was added to the main text (Discussion):

Differences in spectral patterns cannot be solely explained by the distinct frequency content of echolocation and communication calls (Fig. S3). However, considering that orofacial movement in primates (Miller et al., 2015; Roy et al., 2016) and vocalization-specific movements in bats (Eiermann and Esser, 2000) are associated to neural activity in frontal areas, distinct pre-vocal motor related activity for echolocation or communication calling is a plausible explanation for our results. Microstimulation of *C. perspicillata*'s FAF can result in motor effects such as pinna and nose-leaf movements, as well as vocalizations (including echolocation-like calls; (Eiermann and Esser, 2000)). These movements also occur naturally before spontaneous vocalization (Eiermann and Esser, 2000), suggesting that the FAF may be involved in the motor aspect of vocal production. Nevertheless, vocalization-specific neural populations in primates coexist with those related to orofacial movements (Roy et al., 2017). Therefore, the vocal-motor explanation does not necessarily entail that the FAF fails to participate in other forms of vocal preparation beyond the orchestration of motor programs.

4) To evaluate the recording position within the cortical layers, it would be helpful to indicate layer borders in Figure 1d for FAF and AC separately. It is important to consider that the cortical thickness in the AC and FAF might be different. This is especially important as data from different electrode channels were pooled for some of the analyses described in the manuscript (line 229).

Thank you for this suggestion. Based on histological data (see the revised Fig. S1), we demarcated layer borders in Fig. 1d in the AC to improve transparency. Layer borders in the FAF are not easily definable, and as such we decided not to mark them (note the coronal slice in Fig. S1a). We agree that the layer architecture in both structures is important under certain circumstances when electrode channels are

grouped. In this study, channels were grouped taking into account the laminar architecture of the AC, which can be readily defined anatomically. For example, channels in the superficial group consisted of electrodes situated in layer I-II; channels in the middle groups spanned from mid layer III, down to mid layer V; channels in the “deep” group consisted mostly of depths spanning the bottom of layer V, and layer VI. In the FAF, layers are less anatomically and functionally demarcated, making it harder to establish a proper grouping based on laminar architecture.

We argued in response to one of Reviewer #1’s comments, that the channel grouping would not considerably affect our results. We base this reasoning in the fact that dPTE matrices were rather poorly spatially resolved (see Fig. S4), one of the main reasons for this being that LFPs across layers are likely well-correlated tens to hundreds of micrometres away from any one recording site (see, for example, the consequences of referencing to the top electrode in FAF below). Thus, the patterns observed in the main figures of the manuscript are readily visible also when examining the dPTE matrices in Supplementary Figure 4. These matrices depict connectivity in a channel-per-channel basis. Our grouping, while improving the data visualization, does not compromise the conclusions that can be drawn from the connectivity analyses.

In the methods section, (line 590) it is stated that “In the AC, recordings were made mostly in the high-frequency field”. What does “mostly” mean? This should be quantified. How many electrode penetrations were located in other regions? How were recording sites verified after the experiments had been finished? How large was the variation of positions of recording site within the AC and the FAF in the three bats?

Thanks for bringing this up. We decided to respond to the reviewer’s comment in visual manner. Therefore, we made a new Supplementary Figure 1, where we show all penetration sites in AC. These were not corroborated histologically after experiments were finished, but penetration areas were marked according to visual landmarks on the surface of the *C. perspicillata*’s brain (see (Esser and Eiermann, 1999; Hagemann et al., 2010)). We intended to record from the high frequency fields, but as the reviewer may see in Supplementary Figure 1 and **Fig. R3** (kindly see response to Reviewer #2), we cannot discard the presence of sites located close to the border of the primary auditory cortex and the anterior auditory field.

5) In a previous paper using similar methods (Weineck et al., 2020), in the FAF high-frequency power only predicted the vocal output (Weineck et al., 2020, Fig 6). In the present manuscript low frequency power in the FAF predicted vocal output, too (line 146, Fig1g). As the methods to record LFPs seem to be basically the same in both manuscripts, this difference should be commented to support reliability of analyses and methods. If activity in different frequency bands would not be reproducible within different data sets, then much of the detailed analyses presented in the current manuscript would not be very meaningful.

We agree with the reviewer’s comment. All things equal, the patterns in FAF shown in (Weineck et al., 2020) should be similar to those reported in the current study. However, our approach diverges from that of Weineck et al. in a few points. First, the analyses of whether pre-vocal LFP power predicted ensuing vocalization type was done in the 2020 study by means of support vector machine classifiers; in

the current study, we used generalized linear models to evaluate this question. Furthermore, LFP power in the current manuscript was calculated using PSD of 500 ms windows, whereas the power in Weineck et al. was quantified from the spectrograms where smaller time windows are used for spectral analyses. Still with those differences in mind, low-frequency power should yield similar predictive power in both studies, which was not the case even when we re-analysed the Weineck et al. data using GLMs and full 500 ms LFP windows.

There is however one key difference between the two studies. In Weineck et al., the electrode at the top of the shank inserted in the FAF was used as a reference, which was not the case for the current manuscript. Essentially, this means that signals in the top channel were subtracted from all other channels in the shank. Now, a close examination of the LFP signals in FAF (note Fig. 1d) reveals that low-frequency LFPs are well-correlated across channels, indicating that referencing channels to the top electrode could result in a considerable loss of effects seen in the low-frequency bands of the oscillations. We reasoned this to be the cause for the lack of predictive power in the low-frequency bands of the LFP, when considering the data in Weineck et al. (2020).

To test the above hypothesis, we simulated Weineck et al.'s referencing scheme by subtracting the signal recorded in the top FAF electrode from all other channels located in the frontal cortex. The outcomes are depicted in **Fig. R5**. It is immediately noticeable that, even by doing a simple virtual referencing with the top FAF channel, the predictive power of pre-vocal low-frequency LFPs (e.g. δ - α) diminishes considerably. Pre-vocal LFP power in the γ_2 range (60-120 Hz), however, still predicts whether animals vocalize an echolocation or a communication call. We therefore believe that the main difference between the current work and that of Weineck et al. lies in the referencing model used for electrophysiological recordings.

Fig. R5. Off-line referencing of LFP signals in the FAF to the top electrode abolishes ability to predict ensuing call type using low-frequency pre-vocal spectral power. (a) Example LFP traces in FAF around the production of an echolocation and a

communication call. These traces are from the same recording the same depicted in Fig. 1 (main manuscript), but signals from channels 2-16 in FAF were referenced to the top-most electrode. **(b)** Percentage pre-vocal power change across representative LFP bands (δ - α , 1-12 Hz; and γ_2 , 60-120 Hz), relative to a no-voc baseline, across cortical depths in FAF. Values related to echolocation utterances ($n = 138$) are depicted in blue; those related to communication utterances ($n = 734$) are depicted in orange. Data shown as mean \pm sem. **(c)** Effect size (R^2m) of GLMs considering all frequency bands and channels in FAF. Effect sizes were considered small when $R^2m < 0.1$, and medium for $R^2m \geq 0.1$. Effect size values from non-significant models were set to 0. Note the differences between the data shown in this figure, and those shown in Fig. 1 of the study.

Minor comments

Line 52/451. Suppression of neural response to self-produced sounds should not occur during echolocation, as the emitted call is an important marker for echo-delay computation. Although the data by Li et al., 2020 might indicate that corollary discharges might also enhance processing of self-generated sounds, it is not clear if the time-scale on which the dynamics of pre-vocal power described in the manuscript occur would fit the requirement for fast processing of echolocation sequences. Please, comment on this.

The effects of suppression in the bat's AC to self-produced echolocation pulses remain an interesting question. As the reviewer notices, response suppression in cortex does not directly translate to a deterioration of stimulus processing, given that cortical suppression to self-generated vocalizations seems to increase the sensitivity of neurons to vocal feedback under certain circumstances (Eliades and Wang, 2008; Li et al., 2020). Evidence from *C. perspicillata* demonstrates that forward suppression (i.e. not to self-produced sounds, but to a sequence of acoustic inputs at a high rate) at the level of AC does not result in an unresponsive cortex, incapable of dealing with echo-delay information. As reported by Beetz and colleagues, forward suppression ultimately results in sharper delay tuning maps in cortex, thus improving the echolocation system's spatial resolution upon reception (Beetz et al., 2016b). We further note that, in the bat's cortex, suppression to self-produced echolocation sounds would be paired with pulse-echo pair facilitation, depending on the timescales of suppression and facilitation. Pulse-echo pair facilitation refers to the phenomenon whereby neurons in the bat auditory system respond more strongly to pulse-echo paired signals (i.e. a pulse followed by an echo) separated in time by a delay consistent with a neuron's delay tuning (Bartenstein et al., 2014; Hechavarria and Kossl, 2014; Wenstrup and Portfors, 2011). The manner in which suppression to self-produced sounds and facilitation to echo-pulse pairs interact in the bat auditory cortex remains unknown. The current dataset is also not ideal to resolve how pre-vocal power dynamics in the LFPs interact with possible post-vocal suppression / facilitation, in particular regarding the timescales of pre-vocal and post-vocal patterns. Unfortunately, we cannot rigorously quantify the time course and magnitude of suppression to self-produced sounds in *C. perspicillata*'s cortex, as well as its effect in acoustic processing. Future research should be aimed at answering these questions.

Line 112 and Figure 1b. Although the difference was not significant, sonar calls should be generally shorter, compared to the non-sonar calls, as indicated in figure 1a. However, the distribution of sonar-call lengths seems to be double-peaked. Please check.

The reviewer is right. The distribution of echolocation call lengths is double-peaked. However, beyond call length, we do not see major differences in the spectrotemporal properties of the echolocation calls recorded, as the reviewer might note from inspecting **Fig. R1** (responses to Reviewer 1).

As far as I know, the Kolmogorov-Smirnov-Test assumes a normal distribution of data whereas the Wilcoxon Rank Sum Test is a non-parametrical test not assuming normal distribution of data. I think both tests should not be used in combination.

We appreciate the reviewer's comment. Both the Wilcoxon rank sum test and the two-sample Kolmogorov-Smirnov-Test used in this study are non-parametric tests, so they do not assume a normal distribution of the data. In the statistical analyses, we used Wilcoxon rank sum tests for determining whether the medians of two populations differed significantly. However, at one point we used the two-sample Kolmogorov-Smirnov test in order to determine whether call length distributions were significantly different between echolocation and communication, albeit without the medians of each necessarily differing from one another.

Line 120ff, Fig. 1d: To illustrate base line activity, it would be helpful to also show LFP traces which are totally unrelated to vocal activity (say, 1-2 seconds before/after vocalizations, or show the time period used to determine no-voc activity in Fig.1e).

The reviewer is right. For analysis, we chose baseline segments pseudo-randomly, each of 500 ms in length. We took care that the number of segments chosen per penetration matched exactly the number of vocalizations associated to that site, with the objective of reducing possible biases (although this measure is probably not strictly necessary). Because of the pseudo-random selection of segments, we did not show baseline LFP traces in the original manuscript, since they are not directly related to any one vocalization event, and they change for each iteration of the algorithm.

Line 120, Fig. 1d. A scale bar for the y-axis should be added. It must be clear that all traces are shown on the same scale.

Thank you. Indeed, traces are shown in the same scale. However, traces were normalized per structure for visualization. The former means that field potentials shown from FAF were related in scale with one another, with the same being true for those shown from the AC. They were, however, not in scale across regions. We hold that this brings no misleading visual effects, as comparisons of power or amplitude between FAF and AC were not a subject of the current study. Rather, only within-structure comparisons were made for LFP power, but rather across vocalization conditions. That amplitudes were normalized has been highlighted in the figure and its caption.

Line 154. As the FAF receives sensory input via two different pathways (the classical ascending auditory pathway via the AC, but also via the extra-lemniscal pathway by-passing the AC), it should not be exclusively labelled “association cortex”.

We appreciate this remark. It has been corrected in the text.

Line 414/435. While the authors state that “LFPs in FAF and AC are causally related (within a TE framework) during vocal production...” they also claim that there is no causal role of LFPs for initiation or planning of sonar or non-sonar calls. This sounds contradictory. If LFPs in FAF and AC are causally related during vocal production, and LFPs predict vocal output, why is there no causal role of LFPs for initiation or planning of sonar or non-sonar calls? The authors should comment on this in more detail. The same hold true for line 501: “...causal interactions (within a TE framework)...with functional relationship to vocalization”. Here again, a causal relation to vocalization is at least implied. This statement should be avoided if there is no causal role of LFPs for initiation or planning of sonar or non-sonar calls, as the authors claim in line 435.

The reviewer raises an important point here. We believe that the issue stems from lack of clarity in the original text. We do not claim that there is no causal role of LFPs for initiation or planning of vocalizations in FAF. This, at least given the current dataset, would not have been demonstrable. Similarly, our data, beyond correlational evidence, do not allow us to assert that changes in oscillatory activity are indeed causal to either the planning or the initiation of a vocal utterance. These changes do reflect in a predictive fashion whether a call is to occur (by means of increased power), and the type of call the animal produces (by means of the strength of such increase), but a causal link cannot be rigorously established. The pre-vocal spectral patterns that we observe (and in fact replicate, see (Weineck et al., 2020)) strengthen the argument that the frontal-auditory field is a cornerstone region for vocalization in bats.

Similarly, from the fact that oscillations in the FAF-AC network are causally related with strong top-down influence given the transfer entropy framework, does not necessarily follow that these oscillations are causal to the initiation of a call. Thus, a scenario can be imagined wherein there could be statistical causality considering LFPs in the FAF-AC circuit, but no causality in the oscillation-vocalization sense. Again, we also do not assert the latter, which is discussed here only to support our argument in an extreme case. Rigorously speaking, the data and analyses suffice to determine the preferred directions of information flow between frontal and auditory cortices, but not the causality with the act or the planning of vocal production.

In order to establish whether oscillations in the FAF are cause a vocal output, carefully designed experiments must be conducted where the oscillatory activity is manipulated and the vocal production of the animals concurrently monitored. Changes in such oscillations should lead to a vocalization: a case which implies that such changes are sufficient to explain vocal production on their own. A first question could be whether changes in pre-vocal spectral power are necessary and sufficient for vocalization, or whether they are indeed sufficient, or perhaps necessary but not sufficient. Then the roles of other structures in frontal or motor cortices should be weighted in (e.g. preSMA, or M2). Another question

that would rise is whether the oscillations cause the vocal output, or if manipulating them modulates the response of their underlying neuronal substrate, and therefore these changes in the neurons' firing patterns, orchestrated by the LFPs, are the ones responsible for initiating a call. At this stage, our data do not allow to address these important questions. Therefore, in the Discussion, we write as follows:

Consistent with previous reports (Gavrilov et al., 2017; Gilmartin et al., 2014; Hage and Nieder, 2013; Helfrich and Knight, 2016; Kingyon et al., 2015; Weineck et al., 2020), we show that neural activity in the frontal cortex predicts vocal outputs. Taken together, the data from this and previous work suggest that oscillations in frontal regions may be instrumental for vocal production. From our perspective, the above is further supported by call-type specific, pre-vocal LFP spectral dynamics and information transfer patterns in the FAF-AC network. The relationship between oscillations and vocal production remains, nevertheless, correlational: our results do not allow to rigorously assert a causal role of LFPs for the initiation or planning of vocal outputs.

Line 440. "...as previously reported..." Does this refer to Tsunada and Eliades, 2020? Please clarify.

Indeed. We did the necessary changes in the main text.

Fig. 4,5,6. The figures are very difficult to read as they look very much cluttered, especially the line plots in top rows. I would recommend simplifying the figures. Maybe only show the most important information.

Thank you for this suggestion. Much of the data shown in these figures was presented in a cluttered manner, and did not necessarily contribute to the main message of the manuscript. Following the reviewer's suggestion, only the most important information is now shown in the main text. The data in figure 4-6 was moved to the Supplementary Materials (Figs. S5, S6); quantifications and comparisons associated with these data are detailed in the Supplementary Text.

Line 632/638: "...a preliminary classification ...was done based on each call's peak frequency...". "Finally, vocalizations were examined via visual inspection to validate their classification". Were additional parameters other than peak frequency used for classification?

As the reviewer mentions, in a first stage calls were assigned to "echolocation" or "communication" categories solely based on their frequency content (i.e. calls with highest energy content in the 50-100 kHz band were labelled as echolocation). A manual curation followed, where we visually verified, among other things, the classification. Parameters in this case were still reliant on the spectrotemporal structure of each vocalization, including call length, whether the call was frequency modulated (a signature of echolocation calls in *C. perspicillata*; (Kossel et al., 2012; Thies et al., 1998)) and how, and the distribution of power along the frequency axis.

Were the non-sonar-calls a homogenous group, or would a further classification into sub-classes have been possible?

Non-sonar calls were not a homogenous group. It was possible to classify them further, at least considering only the acoustic properties of the calls. Firstly, we divided communication calls into the two above-mentioned groups based on their frequency content (i.e. LF- and HF-communication). Analyses were performed on each of these groups, whose results have been described above. In addition, we performed a classification based on the methodology described in (Hechavarria et al., 2016). These results are summarized in **Fig. R1**, to which the reviewer is kindly referred (see responses to Reviewer 1's comments). We observed that non-sonar calls could be classified into multiple templates, but that the 12 most populated templates accounted for ~80% of the calls (586/734; see **Fig. 1Rc**). In **Fig. 1Rd**, we show the 8 most numerous templates (502/734 calls, ~68%). As is evident in the figure, templates of non-sonar vocalizations indicate that they did not constitute a fully homogenous group (i.e. only one type of call), and that differences among them were to be seen when considering their spectrotemporal dynamics.

Line 1060, Legend to Figure 1: the sub-figure labels are not correct (should be e,f,g not e,e,f)

Thank you. This has been corrected.

References

- Arnsten, A.F.T., Raskind, M.A., Taylor, F.B., and Connor, D.F. (2015). The effects of stress exposure on prefrontal cortex: Translating basic research into successful treatments for post-traumatic stress disorder. *Neurobiol Stress* 1, 89-99.
- Atencio, C.A., Shih, J.Y., Schreiner, C.E., and Cheung, S.W. (2014). Primary auditory cortical responses to electrical stimulation of the thalamus. *J Neurophysiol* 111, 1077-1087.
- Bartenstein, S.K., Gerstenberg, N., Vanderelst, D., Peremans, H., and Firzlaff, U. (2014). Echo-acoustic flow dynamically modifies the cortical map of target range in bats. *Nat Commun* 5, 4668.
- Beetz, M.J., Hechavarria, J.C., and Kossl, M. (2016a). Cortical neurons of bats respond best to echoes from nearest targets when listening to natural biosonar multi-echo streams. *Sci Rep* 6, 35991.
- Beetz, M.J., Hechavarria, J.C., and Kossl, M. (2016b). Temporal tuning in the bat auditory cortex is sharper when studied with natural echolocation sequences. *Sci Rep* 6, 29102.
- Brinklov, S., Jakobsen, L., Ratcliffe, J.M., Kalko, E.K., and Surlykke, A. (2011). Echolocation call intensity and directionality in flying short-tailed fruit bats, *Carollia perspicillata* (Phyllostomidae). *J Acoust Soc Am* 129, 427-435.
- Eiermann, A., and Esser, K.H. (2000). Auditory responses from the frontal cortex in the short-tailed fruit bat *Carollia perspicillata*. *Neuroreport* 11, 421-425.
- Eliades, S.J., and Wang, X. (2008). Neural substrates of vocalization feedback monitoring in primate auditory cortex. *Nature* 453, 1102-1106.
- Esser, K.H., and Eiermann, A. (1999). Tonotopic organization and parcellation of auditory cortex in the FM-bat *Carollia perspicillata*. *Eur J Neurosci* 11, 3669-3682.

- García-Rosales, F., López-Jury, L., González-Palomares, E., Cabral-Calderín, Y., and Hechavarría, J.C. (2020). Fronto-Temporal Coupling Dynamics During Spontaneous Activity and Auditory Processing in the Bat *Carollia perspicillata*. *Frontiers in Systems Neuroscience* 14.
- Gavrilov, N., Hage, S.R., and Nieder, A. (2017). Functional Specialization of the Primate Frontal Lobe during Cognitive Control of Vocalizations. *Cell Rep* 21, 2393-2406.
- Gessinger, G., Page, R., Wilfert, L., Surlykke, A., Brinkløv, S., and Tschapka, M. (2021). Phylogenetic Patterns in Mouth Posture and Echolocation Emission Behavior of Phyllostomid Bats. *Front Ecol Evol* 9.
- Gilmartin, M.R., Balderston, N.L., and Helmstetter, F.J. (2014). Prefrontal cortical regulation of fear learning. *Trends Neurosci* 37, 455-464.
- Hage, S.R., and Nieder, A. (2013). Single neurons in monkey prefrontal cortex encode volitional initiation of vocalizations. *Nat Commun* 4, 2409.
- Hagemann, C., Esser, K.H., and Kossel, M. (2010). Chronotopically organized target-distance map in the auditory cortex of the short-tailed fruit bat. *J Neurophysiol* 103, 322-333.
- Hechavarría, J.C., Beetz, M.J., Macías, S., and Kossel, M. (2016). Distress vocalization sequences broadcasted by bats carry redundant information. *J Comp Physiol A Neuroethol Sens Neural Behav Physiol* 202, 503-515.
- Hechavarría, J.C., and Kossel, M. (2014). Footprints of inhibition in the response of cortical delay-tuned neurons of bats. *J Neurophysiol* 111, 1703-1716.
- Helfrich, R.F., and Knight, R.T. (2016). Oscillatory Dynamics of Prefrontal Cognitive Control. *Trends Cogn Sci* 20, 916-930.
- Hillebrand, A., Tewarie, P., van Dellen, E., Yu, M., Carbo, E.W., Douw, L., Gouw, A.A., van Straaten, E.C., and Stam, C.J. (2016). Direction of information flow in large-scale resting-state networks is frequency-dependent. *Proc Natl Acad Sci U S A* 113, 3867-3872.
- Hommel, B., Chapman, C.S., Cisek, P., Neyedli, H.F., Song, J.H., and Welsh, T.N. (2019). No one knows what attention is. *Atten Percept Psycho* 81, 2288-2303.
- Katsuki, F., and Constantinidis, C. (2014). Bottom-Up and Top-Down Attention: Different Processes and Overlapping Neural Systems. *Neuroscientist* 20, 509-521.
- Kingyon, J., Behroozmand, R., Kelley, R., Oya, H., Kawasaki, H., Narayanan, N.S., and Greenlee, J.D. (2015). High-gamma band fronto-temporal coherence as a measure of functional connectivity in speech motor control. *Neuroscience* 305, 15-25.
- Kossel, M., Voss, C., Mora, E.C., Macías, S., Foeller, E., and Vater, M. (2012). Auditory cortex of newborn bats is prewired for echolocation. *Nature Communications* 3.
- Kothari, N.B., Wohlgenuth, M.J., and Moss, C.F. (2018). Dynamic representation of 3D auditory space in the midbrain of the free-flying echolocating bat. *Elife* 7.
- Li, S., Zhu, H., and Tian, X. (2020). Corollary Discharge Versus Efference Copy: Distinct Neural Signals in Speech Preparation Differentially Modulate Auditory Responses. *Cereb Cortex* 30, 5806-5820.
- Lindsay, G.W. (2020). Attention in Psychology, Neuroscience, and Machine Learning. *Front Comput Neurosci* 14, 29.

Miller, C.T., Thomas, A.W., Nummela, S.U., and de la Mothe, L.A. (2015). Responses of primate frontal cortex neurons during natural vocal communication. *J Neurophysiol* *114*, 1158-1171.

Okonogi, T., and Sasaki, T. (2021). Theta-Range Oscillations in Stress-Induced Mental Disorders as an Oscillotherapeutic Target. *Front Behav Neurosci* *15*.

Roy, A., Svensson, F.P., Mazeh, A., and Kocsis, B. (2017). Prefrontal-hippocampal coupling by theta rhythm and by 2-5 Hz oscillation in the delta band: The role of the nucleus reuniens of the thalamus. *Brain Struct Funct* *222*, 2819-2830.

Roy, S., Zhao, L., and Wang, X. (2016). Distinct Neural Activities in Premotor Cortex during Natural Vocal Behaviors in a New World Primate, the Common Marmoset (*Callithrix jacchus*). *J Neurosci* *36*, 12168-12179.

Takahashi, S., Muramatsu, S., Nishikawa, J., Satoh, K., Murakami, S., and Tateno, T. (2019). Laminar responses in the auditory cortex using a multielectrode array substrate for simultaneous stimulation and recording. *IEEE T Electr Electr* *14*, 303-311.

Thies, W., Kalko, E.K.V., and Schnitzler, H.U. (1998). The roles of echolocation and olfaction in two Neotropical fruit-eating bats, *Carollia perspicillata* and *C. castanea*, feeding on Piper. *Behav Ecol Sociobiol* *42*, 397-409.

Weineck, K., Garcia-Rosales, F., and Hechavarria, J.C. (2020). Neural oscillations in the fronto-striatal network predict vocal output in bats. *PLoS Biol* *18*, e3000658.

Wenstrup, J.J., and Portfors, C.V. (2011). Neural processing of target distance by echolocating bats: functional roles of the auditory midbrain. *Neurosci Biobehav Rev* *35*, 2073-2083.

Yazdan-Shahmorad, A., Lehmkuhle, M.J., Gage, G.J., Marzullo, T.C., Parikh, H., Miriani, R.M., and Kipke, D.R. (2011). Estimation of electrode location in a rat motor cortex by laminar analysis of electrophysiology and intracortical electrical stimulation. *J Neural Eng* *8*, 046018.

Zhang, W., and Yartsev, M.M. (2019). Correlated Neural Activity across the Brains of Socially Interacting Bats. *Cell* *178*, 413-428 e422.

Reviewers' comments:

Reviewer #1 (Remarks to the Author):

In this study, the authors use local field potential recordings to demonstrate that the timing and spatial pattern of oscillations in the FAF-AC network of bats predict the purpose of vocalization, i.e. either echolocation or communication. Interestingly, the flow of information shifts from the auditory cortex to the frontal cortex only when the animal is producing echolocation sounds. These are intriguing results and the data analysis methods open the door to future investigations of vocal control and auditory processing reflected within population activity in the cortex of echolocating bats. The findings are also important for understanding general principles of information processing within the auditory cortex. In this regard, publication of these results will hopefully facilitate further studies on this phenomenon.

The work reported here is original and builds on previously published literature. There are no apparent flaws in the data analysis interpretation and conclusions and the methodology is sound. Though the original version of the manuscript lacked some explanation and details, the authors have done an excellent job of addressing reviewer comments in detail and revising the manuscript, including several figures, based upon new data analysis as well as experiments that make the findings more reliable and explain potential discrepancies with the previously published studies.

I do not have any additional major comments, but would like to caution the authors to be consistent in their use of terminology and carefully check the manuscript for ease of readability before final submission.

A few minor points that can be corrected are noticed noted below.

The title reads better when modified to "Echolocation-related reversal of information flow in a cortical vocalization network"

The reason for this change is that echolocation is a behavior involving pulse-echo pair evaluation and the behavior itself does not reverse information flow; rather, the emission of echolocation pulses reverses information flow. Therefore, "echolocation-related reversal" is a more accurate description of the observed phenomenon.

In the abstract, at line 28, please insert "... information transfer patterns during the post – vocal period specific to echolocation pulse emission.

Line 30" "... sounds in the auditory cortex."

Line 391: do you mean qualitatively similar variation (rather than "differences") within echolocation and communication" ?

Line 393: Change to: "The act of emitting echolocation pulses therefore triggers unique patterns of information-flow reversal in the fronto-auditory network of ...".

Line 395: Correct to: "Our data indicate that the functional connectivity in this ..."

Line 396: Change to: "... by the behavioral intent of vocalization."

Line 404: Change to: "... interactions in the bat's FAF-AC network are associated with auditory processing for echolocation."

Line 498: "... an HF-communication call was an utterance with ..." – delete "communication" (redundant)

Reviewer #2 (Remarks to the Author):

I have read the revised version of this manuscript, and overall it is improved. The two issues that I had been most concerned about in the first version were the notion of causality of FAF-AC interactions and whether the different frequencies of sound stimulation relative to the locations of the recordings could account for the differences seen between echolocating calls and social calls. The authors have adequately addressed the latter concern.

With respect to causality, I commend the authors for doing a difficult stimulation experiment, that is now in Figure 4. I am somewhat confused by the analysis and results. The main thrust of the paper is to quantify FAF-AC interactions and the authors have described a processing loop whereby FAF “prepares” AC prior to the stimulus and then AC “updates” FAF after the stimulus. Stimulation of FAF would be expected to scramble the message. I would expect that the metrics of FAF-AC connectivity that are described throughout the paper would have been disrupted by this manipulation. Did the authors examine these metrics? How do the authors interpret the increase in response strength? Does this imply that FAF tonically inhibits AC responses? The authors should also describe how they came to this particular electrical stimulation paradigm – do they expect that FAF neurons would be stimulated or suppressed by it? Can the authors speculate as to the long latency of the impact on the AC? Finally, the image itself is somewhat hard to read. I think it would be useful to color-code the latency in a way that the reader can look at the image and know whether there is a systematic effect of timing on the response metric. For example, looking at 4F, the reader needs to go back and forth from the legend to the figure to understand the relationship between time and response magnitude. Better yet, the authors should plot time on the x-axis and plot a suitable metric (e.g., response strength or connectivity metric) on the y-axis.

Reviewer #3 (Remarks to the Author):

Garcia-Rosales et al., Echolocation reverses information flow in a cortical vocalization network, NCOMMS-21-19228A-Z

Generally, the manuscript has improved a lot. I am satisfied with the authors' responses to most of my comments.

However, I am still concerned about two points:

1) I am still not fully convinced that the spectral differences between communication and echolocation calls are not a main source of differences in the spectral dynamics of pre-vocal LFPs. Although the HF-communication calls shown in the revised manuscript lack the strong low frequency component of the LF- communication calls, they are still largely different from the echolocation calls. I think the spectral center of gravity would be approximately 20 kHz lower. Furthermore the echolocation calls have most of their energy concentrated in a relatively narrow frequency band (spectral peak shown in S3a) while the HF-Communication calls are more or less broadband without prominent spectral peak. I appreciate, that the authors consider this problem a bit more in the revised manuscript by stating, that “pre-vocal spectral differences are not fully accounted for by differences in the spectral content of echolocation and communication calls” (line 146). And I am fully aware that it is very challenging to experimentally determine the influence of the spectral content of vocalizations used in different behavioral contexts precisely.

Therefore, the authors should at least explicitly state in the manuscript, that an influence of differences in the spectral content of echolocation and communication calls on LFPs cannot definitely be excluded.

2) I am bit puzzled about the response of the authors to my comment (5) concerning the differences of LFP frequency band power in their different publications. Actually, their line of argumentation in the answer to my comment strengthens my initial concerns. Obviously, technical details of the LFP

recording procedures have profound influence on results, and thus on the interpretation of data in a functional context. How is the referencing scheme for the recordings in the current paper? I could not find information on this in the methods section. Furthermore, which of the two referencing schemes (top electrode as reference vs. other scheme) is the most appropriate one, and why? It is still difficult for me to accept the complex functional interpretations made in the current manuscript, when relatively simple technical details have such a strong influence on LFP band activity in different cortical layers. The authors should at least show that their main results are robust and not impaired by the referencing scheme of the electrodes, to proof that the functional interpretations are valid.

We appreciate the efforts of the Editor and the three anonymous referees for their evaluation of the latest version of our study. With this letter, we present a newly revised version of the manuscript. Thanks to the constructive inputs from the reviewers we are confident that the current revision is a step forward in terms of depth, transparency, and readability. An overview of the changes included in this revision is given below:

- The text was proof-read and corrected, with particular focus on increasing readability and consistency of terminology. Several suggestions made by reviewers were included almost verbatim.
- As suggested by Reviewer #1, the title of the manuscript was changed to: “*Echolocation-related reversal of information flow in a cortical vocalization network*”.
- To address remarks of Reviewer #2, we included a new analysis examining the relationship between sound-onset latency (i.e. time) and response strength increase, as a part of the electrical stimulation experiments. These results are depicted in a new figure (Fig. 5). We confirm that auditory cortical response strength increase depends on depth and sound latency, more evidently so in response to echolocation pulses. The modulation of response strength appears to possess intrinsic temporal properties, which we hypothesize could be explained by temporal dynamics in local inhibitory circuits of the AC interacting with top-down inputs from FAF. These views are detailed below, and also discussed in the main text.
- The concerns expressed by Reviewer #3 were taken in full consideration. *We performed analyses on our dataset and were able to fully replicate Weineck et al.’s results by adopting a similar referencing scheme, offline.* These results validate that the outcomes of both studies are not artefactual, but complementary. They highlight the importance of considering high-correlated low-frequency activity in the FAF when determining a reference paradigm. Moreover, with the same *offline* referencing approach, we replicated our original functional connectivity results based on phase transfer entropy analyses. Thus, our data are robust to the choice of reference, and complement our previous observations in *C. perspicillata*’s FAF (Weineck et al., 2020). As we considered that the reviewer’s concerns might be shared by future readers, we included these analyses in a new supplementary figure (Fig. S13). The above is discussed in detail in the response to the reviewer.

Below, the reviewers and the Editor will find a point-by-point response to each comment, as well as a nuanced description of all modifications introduced in the current revision.

Point-by-point response to reviewers

Reviewer #1 (Remarks to the Author):

In this study, the authors use local field potential recordings to demonstrate that the timing and spatial pattern of oscillations in the FAF-AC network of bats predict the purpose of vocalization, i.e. either echolocation or communication. Interestingly, the flow of information shifts from the auditory cortex to the frontal cortex only when the animal is producing echolocation sounds. These are intriguing results and the data analysis methods open the door to future investigations of vocal control and auditory processing reflected within population activity in the cortex of echolocating bats. The findings are also important for understanding general principles of information processing within the auditory cortex. In this regard, publication of these results will hopefully facilitate further studies on this phenomenon.

The work reported here is original and builds on previously published literature. There are no apparent flaws in the data analysis interpretation and conclusions and the methodology is sound. Though the original version of the manuscript lacked some explanation and details, the authors have done an excellent job of addressing reviewer comments in detail and revising the manuscript, including several figures, based upon new data analysis as well as experiments that make the findings more reliable and explain potential discrepancies with the previously published studies.

I do not have any additional major comments, but would like to caution the authors to be consistent in their use of terminology and carefully check the manuscript for ease of readability before final submission.

A few minor points that can be corrected are noticed noted below.

The title reads better when modified to "Echolocation-related reversal of information flow in a cortical vocalization network"

The reason for this change is that echolocation is a behavior involving pulse-echo pair evaluation and the behavior itself does not reverse information flow; rather, the emission of echolocation pulses reverses information flow. Therefore, "echolocation-related reversal" is a more accurate description of the observed phenomenon.

In the abstract, at line 28, please insert "... information transfer patterns during the post – vocal period specific to echolocation pulse emission.

Line 30" "... sounds in the auditory cortex."

Line 391: do you mean qualitatively similar variation (rather than "differences") within echolocation and

communication" ?

Line 393: Change to: "The act of emitting echolocation pulses therefore triggers unique patterns of information-flow reversal in the fronto-auditory network of ...".

Line 395: Correct to: "Our data indicate that the functional connectivity in this ..."

Line 396: Change to: "... by the behavioral intent of vocalization."

Line 404: Change to: "... interactions in the bat's FAF-AC network are associated with auditory processing for echolocation."

Line 498: "... an HF-communication call was an utterance with ..." – delete "communication" (redundant)

We thank the reviewer for their evaluation of our study, and for the constructive and very helpful comments in both revisions. The reviewer's suggestions were adopted in our manuscript verbatim from the above remarks. Furthermore, in accordance with the referee's suggestions, we proof-read the text to ensure consistency of terminology, thus improving the manuscript's transparency and readability.

Reviewer #2 (Remarks to the Author):

I have read the revised version of this manuscript, and overall it is improved. The two issues that I had been most concerned about in the first version were the notion of causality of FAF-AC interactions and whether the different frequencies of sound stimulation relative to the locations of the recordings could account for the differences seen between echolocating calls and social calls. The authors have adequately addressed the latter concern.

With respect to causality, I commend the authors for doing a difficult stimulation experiment, that is now in Figure 4. I am somewhat confused by the analysis and results. The main thrust of the paper is to quantify FAF-AC interactions and the authors have described a processing loop whereby FAF "prepares" AC prior to the stimulus and then AC "updates" FAF after the stimulus. Stimulation of FAF would be expected to scramble the message. I would expect that the metrics of FAF-AC connectivity that are described throughout the paper would have been disrupted by this manipulation. Did the authors examine these metrics? How do the authors interpret the increase in response strength? Does this imply that FAF tonically inhibits AC responses? The authors should also describe how they came to this particular electrical stimulation paradigm – do they expect that FAF neurons would be stimulated or suppressed by it? Can the authors speculate as to the long latency of the impact on the AC? Finally, the image itself is

somewhat hard to read. I think it would be useful to color-code the latency in a way that the reader can look at the image and know whether there is a systematic effect of timing on the response metric. For example, looking at 4F, the reader needs to go back and forth from the legend to the figure to understand the relationship between time and response magnitude. Better yet, the authors should plot time on the x-axis and plot a suitable metric (e.g., response strength or connectivity metric) on the y-axis.

Thank you for assessing the revised version of our manuscript. We are very grateful for the reviewer's constructive remarks.

Indeed, one of our main objectives with the stimulation experiments was to examine how manipulating the FAF would alter connectivity in the fronto-auditory circuit. Like the reviewer, we hypothesized some form of connectivity disruption linked to the electric microstimulation. Unfortunately, as we mentioned in a previous response (the former revision), although very useful for perturbing the circuit, electric microstimulation did not allow to consistently measure FAF signals while -and shortly after- electrically stimulating. We believe this is a caveat of using the same A16 laminar probe for simultaneous stimulation and recording. Because of the latter we did not calculate connectivity matrices, but focused on how FAF stimulation modulates AC responses. This line of thinking was inspired by one of the reviewer's very insightful comments in the previous review round, when it was mentioned that it would be important to determine if perturbations could alter how AC updates FAF, or how AC processes incoming information. Our main result was that rhythmic FAF stimulation enhances AC responses to echolocation sounds, more strongly at specific sound-onset latencies.

It is difficult to provide an interpretation of the increase in AC response strength without major speculation. However, the inhibition hypothesis is very much a plausible one. In the literature it is often considered that the top-down control of sensory areas, associated to some sort of low-frequency oscillatory functional connectivity (e.g. coherence or Granger causality), is ultimately related to top-down modulated inhibitory mechanisms. How exactly this occurs depends on the nature of the top-down connection. For example, long-range inhibitory neurons might target excitatory neurons in the modulated area, or excitatory neurons in the modulating region might target inhibitory circuits on the receiver side. The former is somewhat more unlikely as inhibitory connections in cortex are typically local; the latter seems to be a more plausible scenario (e.g. (Sakata and Harris, 2009)).

Importantly, to properly unravel the former, it is required to understand the consequences of bi-phasic electrical pulse microstimulation locally in FAF. It matters to comprehend what type of neurons are mostly affected by the manipulation, as well as how they are affected (e.g. would their response be enhanced or suppressed?). Typical electrical microstimulation outcomes in the literature show an overall increase of neural activity in the stimulated area (e.g. (Atencio et al., 2014; Takahashi et al., 2019; Yazdan-Shahmorad et al., 2011); see (Tehovnik et al., 2006) for a concise review). Therefore, we can speculate that electrically stimulating the FAF produces an increase in activity in the frontal region. However, the problem persists if one does not understand whether the neurons at hand are mostly inhibitory or excitatory. It is so far unknown what cell types are predominant in the bat's FAF, but recent data from the

mouse PFC suggests a majority of excitatory neurons (Bhattacharjee et al., 2019). It also remains unknown the precise nature of the (direct or indirect) projection into the AC (is it to excitatory networks, to inhibitory microcircuits in cortex, or both?).

Our interpretation is the following: we propose that top-down projections in the FAF reach the AC and modulate cortical gain by affecting the temporal patterns of local inhibitory networks. These networks appear to possess intrinsic rhythms, of which the periodicity that we observe when analyzing sound-onset latency vs. response strength increase (see Fig. 5a in the revised version; we also elaborate below) might be a reflection. It could also account for why, in the case of echolocation, effects of the distant manipulation are seen more than 500 ms after the last electrical pulse (note the sound onset latency of 635 ms in Fig. 4i, and Fig. 5a). In short, when a pulse sequence (with 2 Hz repetition rate in our study) is delivered to the FAF, circuits in the local modulatory networks in AC may be altered via long-range connectivity and set to oscillate at intrinsic frequencies (ca. 4 Hz, at least from our data). However, we stress that the above is yet to be empirically verified. What we can show is the existence of a net enhancing effect in AC.

The particular electrical stimulation paradigm was decided based on several factors. As mentioned above, perturbation experiments were inspired by previous comments from the reviewer, and were devised to answer whether responses in AC were affected by FAF manipulation. We chose a 6 pulse sequence with a 2 Hz repetition rate because we observed that the main effects of our original study occurred mostly in the δ -band (e.g. information flow reversal in 2-4 Hz LFPs). Six pulses (and not less) were chosen with the aim of achieving some degree of entrainment in the FAF at δ ; a higher number of pulses in the sequence could not be used given that the paradigm was long and awake animals were only allowed to be in the setup for 4 hours or less (as per experimental permit). A pulse intensity of 2 μ A was chosen after confirming *online* and *in-vivo* that electrical artefacts in AC were at least visually non-existent, in order to minimize possible confounding effects of electrical pulses affecting the AC by passively traveling from FAF. This was also confirmed *offline*, as shown in the manuscript.

In terms of the long-lasting impact in the AC, we examined as long as 885 ms after the last pulse of electrical stimulation. As the reviewer will note in Fig. 5a, for echolocation, still some net response enhancement was present up to this latencies (although the reliability of significant response increase is at this point somewhat deteriorated; see Fig. 4i). Whether this effect would hold for a longer period of time (e.g. at a later sound-onset latency) we do not know. This may depend on the specific time constants of the putative local inhibitory networks in AC that we discussed above.

Finally, we took the reviewer's suggestion regarding the color coding in Fig. 4, and hopefully managed to find a palette that facilitates identifying sound latency without constantly having to refer to the legend. In addition, and also following the reviewer's recommendation, we included a new analysis and a new figure in the text (now Fig. 5). In this figure, we illustrate the relationship between sound onset latency (i.e. time) and percentage response increase (i.e. Estim response strength vs. no-Estim response strength). The

results have been partially described in the above paragraphs, and are outlined in the Results section as follows:

“As suggested by the data depicted in Fig. 4, changes in response strength between Estim and no-Estim conditions depended both on sound onset latency and AC depth (N-way ANOVA tests; for latency, channel and channel*latency: $p < 10^{-6}$; ANOVA detailed statistics given in Tables S1, 2). Fig. 5 illustrates the effects of latency on response increase (i.e. Estim related vs. no-Estim related response strengths, expressed as a percentage) were most evident when considering responses to echolocation sounds (Fig. 5a). Notably, response increase was largest when the echolocation pulse was presented at a latency of 135 ms (see also Fig. 4i). Response strength increase depended on sound latency with a certain degree of periodicity (e.g. compare sound latencies of 135, 385, 635, and 885 ms with others), with approximately double the frequency of electrical stimulation (4 Hz). Thus, electrical stimulation of the FAF increased responses in AC with a long-lasting effect (hundreds of milliseconds) exhibiting a particular temporal pattern, suggesting that response enhancement is potentially mediated by local circuits in auditory cortex with intrinsic properties (see below).”

The following was also added to the Discussion:

“The precise mechanisms by which FAF microstimulation affects AC response remains to be clarified. It has been suggested that top-down interactions modulate sensory areas by means of inhibitory effects potentially capitalizing on low-frequency interactions (see (Bastos et al., 2012; Sakata and Harris, 2009; Sanchez-Vives and McCormick, 2000)). Given the low-frequency (directed) functional coupling in the FAF-AC network, we hypothesize that top-down projections from the FAF reach the AC and modulate cortical gain by affecting the temporal dynamics of local inhibitory networks (e.g. (Elhilali et al., 2004)). These networks might possess intrinsic, low-frequency rhythms, potentially reflected in the quasi-periodic pattern seen in the relationship between sound-onset latency and AC response increase (**Fig. 5a**). Nevertheless, such observations remain speculative and require empirical validation.”

Reviewer #3 (Remarks to the Author):

Garcia-Rosales et al., Echolocation reverses information flow in a cortical vocalization network, NCOMMS-21-19228A-Z

Generally, the manuscript has improved a lot. I am satisfied with the authors' responses to most of my comments.

However, I am still concerned about two points:

1) I am still not fully convinced that the spectral differences between communication and echolocation calls are not a main source of differences in the spectral dynamics of pre-vocal LFPs. Although the HF-communication calls shown in the revised manuscript lack the strong low frequency component of the LF-

communication calls, they are still largely different from the echolocation calls. I think the spectral center of gravity would be approximately 20 kHz lower. Furthermore the echolocation calls have most of their energy concentrated in a relatively narrow frequency band (spectral peak shown in S3a) while the HF-Communication calls are more or less broadband without prominent spectral peak.

I appreciate, that the authors consider this problem a bit more in the revised manuscript by stating, that “pre-vocal spectral differences are not fully accounted for by differences in the spectral content of echolocation and communication calls” (line 146). And I am fully aware that it is very challenging to experimentally determine the influence of the spectral content of vocalizations used in different behavioral contexts precisely.

Therefore, the authors should at least explicitly state in the manuscript, that an influence of differences in the spectral content of echolocation and communication calls on LFPs cannot definitely be excluded.

We thank the reviewer for taking the time to evaluate our revised manuscript, and for their constructive remarks that have led to improving the transparency of our results.

It is very challenging to empirically determine the influence of the calls’ spectral content in our data for various reasons. We followed the reviewer’s recommendation and included in the main text what was suggested to us. The section in the corrected version of the manuscript reads as follows:

“Thus, pre-vocal spectral differences are not fully accounted for by differences in the spectral content of echolocation and communication utterances. However, an influence of differences in the spectral content of echolocation and communication calls on LFPs cannot definitely be excluded.”

2) I am bit puzzled about the response of the authors to my comment (5) concerning the differences of LFP frequency band power in their different publications. Actually, their line of argumentation in the answer to my comment strengthens my initial concerns. Obviously, technical details of the LFP recording procedures have profound influence on results, and thus on the interpretation of data in a functional context. How is the referencing scheme for the recordings in the current paper? I could not find information on this in the methods section. Furthermore, which of the two referencing schemes (top electrode as reference vs. other scheme) is the most appropriate one, and why? It is still difficult for me to accept the complex functional interpretations made in the current manuscript, when relatively simple technical details have such a strong influence on LFP band activity in different cortical layers. The authors should at least show that their main results are robust and not impaired by the referencing scheme of the electrodes, to proof that the functional interpretations are valid.

We fully understand the concerns raised by the reviewer in the above point. However, we are certain that the functional interpretations derived from our data are valid, and are not an artefact of the referencing schemes used. Indeed, differences between our study and that of Weineck et al. regarding the predictive

power of low-frequency LFPs can be *fully* accounted for by the choice of reference, without being spurious electrical effects devoid of functional implications.

We invite the reviewer to examine **Fig. R1**, which is also now part of the manuscript as **Supplementary Figure 13**. This figure depicts results of analyses performed with the original dataset of the current manuscript, but with a simple alteration: the referencing scheme of Weineck et al. was adopted *offline* by merely subtracting from FAF field potentials the signal of the top channel in that structure (**Fig. R1a**). Assuming the top FAF channel as a reference disproportionately affects low-frequency LFPs in the frontal cortex, which are highly correlated across electrodes (compare **Fig. R1a** with Fig. 1d from the main text). As a consequence, low-frequency power fails to predict ensuing call type (i.e. echolocation or communication), without affecting the predictive power of gamma-band (γ_2) activity (**Fig. R1b**). We confirmed this statistically by running the same GLM protocol on our altered dataset (**Fig. R1c**).

Crucially, when applying the same GLM procedure on Weineck et al.'s data, a strikingly similar pattern of prediction emerges (**Fig. R1d**; compare to panel c). In other words, our *offline* referencing scheme *reliably replicates the results of the previous study, indicating that the lack of low-frequency effects in Weineck et al. were a consequence of using the top FAF channel as a reference*. In that study, such referencing was done via physical implementation in the recording setup.

Fig. R1. (Replicated from Fig. S13). Referencing schemes account for dissimilarities between the current study and that of Weineck et al. (2020). (a) LFPs in the bat FAF during vocalization (echolocation and communication) across various depths. The data is the same as that presented in

Fig. 1d; however, the top channel in the FAF was used as a reference for LFPs in that structure (yielding a flat, zero-line at the topmost trace). The referencing was done offline (i.e. in the post-processing stage) by simply subtracting the LFP of each FAF channel from the first channel in the same structure. This referencing scheme mostly affects low frequency oscillations, which are highly correlated across electrodes in FAF (e.g. see **Fig. 1d**). **(b)** As a consequence, differences between pre-vocal power changes related to the utterance of echolocation and communication calls in low frequencies (e.g. δ , θ , or α) are strongly affected (c.f. **Fig. 1f**). For γ_2 frequencies, however, pre-vocal power differences are still evident. Power changes preceding the vocalization of an echolocation or a communication call are shown across all channels in FAF ($n = 138$ echolocation, blue; $n = 734$ communication, orange; data as mean \pm sem). **(c)** When using the top channel in the FAF as reference, low-frequency (δ - α) pre-vocal power changes in FAF lose their predictive power to a great extent (c.f. **Fig. 1g**). Nevertheless, LFPs in the γ_2 -band continue to predict whether animals produce echolocation or communication utterances. **(d)** Analysis of the data from Weineck et al. (2020), where the top FAF channel was used as a reference online (i.e. determined by hardware in the setup), with the same GLM paradigm used in the current study. Note the similarities between the predictive power of the pre-vocal LFP at various frequencies between Weineck et al.'s dataset and that of our current dataset with altered referencing (panel **c**). These analyses demonstrate that dissimilarities between the spectral patterns observed in FAF field potentials between both studies can be attributed to the nature of the referencing schemes used. Low frequency effects may have been overlooked in the study by Weineck et al. as a consequence of using the top FAF channel as reference, on account of the well-correlated low-frequency activity across electrodes in the frontal cortex. **(e)** In terms of the functional connectivity measured with the dPTE metric, we still observe information flowing top-down (FAF \rightarrow AC direction) during pre-vocal periods for δ -frequencies of the LFP. Information flow reverses after echolocation in this frequency band, but not after communication vocalization. Thus, imposing an artificial top-channel referencing scheme does not drastically affect our original observations, namely an echolocation-related reversal of information flow in the FAF-AC network. Data in the dPTE matrices is shown following the conventions of **Fig. S4** (note the legend in the panel).

While the referencing in Weineck et al. precluded the uncovering of low-frequency effects in pre-vocal LFPs for predicting call type, the findings from Weineck et al. remain valid in other frequency bands (such as gamma). The information regarding referencing is mentioned in Weineck et al in the methods section where we said:

“The reference of each electrode array was short-circuited with the respective top recording channel (the electrode closest to the brain surface) to obtain local signals and prevent movement artefacts.”

The same is true for the current revision of the present manuscript. We apologize for not including this information in the original version. We wrote (please note the text in italics):

“Probes were connected to a micro-preamplifier (MPA 16, Multichannel Systems, MCS GmbH, Reutlingen, Germany), and acquisition was done with a single, 32-channel portable system with integrated digitization (sampling frequency, 20 kHz; precision, 16 bits) and amplification steps (Multi Channel Systems MCS GmbH, model ME32 System, Germany). *Reference electrodes (silver wires) were used for each recording shank (i.e. in FAF or AC) at a different area of the brain (for FAF: a non-auditory lateral, ipsilateral region; for the AC: a non-auditory occipital, ipsilateral region). Wires were carefully placed to rest between the skull and the dura matter. For each laminar electrode, reference and ground were short-circuited; the ground was however common in the acquisition system (i.e. the ME32).* Acquisition was online-monitored and stored in a computer using the MC_Rack_Software (Multi Channel Systems MCS GmbH, Reutlingen, Germany; version 4.6.2).”

As for which referencing scheme is better, we do not think that one scheme is better than the other, in the same way that we do not believe in a perfect referencing scheme suited for all electrophysiological approaches. All schemes have advantages and disadvantages. A highly localized reference, as the one used in Weineck et al., allows for “cleaner” recordings when phenomena like movement artefacts are a concern. For example, several spike detection algorithms and EEG pipelines subtract a global median from all channels to minimize external noise. This comes at the cost of highly-correlated, yet non-artefactual activity, which is also a known trade-off when performing current source density (CSD) analyses. Such procedures are extremely helpful when signal sources are sufficiently separated, but they can lead to misses when electrode separation is in the order of micrometers. Experiments from our previous study were the first recordings we ever made in vocalizing bats and at the time we were concerned about possible movement noise. We thus used the top channels in the frontal cortex and the caudate nucleus as local references from each structure. Again, this is not incorrect, it is just more local than the referencing scheme used here. Nevertheless, our current data make it clear that the effect of highly-local references must be taken into account when examining low-frequency oscillations in the bat's frontal cortex.

For the current manuscript, the FAF and AC electrodes had independent references located outside of the areas from which recordings were done (for FAF: a non-auditory lateral, ipsilateral region; for the AC: a non-auditory occipital, ipsilateral region). These references were short-circuited with the ground of each laminar electrode (a common procedure in electrophysiology), while the recording hardware (Multichannel Systems ME32) forced a common ground on both recording shanks. This approach, albeit relatively more vulnerable to movement noise, does not affect well-correlated activity across channels within FAF or AC. However, signals appearing on the ground or caught by reference wires might become common in the recordings from both structures, which may lead to possible misinterpretations of common ground signals as correlated neural activity across regions (i.e. recorded from the two shanks). This did not happen in our current study.

Here, we demonstrate patterns of activity that are specific to frontal and auditory cortex, with complementary vocalization type specific dynamics across regions both in LFP frequency and in prediction effect. Inter-areal correlations due to common ground/reference appear as zero-lag

synchronization between the regions of interest, yet in the past we have demonstrated using a similar referencing scheme the existence of lagged FAF-AC coherence (García-Rosales et al., 2020). In the current manuscript, the phase transfer entropy (PTE) approach measures efficient functional connectivity that is lagged between regions and which cannot be solely attributed to first-order correlations across signals. Note that, as mentioned in the Methods section, a signal X casually influences another signal Y (in a transfer entropy sense) if the uncertainty about the future of Y can be reduced from knowing both the past of signal X and signal Y, as compared to knowing the past of signal Y alone. By this definition, lagged interactions play a crucial role.

We tested whether our main results from the dPTE were robust to imposing a top-channel referencing scheme on the original dataset in FAF. As shown in **Fig. R1e**, during pre-vocal periods information flowed predominantly in the FAF→AC direction (i.e. top-down), both in the echolocation and communication case. Echolocation-related information flow reversal occurred during post-vocal periods, when information flowed predominantly in the AC→FAF direction (i.e. bottom-up) after an animal emitted a sonar pulse. Such flow reversal was unique to echolocation production. The reviewer may note that these data afford the same conclusions advanced in the original manuscript.

As previously mentioned, we understand the reviewer's concerns and moreover consider they could be shared by future readers. Therefore, we decided to include **Fig. R1** in the supplementary materials of the manuscript (**Fig. S13**). As mentioned above, a description of the referencing scheme has also been added to the current revision of the study.

Finally, Supplementary Figure 13 is introduced in the main text in the Results section. The segment reads as follows (kindly note the text in the italics):

*“Low- and high-frequency power increase (mostly in the δ - α and γ_2 bands) in FAF predicted whether animals produced echolocation or communication calls, typically with moderate effect sizes ($p < 0.05$; $R^2_m \geq 0.1$), highest in middle-to-deep electrodes (i.e. depths $> 300 \mu\text{m}$; **Fig 1g**, left). *The fact that low-frequencies predict ensuing call types complements a previous study in C. perspicillata's FAF (Weineck et al., 2020), wherein low-frequency effects went unnoticed likely due to a local referencing scheme that affected low-frequency signals correlated across neighbouring electrodes (see Fig. S13).*”*

We hope that this increases the transparency of our study and aids the reader while navigating our complementary results.

References

- Atencio, C.A., Shih, J.Y., Schreiner, C.E., and Cheung, S.W. (2014). Primary auditory cortical responses to electrical stimulation of the thalamus. *J Neurophysiol* *111*, 1077-1087.
- Bastos, A.M., Usrey, W.M., Adams, R.A., Mangun, G.R., Fries, P., and Friston, K.J. (2012). Canonical microcircuits for predictive coding. *Neuron* *76*, 695-711.
- Bhattacharjee, A., Djekidel, M.N., Chen, R., Chen, W., Tuesta, L.M., and Zhang, Y. (2019). Cell type-specific transcriptional programs in mouse prefrontal cortex during adolescence and addiction. *Nat Commun* *10*, 4169.
- Elhilali, M., Fritz, J.B., Klein, D.J., Simon, J.Z., and Shamma, S.A. (2004). Dynamics of precise spike timing in primary auditory cortex. *J Neurosci* *24*, 1159-1172.
- García-Rosales, F., López-Jury, L., González-Palomares, E., Cabral-Calderín, Y., and Hechavarría, J.C. (2020). Fronto-Temporal Coupling Dynamics During Spontaneous Activity and Auditory Processing in the Bat *Carollia perspicillata*. *Frontiers in Systems Neuroscience* *14*.
- Sakata, S., and Harris, K.D. (2009). Laminar structure of spontaneous and sensory-evoked population activity in auditory cortex. *Neuron* *64*, 404-418.
- Sanchez-Vives, M.V., and McCormick, D.A. (2000). Cellular and network mechanisms of rhythmic recurrent activity in neocortex. *Nat Neurosci* *3*, 1027-1034.
- Takahashi, S., Muramatsu, S., Nishikawa, J., Satoh, K., Murakami, S., and Tateno, T. (2019). Laminar responses in the auditory cortex using a multielectrode array substrate for simultaneous stimulation and recording. *Ieee T Electr Electr* *14*, 303-311.
- Tehovnik, E.J., Tolias, A.S., Sultan, F., Slocum, W.M., and Logothetis, N.K. (2006). Direct and indirect activation of cortical neurons by electrical microstimulation. *J Neurophysiol* *96*, 512-521.
- Weineck, K., Garcia-Rosales, F., and Hechavarría, J.C. (2020). Neural oscillations in the fronto-striatal network predict vocal output in bats. *PLoS Biol* *18*, e3000658.
- Yazdan-Shahmorad, A., Lehmkuhle, M.J., Gage, G.J., Marzullo, T.C., Parikh, H., Miriani, R.M., and Kipke, D.R. (2011). Estimation of electrode location in a rat motor cortex by laminar analysis of electrophysiology and intracortical electrical stimulation. *J Neural Eng* *8*, 046018.

Reviewers' comments:

Reviewer #2 (Remarks to the Author):

I have re-read the manuscript and the authors' comments. The new analysis in Fig is very interesting and adds another interesting dimension to the FAF stimulation data. My concerns at this point are minor:

Fig 4a: Please add some kind of x-axis under the waveform on the left so the reader has a more intuitive sense as to the temporal relationships of all of the events.

Fig 4 f and g: Is there a reason that these two are split out based on latency? If it is just for clarity, then please state so in the text somewhere.

Line 277 – do the authors mean Fig 4f?

Line 1116 – should be “confidence intervals”

Reviewer #3 (Remarks to the Author):

I can now accept the revised manuscript.

We would like to thank the three anonymous referees for their evaluation of our study and for the constructive criticism provided. We submit a revised version of the manuscript, taking into consideration the latest remarks from the reviewers. All comments from Reviewer #2 were addressed, and can be found answered point-by-point below.

Finally, we proof-read the text once again to ensure consistency, transparency and readability.

Reviewer #2 (Remarks to the Author):

I have re-read the manuscript and the authors' comments. The new analysis in Fig is very interesting and adds another interesting dimension to the FAF stimulation data. My concerns at this point are minor:

Thank you for your comments and the all feedback provided.

Fig 4a: Please add some kind of x-axis under the waveform on the left so the reader has a more intuitive sense as to the temporal relationships of all of the events.

Thank you for this suggestion. We added a temporal scale below the left waveform for the benefit of readers.

Fig 4 f and g: Is there a reason that these two are split out based on latency? If it is just for clarity, then please state so in the text somewhere.

Indeed, these two panels are split for clarity. This was specified the text.

Line 277 – do the authors mean Fig 4f?

Thank you: we did mean Fig. 4f. This has been corrected.

Line 1116 – should be “confidence intervals”

Thank you for noticing this. We corrected the typo.

Reviewer #3 (Remarks to the Author):

I can now accept the revised manuscript.

We take this opportunity to thank the reviewer for the provided feedback and comments. We believe that the reviewer's insights contributed largely to the transparency and solidity of our work.